# Repeated plague infections across six generations of Neolithic Farmers

Frederik Valeur Seersholm[1 ✉], Karl-Göran Sjögren[2], Julia Koelman[3], Malou Blank[2], Emma M. Svensson[3], Jacqueline Staring[4], Magdalena Fraser[3], Thomaz Pinotti[1,5], Hugh McColl[1], Charleen Gaunitz[1], Tatiana Ruiz-Bedoya[3,6], Lena Granehäll[3,7], Berenice Villegas-Ramirez[3], Anders Fischer[8], T. Douglas Price[2], Morten E. Allentoft[1,9], Astrid K. N. Iversen[10], Tony Axelsson[2], Torbjörn Ahlström[11], Anders Götherström[12,13], Jan Storå[13], Kristian Kristiansen[1,2], Eske Willerslev[1,14], Mattias Jakobsson[3,15], Helena Malmström[3,15] & Martin Sikora[1 ✉]

In the period between 5,300 and 4,900 calibrated years before present (cal. BP), populations across large parts of Europe underwent a period of demographic decline[1,2]. However, the cause of this so-called Neolithic decline is still debated. Some argue for an agricultural crisis resulting in the decline[3], others for the spread of an early form of plague[4]. Here we use population-scale ancient genomics to infer ancestry, social structure and pathogen infection in 108 Scandinavian Neolithic individuals from eight megalithic graves and a stone cist. We find that the Neolithic plague was widespread, detected in at least 17% of the sampled population and across large geographical distances. We demonstrate that the disease spread within the Neolithic community in three distinct infection events within a period of around 120 years. Variant graph-based pan-genomics shows that the Neolithic plague genomes retained ancestral genomic variation present in *Yersinia pseudotuberculosis*, including virulence factors associated with disease outcomes. In addition, we reconstruct four multigeneration pedigrees, the largest of which consists of 38 individuals spanning six generations, showing a patrilineal social organization. Lastly, we document direct genomic evidence for Neolithic female exogamy in a woman buried in a different megalithic tomb than her brothers. Taken together, our findings provide a detailed reconstruction of plague spread within a large patrilineal kinship group and identify multiple plague infections in a population dated to the beginning of the Neolithic decline.

The emergence of agriculture during the Neolithization brought about one of the most profound lifestyle changes in the history of modern humans. The shift in subsistence strategy from hunting, fishing and gathering to farming paved the way for a marked increase in population density and the establishment of larger and more permanent settlements[2]. However, the flourishing economy of the Neolithic came to a sudden halt in many regions of Northern Europe around 5300–4900 calibrated years before present (cal. BP)[5], in which a marked reduction in the number of human remains radiocarbon-dated to this period suggests a population decline. Coined the Neolithic decline[1], this demographic bust coincides with the cessation of megalith building in the area and has been suggested to be one of the factors facilitating the Corded Ware expansion into Europe (4800–4400 cal. BP)[6]. Although several scenarios have been put forward, no single driving factor has hitherto been linked to this decline and this enigma is still heavily debated in the literature[3,5]. Nevertheless, recent findings demonstrating that an ancestral form of *Yersinia pestis* was present in Sweden at this time could potentially solve this debate[4].

*Yersinia pestis*, the infectious agent of plague, split from its most recent ancestor *Yersinia pseudotuberculosis* some time within the past 50,000 years and has been infecting humans since prehistoric times. The vast majority of prehistoric plague genomes are from Late Neolithic and Bronze Age (LNBA) individuals dating to 4700–2400 cal. BP (refs. 7–9). These genomes fall within two distinct lineages that can be distinguished by the absence (LNBA⁻) or presence (LNBA⁺) of the *ymt* gene[8]. The *ymt* gene is crucial for the bacterium's survival in the flea digestive tract when the source is an infected mouse, black rat or human, and hence for the development of bubonic plague.

[1]Lundbeck Foundation GeoGenetics Centre, Globe Institute, University of Copenhagen, Copenhagen, Denmark. [2]Department of Historical Studies, University of Gothenburg, Gothenburg, Sweden. [3]Human Evolution, Department of Organismal Biology, Uppsala University, Uppsala, Sweden. [4]Lygature, Utrecht, the Netherlands. [5]Laboratório de Biodiversidade e Evolução Molecular (LBEM), Universidade Federal de Minas Gerais, Belo Horizonte, Brazil. [6]Department of Cell and Systems Biology, University of Toronto, Toronto, Ontario, Canada. [7]Institute for Mummy Studies Eurac Research, Bolzano, Italy. [8]Sealand Archaeology, Kalundborg, Denmark. [9]Trace and Environmental DNA (TrEnD) Laboratory, School of Molecular and Life Sciences, Curtin University, Perth, Western Australia, Australia. [10]Nuffield Department of Clinical Neurosciences, Weatherall Institute of Molecular Medicine, University of Oxford, Oxford, UK. [11]Department of Archaeology and Ancient History, Lund University, Lund, Sweden. [12]Centre for Palaeogenetics, Stockholm University and the Swedish Museum of Natural History, Stockholm, Sweden. [13]Department of Archaeology and Classical Studies, Stockholm University, Stockholm, Sweden. [14]Department of Zoology, University of Cambridge, Cambridge, UK. [15]Palaeo-Research Institute, University of Johannesburg, Johannesburg, South Africa. ✉e-mail: frederikseersholm@gmail.com; martin.sikora@sund.ku.dk

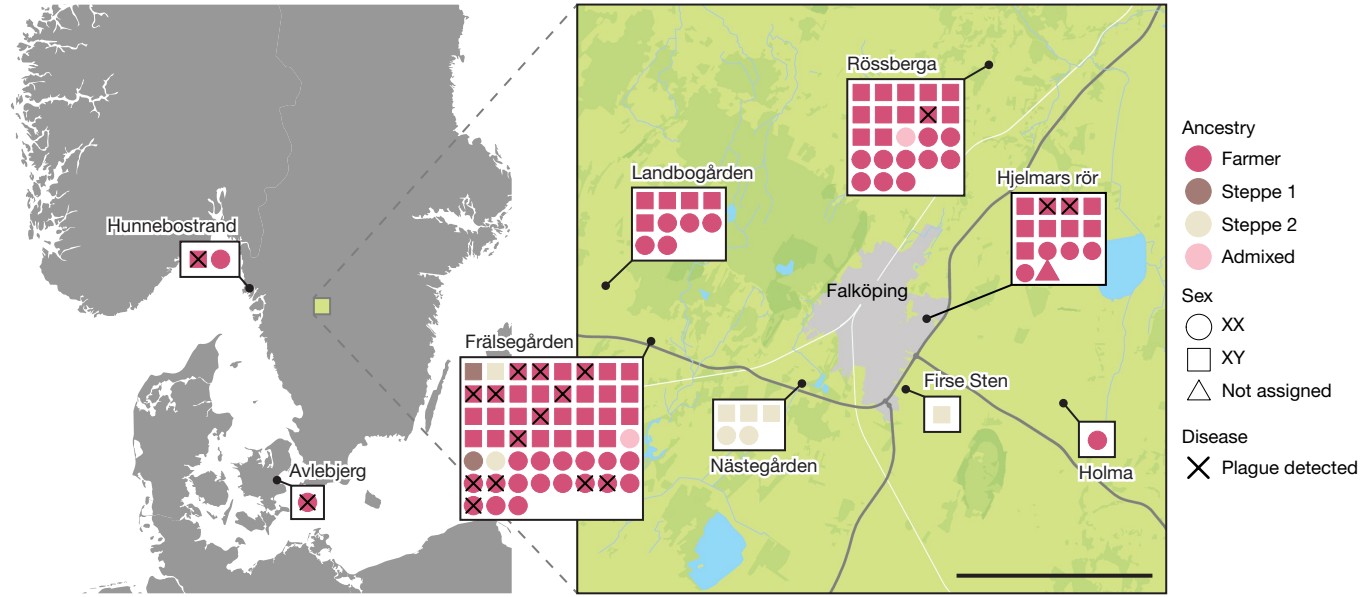

**Fig. 1 | Overview of sample locations.** Each individual is represented by coloured shapes, with squares and circles representing males and females, respectively, and triangles representing sex unknown. Colours indicate genetic ancestry and black crosses designate plague-positive individuals. Scale bar, 5 km.

Until recently, all known prehistoric plague strains fell within these two LNBA clades; however, findings published in Rascovan et al.[4] and Susat et al.[10] demonstrated the presence of earlier diverging *Y. pestis* lineages (pre-LNBA). These two ancestral *Y. pestis* genomes were identified from a Swedish individual with Neolithic Farmer (that is, Anatolian-derived) ancestry (5035–4856 cal. BP)[4] and an individual from Latvia with hunter-gatherer ancestry (5300–5050 cal. BP)[10], respectively. Although these genomes are of very similar age and ancestral to all other plague genomes available, the two studies arrive at different conclusions: Rascovan et al. argue that their finding supports a role of plague in the Neolithic decline whereas Susat et al. conclude that these early plague forms are probably a result of sporadic zoonotic events.

In Scandinavia the Neolithic decline coincides with the disappearance of the Funnelbeaker/Trichterbecher cultural complex (TRB)[11] and the end of the first wave of megalith building. Despite their ubiquity in the Scandinavian landscape, controversy still exists regarding the mortuary practices and social background of people associated with the megaliths. For example, it has long been assumed that these tombs were used by kinship or family groups[12], but currently few data exist to substantiate this hypothesis. Osteological analyses show the presence of both sexes in variable proportions, and also of individuals of different ages at death, consistent with a random selection from a living population. By contrast, the number of buried individuals per generation is small and would suggest that only a restricted segment of the population was interred in these tombs[13–15]. It is also debated whether complete bodies were directly introduced and perhaps later rearranged and disturbed, or whether placement in the chamber was only the final act in a longer series of mortuary rituals performed over a longer time and at different locations[14,16–19]. Furthermore it is not known which factors determined burial location, or how different parts of the tombs were typically used.

To elucidate social structure and plague infection frequency in the Scandinavian Middle Neolithic, we analysed ancient human DNA from nine multi-individual burial structures: seven megaliths in Falbygden, inland western Sweden (Landbogården, Frälsegården, Nästegården, Firse sten, Holma, Hjelmars rör and Rössberga), one megalith on the Swedish west coast (Hunnebostrand) and a stone cist in Denmark (Avlebjerg; Supplementary Note 1). We set out to investigate whether the plague cases from ref. 4 were isolated events or whether there was evidence of plague in more individuals at different sites and in different Scandinavian regions during the Neolithic. Furthermore, we aimed to investigate

kinship and social relations in several of the best-described megaliths from Sweden to better understand potential disease transmission.

## DNA sequencing and population structure

To investigate the disease frequency and geographical distribution of the Neolithic plague, we sequenced 174 samples from ancient human skeletal remains from across Scandinavia (Supplementary Table 2). After merging data from samples that derived from the same individual and exclusion of low-coverage human data (below 0.01×; Methods), we produced a final dataset representing 108 individuals from nine sites (Fig. 1 and Supplementary Table 1). We found a slight male sex bias in the dataset (58% males; Fig. 1), reflected across all sites except from the Rössberga passage grave, where no sex bias was observed.

To investigate the broader population genetic structure in our data, we merged our dataset with a panel of 1,430 ancient shotgun-sequenced genomes (Supplementary Table 7) and visualized genetic affinities using principal component analysis (PCA; Fig. 2c). We found that the vast majority of individuals analysed (*n* = 96) fell within the broad cluster of European Neolithic populations and Anatolian Farmers. This finding is in agreement with our radiocarbon dating results, which date this group to 5200–4900 cal. BP, associating these individuals with the TRB culture of Southern Scandinavia and Scandinavian MN A (Fig. 2a). Furthermore, we also found evidence of two distinct and slightly younger groups of individuals with Steppe-related ancestry. The first group (Steppe 1, *n* = 2) is dated to around 4400 cal. BP (Scandinavian MN B) and the second group (Steppe 2, *n* = 8) to 4100–3000 cal. BP (Scandinavian LN to LBA). This distinction is corroborated by chromosome Y haplogroup results, which indicate that all Farmer ancestry males have the haplogroup I2 whereas the two Steppe ancestry groups are represented exclusively by haplogroups R1 and I1, respectively (Supplementary Table 1). In general we find that each site is represented by people of similar ancestry (Fig. 1), but at one site (Frälsegården) we find evidence of the continued use of the burial chamber by three temporally and genetically distinct populations.

Lastly, we find that two genomes contain a substantial fraction of hunter-gatherer ancestry, suggesting recent admixture between the TRB group and Scandinavian hunter-gatherers (Fig. 2). One woman from Frälsegården (FRA108) appears to have equal proportions of hunter-gatherer and Neolithic Farmer ancestry; we find that she

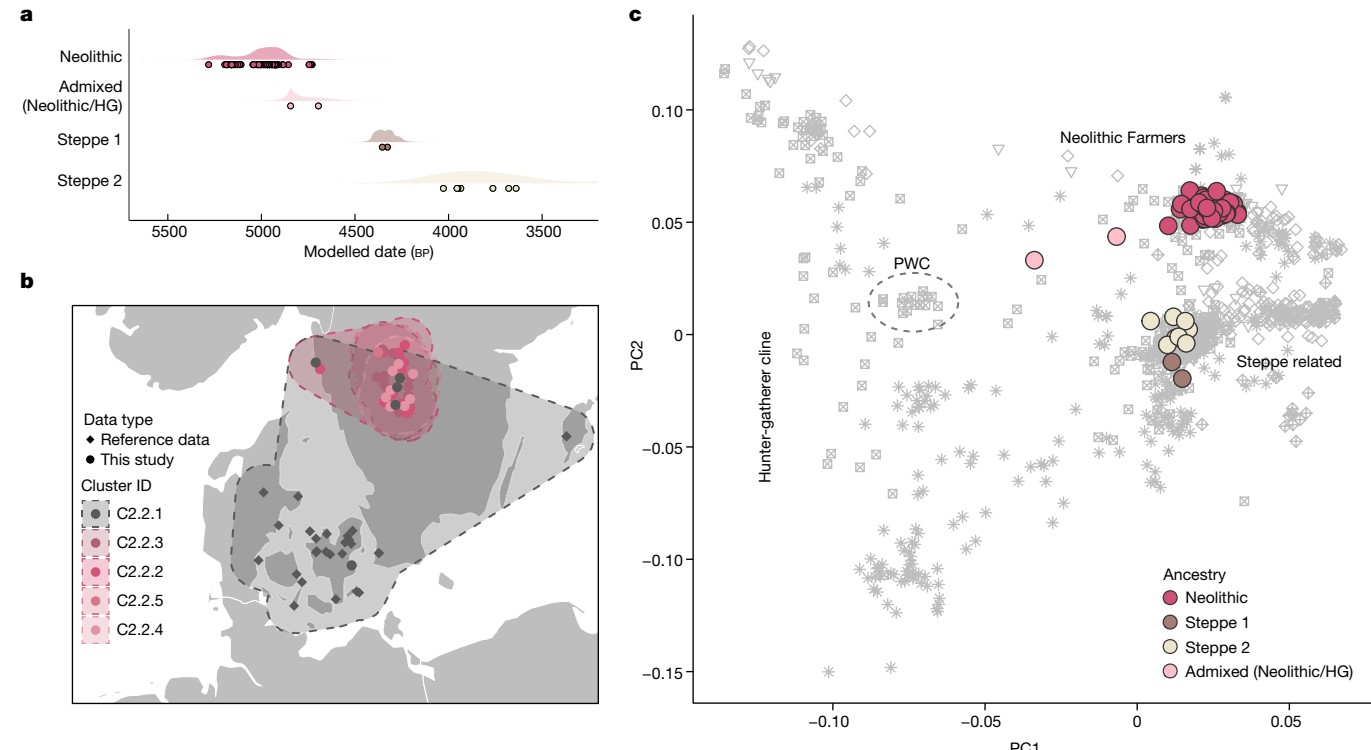

**Fig. 2 | Age and ancestry. a**, Median radiocarbon date (years cal. BP) for samples, grouped by genetic ancestry (Supplementary Tables 1 and 4). Shaded area represents distribution of kernel density estimation for the group. **b**, Geographical distribution of IBD clusters of Scandinavian individuals with Neolithic Farmer ancestry (Supplementary Table 8). **c**, PCA of genomes sequenced in this study (coloured by ancestry) and a reference panel of 810 previously sequenced ancient shotgun genomes from Europe (grey shading; Supplementary Table 7). HG, hunter-gatherer.

was most probably a first-degree offspring of these two distinct sociocultural groups (Extended Data Fig. 1 and Supplementary Note 5). Similarly for the other individual, a woman from the site Rössberga (ROS027), we find roughly 34% hunter-gatherer ancestry and 66% Neolithic Farmer ancestry, suggesting that she might have lived two or three generations after the admixture event. Although northern groups of hunter-gatherers with Mesolithic ancestry cannot be ruled out[20], the most likely source of hunter-gatherer DNA in these two admixed individuals is the Pitted Ware Culture (PWC; 5400–4300 cal. BP), which, in Sweden, overlapped with the TRB culture in both time and space[21,22]. It is notable that both women date to the end of the TRB period, perhaps reflecting demographic crisis within the TRB population and/or a loosening of social and cultural boundaries.

## Distinct Neolithic IBD clusters

To disentangle the fine-scale structure between TRB individuals, we compared identity-by-descent (IBD) sharing across the panel and clustered individuals with close affinities into related IBD groups (Methods). We found that the IBD clustering of Northern European Neolithic Farmers was, to a large extent, driven by the close familial relations within Falbygden. Accordingly, we identified four IBD clusters of individuals from Falbygden (n = 17, 13, 8 and 4) and one cluster of all other Neolithic individuals from Denmark and Sweden (n = 32; Fig. 2b). As expected, most of our samples were clustered into the four Falbygden groups whereas our sample from Denmark (Avlebjerg) fell within the unrelated group from Denmark and Sweden (Fig. 2b and Supplementary Table 8). Interestingly, we also found that three TRB individuals from Falbygden clustered with the Danish/Swedish IBD group, suggesting that these individuals came from outside the Falbygden area. This is supported by Sr isotope data for one of them (FRA106,

adult male, 87 Sr/86 Sr 0.717011), indicating an upbringing outside the Cambro-Silurian bedrock of Falbygden. Moreover, for the two genomes sequenced from the site Hunnebostrand in coastal western Sweden, we found that one genome (HUN002, an adult male) clustered with one of the Falbygden groups whereas the other (HUN001, a female around 70 years of age) clustered with the Danish/Swedish group, suggesting different origins for these two individuals. Sr isotopes in the two individuals confirm different childhood residence (Supplementary Table 5). Both are consistent with childhood in Falbygden, but they could also originate from other places in Scandinavia outside Denmark.

The IBD-sharing results also support the presence of two distinct Steppe-related groups in our dataset discussed above. The early group (chrY haplogroup: R1a1a, age approximately 4400 cal. BP) clusters with a large group of individuals from across Europe of Corded Ware ancestry, including individuals from Battle Axe Culture contexts in Sweden. The later group (chrY haplogroup:I1, age approximately 4100–3000 cal. BP), on the other hand, clusters exclusively with contemporaneous individuals from Eastern Denmark, Sweden and Norway, reflecting results from ref. 23. Using DATES, we were able to date the admixture of 'Steppe' and 'Farmer' DNA in these two groups[24]. For both groups we found that admixture most probably happened around 4750 cal. BP (Extended Data Fig. 2). In agreement with recent results showing that Steppe-related groups first appeared in Eastern Europe around 4,800 years ago[23], this finding suggests that admixture occurred in a single pulse before the arrival of Corded Ware complex (CWC) groups in Sweden.

## Social structure across four pedigrees

We investigated close familial relations in the dataset and categorized pairs of individuals as either first-, second- or third-degree relatives (Methods). Using these data we were able to reconstruct one large

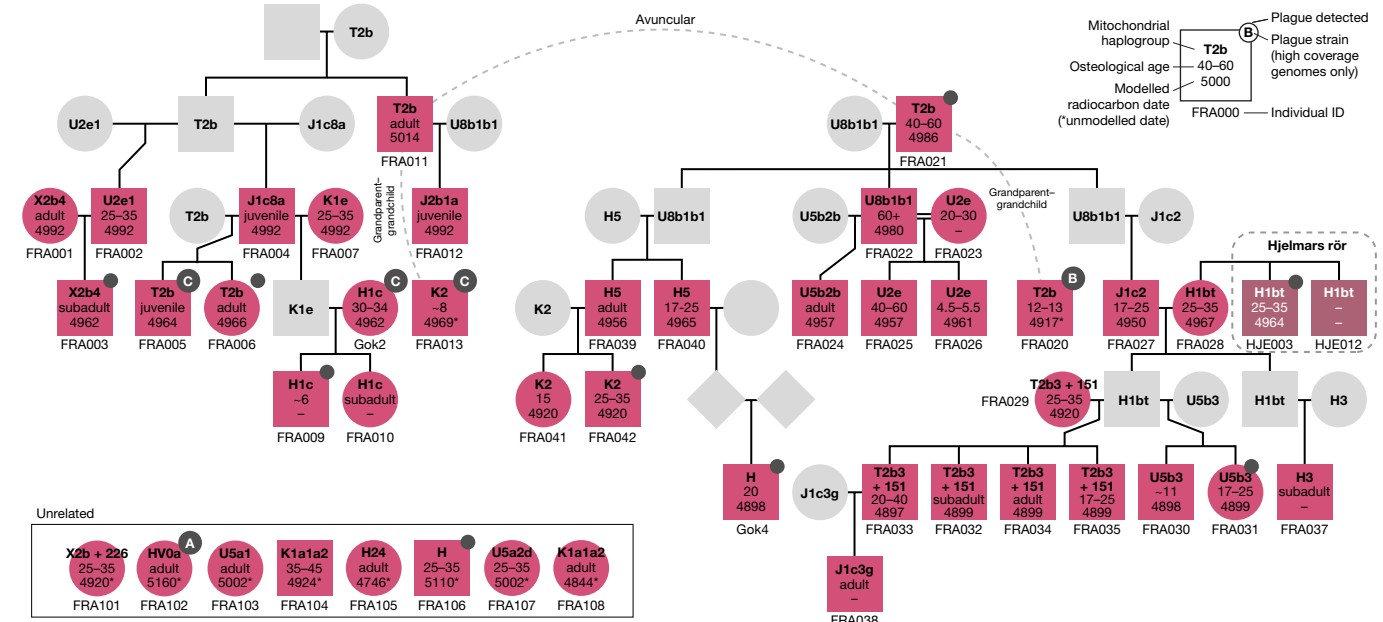

**Fig. 3 | Pedigree 1 reconstructed from genomic data.** Squares and circles represent males and females, respectively, and diamonds indicate sex unknown. Information on mitochondrial haplogroup, osteological age estimation and modelled radiocarbon date is indicated inside each shape. Pink and dark mauve indicate the sites Frälsegården and Hjelmars Rör, respectively, with grey representing inferred individuals. Solid black lines indicate first-degree relations, dashed grey lines signify unknown second-degree relationships and double black lines indicate mating between related individuals. Supplementary Fig. 2 shows the full pedigree, including individuals with uncertain kinship. For unrelated individuals, radiocarbon dates are not modelled but are reported only as the median of the calibrated date.

and three smaller pedigrees from the sites Frälsegården, Hjelmars Rör, Landbogården and Rössberga (Fig. 3 and Extended Data Fig. 3). The vast majority of individuals that we were able to place in a pedigree were excavated at the Frälsegården passage grave, which is also the most densely sampled site, with 54 genomes sequenced out of an estimated total of 78 individuals buried at the site[19]. The Frälsegården pedigree includes 61 individuals in total (38 sampled and 23 inferred individuals) spanning six generations. The pedigree comprises two sublineages/subfamilies (left and right side of Fig. 3, respectively). We found that the subfamily to the left had an unsampled progenitor whereas one male progenitor was found to be the ancestor of all male lineages in the subfamily to the right, through his three sons.

The pedigree is strongly patrilineal in nature and, except for a single woman (FRA023), all female individuals with offspring appear to come from outside the lineage. In fact, in one case we find direct evidence of female exogamy: we identified three siblings—two brothers and their sister (HJE003, HJE012 and FRA028, respectively)—in which the brothers were buried at the site Hjelmars Rör (highlighted in dark mauve in Fig. 3, dashed box) whereas their sister was buried at Frälsegården, 8 km distant. At Frälsegården this female gave rise to a large family with seven grandchildren, indicating that she moved away from her family during her lifetime to start her own family in a new settlement. Based on Sr isotope ratios, for which no significant difference between males and females was observed (Supplementary Table 5), we suggest that such short relocations within the Falbygden area were common.

In addition, it has previously been suggested that inbreeding, although infrequent, occurred in the Neolithic[25,26]. We identify direct evidence of such inbreeding in two brothers (FRA009 and FRA010) who are the offspring of third-degree relatives (indicated by double black lines in Fig. 3). The close relatedness of the parents is confirmed by significantly elevated levels of long runs of homozygosity (ROHs) in their two children as compared with the remainder of the population (Extended Data Fig. 4).

We identify several other individuals that were related to individuals of this kin group but cannot be placed with confidence in the pedigree (Supplementary Fig. 2). Moreover, of all the individuals sampled from Frälsegården, we find that only eight were not related to anyone else at this site and, of these, six are women. This finding confirms the patrilineal social structure at the site and suggests that these six women were married into the family but did not produce offspring who were buried within the tomb. Although it is possible that these women did not give birth before dying, it is perhaps more likely that all their offspring were daughters who moved away and were buried in other tombs. Furthermore, three of the unrelated individuals also appear to carry slightly different ancestry than the rest of the group—the two unrelated males (FRA104 and FRA106) clustered with the Swedish/Danish IBD group discussed above, whereas a female (FRA108) is one of the individuals with both hunter-gatherer and farmer ancestry.

The chronological span of the six-generation pedigree at Frälsegården can be estimated at approximately 150–180 years if we assume a mean generation length of 25–30 years[27]. Because many of the individuals have been directly dated, we also estimated the chronology by Bayesian modelling. This gave very similar results and supported the overlapping datings of the two branches at both Frälsegården and Landbogården, with a potentially earlier start of the Frälsegården left branch (Extended Data Fig. 5, Supplementary Figs. 7–9, Supplementary Note 3 and Supplementary Tables 11–13).

## Burial locations and kinship

Based on the burial locations of each individual within the Frälsegården passage grave, we find a clear connection between kinship and burial location. The northern part of the passage grave appears to hold generations one and two in the pedigree. These individuals have received a very special treatment. The cranium of the male progenitor (FRA021) is buried beneath a limestone slab beside the crania of his son (FRA022) and daughter-in-law (FRA023) (Extended Data Fig. 6). Furthermore, a maxilla fragment of a male individual (FRA020), an unknown second-degree relative to the progenitor, is also buried in the northern part of the chamber. Another second-degree male relative to

the progenitor (FRA011) is found in the northern end and is represented by an isolated mandible. This male belongs to the first generation of the left-side kinship branch. We also identify a young woman (FRA102) unrelated to anyone else in the pedigree in the northern half of the tomb. This individual is an outlier in many ways: her body was given a special treatment because she was buried as a tight package of partially articulated bones (individual C in the excavation report). Moreover, the Sr value (0.717345) in her lower left M1 tooth suggests she spent her early years outside the Falbygden geology although her origin cannot be pinpointed more precisely. Lastly, she appears to have been affected by an ancestral form of the plague (discussed below).

There is a clear division in the placement of the later generations, because the right-side branch is found in the northern half and the left-side branch in the southern half of the chamber (Supplementary Fig. 6). Moreover, genetically female individuals are concentrated in the central part of the chamber whereas males are more evenly distributed (Supplementary Fig. 4).

In the central part of the passage grave the correlation between burial location and generation is not as clear as in the northern part. Nevertheless, there is a general trend of earlier generations being buried in the northern part of the central area whereas later generations are buried more centrally. Furthermore, we find that the four siblings (FRA032–FRA035) from generation five are buried together with their half-brother (FRA030) whereas their half-sister (FRA031) was buried elsewhere in the tomb. This indicates that the male line was more relevant in determining burial location. In contrast to the northern area, the central part holds many complete or partially articulated skeletons, as well as disarticulated bones.

Overall, it is possible to interpret the data from kinship and burial location as evidence for a time-dependent filling of the tomb, starting in the northern (and possibly in the southern) part of the chamber and slowly moving towards the centre. The difference in mortuary treatment can be viewed in different ways. The skulls of the first- and second-generation individuals could have been selected for special treatment and moved to the northern part, presumably from burials in the central area. In this case, postcranial remains of these individuals should be found within the chamber. Another possibility is that selected parts of these individuals were brought in from elsewhere, either as part of a programme of secondary burial or as foundation deposits from another burial.

Close familial relations are also recognized: we identify two nuclear families buried very closely together. Thus, biological kinship and sex seem to have been socially recognized and were used as important tools to categorize Neolithic people in Falbygden, and were determining factors for the placement and handling of dead members of society.

## Three Neolithic plague strains

We carefully examined all the sequenced data for known human pathogens (Methods and Supplementary Table 10). Strikingly, this screening showed that the most frequently found pathogen overall was the plague-causing bacterium *Y. pestis* (18 out of 108 sampled individuals, 17%). We identified five other pathogens that we consider authentic (Supplementary Table 10); of these, two were identified in more than one individual—*Yersinia enterocolitica*, the causative agent of yersiniosis (four of 108, 4%) and *Borrelia recurrentis*, which causes louse-borne relapsing fever (five of 108, 5%)—and indicate the presence of body lice in Falbygden. Interestingly, in one individual (FRA013) we identified a coinfection with *Y. pestis* and *Y. enterocolitica* at 3.9× and 1.8× coverage, respectively (Methods).

We found that plague frequency varied across sites, with the highest rate at Frälsegården (13 of 47, 28%, excluding Steppe-related individuals) and markedly lower rates at all other sites. Plague-positive individuals were found not only in the Falbygden area but as far south as Zealand, Denmark and by the Swedish west coast north of Gothenburg.

These findings indicate that the plague was widespread in southern Scandinavia 5,000 years ago.

Of the 18 plague-positive individuals, six were classified as tentative detections with coverage below 0.01× whereas seven could be classified as lower-coverage partial genomes (0.01–1.5×) and five were classified as higher-coverage partial genomes (over 1.5×; Supplementary Table 6). For the five higher-coverage genomes (11.5×, 10.5×, 6.4×, 4.5× and 1.8×) there were sufficient data to carry out single-nucleotide polymorphism (SNP) calling and reconstruct the full plague phylogeny, together with previously published ancient and modern genomes (Fig. 4, Extended Data Fig. 7 and Supplementary Fig. 5). In agreement with previous results[4], all newly sequenced genomes fell basal to the cluster of LNBA plague strains (Fig. 4). We found that the plague genomes from individuals FRA013 and FRA005 were identical to the previously published Gökhem2 genome from the same archaeological site and time period[4]. By contrast, the plague genome from FRA020 differed from the three identical genomes at three positions (Extended Data Fig. 8d)—we designate these as pre-LNBA strains C and B, respectively. Surprisingly, the plague genome from individual FRA102 is markedly different from strains C and B, from which it differs on 29–34 positions (Extended Data Fig. 8a). This genome falls basal to all known plague diversity except for strain RV2039, dated to 5300–5050 cal. BP (ref. 10). We designate this genome pre-LNBA strain A (Fig. 4).

Although no phylogenetic information can be extracted from the tentative plague detections (below 0.01×), lower-coverage genomes (0.01–1.5×) are of sufficient coverage to be placed onto the plague phylogeny (Extended Data Fig. 9). As expected, these placements confirm that all plague genomes sequenced in this study cluster together with the higher-coverage plague genomes from the Frälsegården cluster. In general these placements do not allow us to distinguish between strains A, B and C. There are, however, a few exceptions: the relatively high coverage of the plague genome from FRA106 (1.1×) produced a more accurate placement, which suggests that this strain might belong to strain C. Similarly, the higher number of unique SNPs in the more ancestral genomes from strain A and RV2039 means that lower-coverage genomes similar to ancestral strains are more easily placed onto the phylogeny. Accordingly, we find that individual FRA021 had a plague form similar to strain A whereas the individual from Denmark (AVL001) appears to have a plague form ancestral to strain A (Extended Data Fig. 9).

The distribution of plague-positive individuals in the pedigree presented in Fig. 3 does not readily support a swift and deadly plague epidemic, because plague is detected in all generations except two and six. However, when information on the different plague strains is taken into account, it becomes apparent that plague infections are stratified both chronologically and phylogenetically into two separate clusters. The most ancestral form of the plague (strain A) is also detected in the oldest individual from this study (FRA102), who was buried in the northern part of the chamber with individuals from generations one and two despite being unrelated to everyone in the pedigree. Based on placements of lower-coverage genomes, we also identify a plague form similar to strain A in FRA021, the progenitor of the right-side subfamily. This finding suggests spread of an early form of the disease in the first generation of the pedigree. The mortality rate of this form of the plague is unknown, but the pedigree clearly illustrates that both the family as a whole and the line of FRA021 survived the disease.

The second cluster of plague infections occurs in generations three to five and is caused by strains B and C. Strain C is found in generation four of the left-side family, in which all individuals appear to have been infected by the plague. Strain B on the other hand, is found in a single individual from the right-side subfamily, most probably from generation three. Given the genetic distance between strains B and C and their different distribution across the pedigree, it is possible that the two strains represent separate infection events: one in the right-side subfamily (strain B) and another in the left-side subfamily

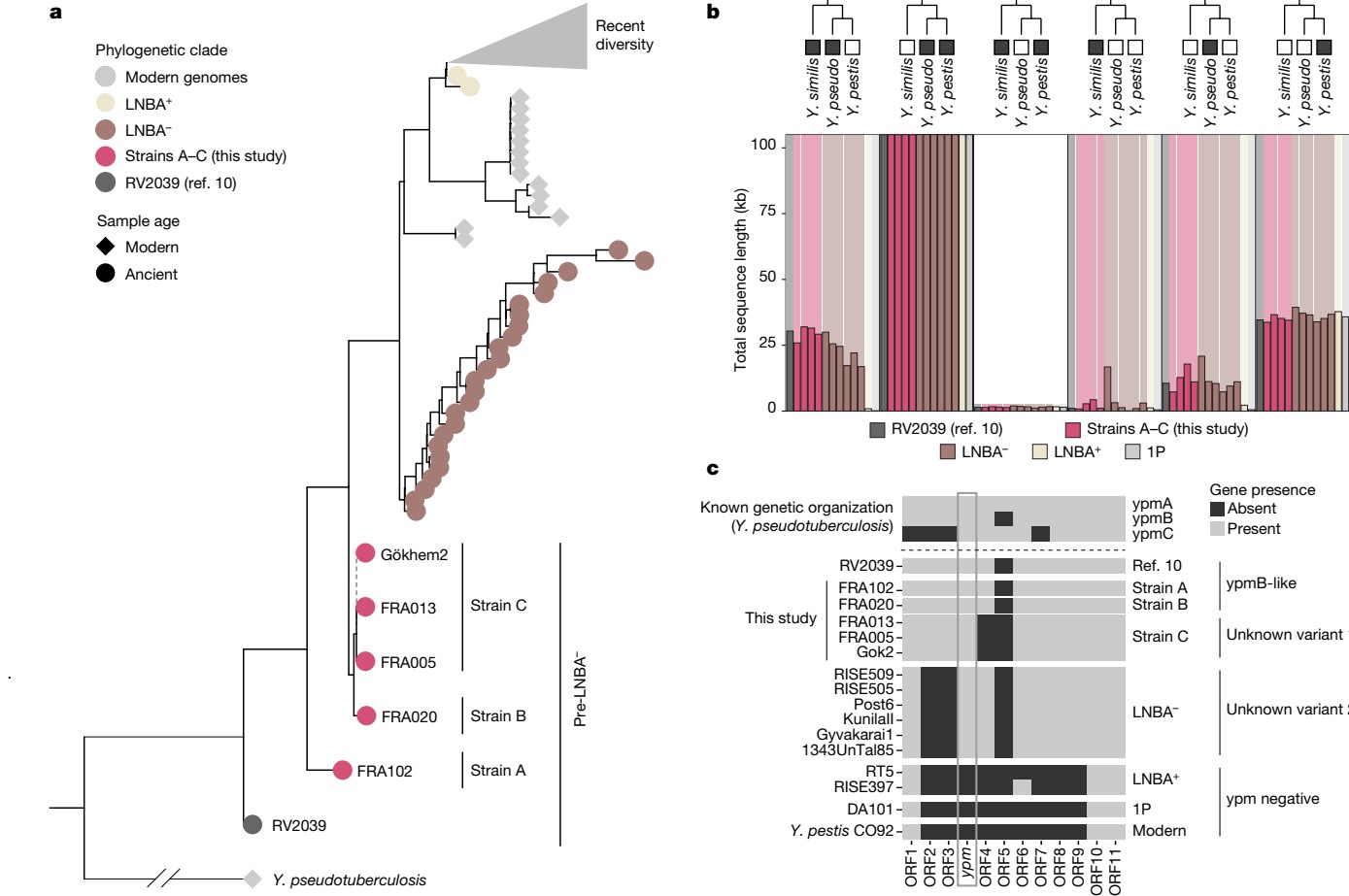

**Fig. 4 | Plague genomes. a**, Phylogenetic relationship between previously published plague strains and the data produced for this study. Each circle represents one plague genome, coloured by phylogenetic clade. For clarity, all plague strains more divergent than the LNBA clade are collapsed (grey triangle); Supplementary Fig. 5 shows the full tree. **b**, Barplot showing the amount of reference graph gene content covered by ancient *Y. pestis* samples. Reference graph nodes were stratified into groups of presence/absence pattern among modern *Y. pseudotuberculosis* complex species, indicated by cartoon phylogenies. The shaded background in the plot represents the total length of reference graph nodes for a given group; coloured bars represent the total length of nodes within that group present in a particular sample. Values on the *y* axis are capped at 100 kb to aid visualization; groups of ancient strains are indicated by bar colour. **c**, Presence/absence of genes in the unstable genomic region surrounding the superantigen *ypm* gene in *Y. pseudotuberculosis* IP 31758. A gene is defined as being present if the ratio of observed over expected breadth of coverage (given the depth of coverage across the genome) is over 10% for that gene–sample combination (Extended Data Fig. 10). 1P, first pandemic.

(strain C). Yet, even though both strains appear to be contemporaneous in Fig. 3, they could be temporally distinct because the error margins of the chronological modelling allow for considerable variation in the modelled dates (Extended Data Fig. 5). Unfortunately, it is not possible to further assess this hypothesis because we are unable to distinguish strains B and C in most of the lower-coverage genomes (Extended Data Fig. 9).

We can only speculate on the impact of the disease on the local population, but we note that all of the hallmarks expected in a swift and deadly epidemic are present for the spread of strain C in the left-hand subfamily: (1) the frequency of plague-positive individuals is exceptionally high, (2) the disease is restricted to the last two generations and (3) all higher-coverage genomes from this subfamily are identical. Taken together, these observations suggest that an outbreak of strain C could have led to the demise of the left-side subfamily, perhaps driven by increased pathogenicity due to recombination around the *ypm* locus in this strain (discussed below).

It is challenging to directly gauge the pathogenicity and route of transmission in ancient plague strains, but the high number of infected individuals analysed here suggests that the disease was able to spread within the population. To detect genetic variation that could hint at the pathogenicity of the Neolithic plague, we first analysed the coverage of classic plague virulence genes (Supplementary Fig. 6). Using this approach, we found the same virulence genes (*YpfΦ prophage* and *ymt*) to be absent in the Falbygden plague strains, as for both LNBA⁻ strains and pre-LNBA strain RV2039. Such analyses of gene content based on read mapping to a single-reference genome are unable to capture the full spectrum of genome variation among ancient *Y. pestis* strains. In particular, genomic variation absent in the reference genome but present in an ancient lineage cannot be detected using this approach. To overcome this limitation we investigated the gene content variation of ancient plague lineages by mapping their sequencing reads to a pangenome variation graph encompassing 82 complete assemblies of the *Y. pseudotuberculosis* complex (56 *Y. pestis*, 24 *Y. pseudotuberculosis* and one *Yersinia similis*). Partitioning the variant graph nodes into groups based on their presence pattern in the three species, we found that early divergent ancient plague lineages (pre-LNBA⁻, Frälsegården and LNBA⁻) harbour up to 50 kb gene content present in some or all *Y. pseudotuberculosis/Y. similis* strains that is absent in later strains (LNBA⁺, 1P, modern; Fig. 4b).

Intriguingly, among the genomic regions identified as containing variation absent in later plague strains was a chromosomal locus including the gene encoding for *Y. pseudotuberculosis*-derived mitogen (*ypm*), a superantigenic toxin found in some strains of *Y. pseudotuberculosis*[28].

All early diverging plague strains showed evidence for the presence of *ypm* whereas the entire locus is absent in all lineages diverging after and including the LNBA[+] type strain RT5. To distinguish the three known *ypm* loci (*ypmA*, *ypmB* and *ypmC*) we characterized gene presence/absence in the genetic environment surrounding this unstable locus (Fig. 4c). We identify a pattern of genetic organization resembling the ancestral locus, *ypmB*, in RV2039 and Falbygden strains A and B, and we identify two hitherto unknown combinations of genes in the *ypm* locus for strain C and later LNBA[−] strains. Although *Y. pseudotuberculosis* strains carrying the *ypmB* allele have been associated with low pathogenicity in humans[29], the virulence of the unknown *ypm* allele identified in strain C cannot be determined. It is, however, well established that certain *ypm* alleles have epidemic potential. The production of the superantigen toxin YPMa, for example, is crucial to the pathogenesis of Far East scarlet-like fever, a severe systemic infectious disease with symptoms including a scarlet-like skin rash and toxic shock syndrome[30]. Far East scarlet-like fever has been described as an 'epidemic manifestation' of *Y. pseudotuberculosis* infection in humans and, in 1959, an outbreak in Vladivostok, USSR, caused the hospitalization of 200 out of 300 patients[30,31].

Given the presence of ancestral *Y. pseudotuberculosis* variation in both pre-LNBA and LNBA[−] strains, it is possible that these plague forms followed a faecal–oral transmission route and showed attenuated pathogenicity. Nevertheless, the presence of the *ypm* gene and, in particular, the finding of a hitherto unknown combination of genes in the *ypm* locus for strain C, could suggest increased virulence. Hence, whereas infectivity, morbidity and mortality might have varied among these early plague strains, it is possible that a recombination event in the *ypm* locus rendered strain C virulent. In combination with other factors, and perhaps other diseases, strain C could thus have played a role in the Neolithic decline. Nonetheless, all individuals from pedigree 1 were buried in the Frälsegården tomb and someone must have survived to bury them. Furthermore, the demographic profile of the graves does not indicate catastrophic mortality[13], which, together with the high infection rates, could suggest a less severe or chronic disease manifestation. Lastly, it is well established that the later LNBA[−] clade of plague was prevalent and widespread across Eurasia for more than 2,000 years without markedly affecting Bronze Age population sizes[8].

## Perspectives and conclusions

By the end of the Neolithic at least three main plague lineages had evolved: the most ancestral RV2039 lineage[10], the Falbygden clade (strains A, B and C) and the lineage that would eventually evolve into the Bronze Age radiation of plague[8]. All three lacked the *ymt* gene, crucial for the bacterium's survival in the flea digestive tract. Consequently, flea-based transmission of LNBA[−] and pre-LNBA[−] plague strains is unlikely and the manifestation of Neolithic plague unlikely to resemble bubonic plague[32]. It has previously been hypothesized that the pre-LNBA RV2039 lineage and other early forms of the plague were less transmissible and that they represented sporadic zoonotic events[10]. By detecting plague in approximately one in every six sampled individuals, we conclusively show that plague infection was not a rare event in Neolithic Scandinavia. Given the high frequency of the disease, it is possible that it was spreading within the population following human–human, and potentially human–louse–human[33], transmission. However, it is worth mentioning that the plague rate of 17% reported here does not necessarily reflect the true prevalence of the disease. For example, the plague detection rate might not be representative of the population as a whole because it is a measure of disease frequency within the sampled population, which is restricted to well-preserved individuals buried in tombs. Furthermore, only a fraction of plague-positive cases is expected to carry detectable levels of DNA from *Y. pestis*. In ref. 34, a quantitative PCR screening of known

plague victims showed a detection rate of 5.7% in bones and 37% in teeth, suggesting that the true frequency of the Falbygden plague could be significantly higher than 17%.

The pathogenicity of the ancestral form of plague has been heavily debated in the literature[4,8–10,35] and, without an experimental set-up to directly gauge the virulence of the bacterium, we rely on circumstantial evidence. Our contribution to this discussion is threefold: (1) we track the presence of multiple plague strains across an extended pedigree, suggesting that plague infection was common in the community; (2) we show that the Falbygden plague strains carried a new and a known variant of the *ypm* locus from *Y. pseudotuberculosis*, suggesting that the disease manifestations, morbidity and mortality of these early plague strains might differ substantially; and (3) the Falbygden plague strains carried variable amounts of genetic material from *Y. pseudotuberculosis*, suggesting that the transmissibility and transmission route(s) might have varied somewhat. These results demonstrate that the Neolithic plague was prevalent and potentially lethal. Together with the fact that these plague cases are found in one of the last populations with Neolithic Farmer ancestry observed in Scandinavia, we believe that plague could have been a contributing factor to the Neolithic decline.

In addition to the plague, this study provides new data on familial relations in Middle Neolithic Sweden by reconstructing one large and three smaller pedigrees. We show that the social organization in Falbygden, as represented by the population sampled within these tombs, was patrilineal and patrilocal. This result corroborates findings from the Hazleton North long cairn[36]. However, we note significant divergences from the result reported in ref. 36: we identify four males with multiple reproductive partners, but we do not find any instances in which females produced offspring with different males. This is in contrast to the results from Hazleton North long cairn[36], in which females with multiple partners far outweigh males (five females versus one male). Furthermore, the subdivision into matrilineal subgroups is not visible at Frälsegården, in which subgroups tend to show a patrilineal structure. Another comparison can be made with the Neolithic cemetery at Gurgy, France[37], where a patrilineal and patrilocal structure was also found. In contrast to Falbygden, all couples at Gurgy were monogamous.

Lastly, these kinship data provide some of the first concrete examples of social practices commonly discussed in archaeology. We identify a set of three siblings, two males and one female, in which the female appears to have moved to a neighbouring group in which she established her own family. Furthermore, we also report a single instance of third-degree relatives producing offspring resulting in long stretches of ROHs in both of their sons.

Taken together, the data presented here provide a highly detailed and intimate snapshot of what life was like in Neolithic Falbygden, Sweden. The social structure was organized along male kinship lines, and females generally came from other kin groups. Because plague was infecting a significant proportion of the population, excess mortality associated with the disease could have undermined the long-term viability of society, leading to the eventual collapse of this form of Neolithic society.

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

## Methods

### Sampling, library building and sequencing

We sampled a total of 133 teeth, 38 petrous bones and three femur samples from the collections at the University of Gothenburg (Gothenburg, Sweden), Statens Historiska Museum (Stockholm, Sweden) and Laboratory of Biological Anthropology (University of Copenhagen, Denmark). Laboratory processing of samples was carried out at the clean laboratory facilities at Globe Institute, University of Copenhagen and at the Human Evolution ancient DNA laboratory at Uppsala University, following strict ancient DNA guidelines. Subsampling and mechanical cleaning of the samples were carried out by removing the surface layer using a drill bit, followed by sampling of the dense bone material of the outer cementum layer for teeth, or the cochlea for petrous bones. DNA extraction was carried out on approximately 100–150 mg of bone material as previously described (ref. 6, modified after ref. 38), followed by a uracil-DNA glycosylase treatment step to remove deaminated cytosines for the majority of samples (Supplementary Table 3). Subsequently, double-stranded DNA-sequencing libraries were built following ref. 39. DNA sequencing was performed on a NovaSeq 6000 instrument (Illumina) in paired-end configuration at the GeoGenetics Sequencing Core (Copenhagen) using the S4 200 cycle kit (v.1.5). Furthermore, for selected samples with promising results but low content of endogenous DNA, single-stranded libraries were built as described in ref. 40 followed by sequencing on the Illumina NovaSeq 6000 platform (Supplementary Table 3).

The Uppsala samples were decontaminated using ultraviolet irradiation, 0.5% sodium hypochlorite and deionized Millipore water. The outer surface layers were removed as described above, and 50–150 mg of either bone powder or bone pieces was collected using drill bits or diamond cutting wheels. DNA was extracted using silica-based protocols[41–43]. Double-stranded DNA libraries were prepared as described above, with some being uracil-DNA glycosylase treated (Supplementary Table 3)[44]. Some libraries were enriched using European MYbaits from MYcroarray following the manufacturer's instructions. Sequencing was performed at SciLifeLab SNP&Seq in Uppsala using Illumina HiSeq 2500, HiSeq X Ten or NovaSeq platforms and 125 or 150 base pair paired-end chemistry.

### Capture enrichment

For libraries in which shotgun-sequencing results suggested relatively high concentrations of plague DNA, in-solution capture enrichment of *Y. pestis* DNA was carried out. We used the myBaits kit from Arbor Sciences described in ref. 45 and carried out hybridization capture following the manufacturer's High Sensitivity protocol, but using only a single round of enrichment and 1.1–4.4 µl of baits per reaction. We pooled between one and 11 samples per capture reaction and reamplified the post-capture pools for 16 cycles (Supplementary Table 6).

### Basic bioinformatic processing and PCAs

Demultiplexed fastq files were trimmed for adapter sequences, and overlapping paired-end reads were collapsed using AdapterRemoval (v.2.0)[46]. Next, trimmed reads were mapped to the human reference genome (hg19) using bwa aln (v.0.7.17). The resulting alignments were converted to BAM files and merged, sorted and filtered using Samtools (v.1.12). The resulting BAM files were merged at sample level (Supplementary Table 2), and duplicates were identified using MarkDuplicates (v.2.27.4) from Picard Tools. Next, duplicate reads and reads with mapping quality less than 30 were removed from the sample-level BAM files using Samtools. Lastly, samples deriving from either the same individual or a monozygotic twin were merged based on both an initial run of ngsRelate and results from CrosscheckFingerprints (Picard; Supplementary Table 1). Because it is impossible, based on genetics, to distinguish whether two identical samples are derived from the same individual or from monozygotic twins, we merge all genetically identical samples. Nevertheless, in the case of FRA029, both samples belonging to this genetic individual are actually derived from the upper right first molar. This observation demonstrates that we have at least one pair of monozygotic twins in our dataset.

Average depth of coverage values was calculated using BEDTools genomecov (v.2.30.0), and mitochondrial haplogroups were assigned using mutserve (v.1.3.0) and haplogrep (v.2.1.25). Chromosome Y haplogroups were assigned using a tool developed in house and described in Supplementary Note 2.

For PCA we called pseudohaploid genotypes by random selection of one of the variants from mpileup (Samtools) that passed mapping and base quality filters (over 30). The resulting panel of pseudohaploid genotypes were merged with a panel of 1,430 previously published ancient shotgun genomes. We used smartpca from eigensoft to carry out PCA on 2,086,279 transversion-only SNPs with minor allele frequency of over 0.1%. All samples were projected onto the variation of the reference panel.

### IBD analysis

For identity-by-descent analysis we imputed the genotypes of 82 individuals with sufficient coverage (over 0.1×) using Glimpse[47] (v.1.1.1; https://github.com/odelaneau/GLIMPSE/tree/glimpse1), as described in ref. 23. Imputed genomes were then merged with a panel of imputed shotgun-sequenced genomes (Supplementary Table 7). Genomic segments shared via identity-by-descent were identified in all pairs of ancient samples using IBDseq[48] (v. r1206). Next we excluded the lowest-coverage individual in all pairs of close relatives (shared IBD segments over 1,500 centimorgans (cM), resulting in a total of 54 individuals from this study remaining for clustering (Supplementary Table 8). Genetic clustering was carried out based on IBD-sharing patterns in all pairs of samples, as described in ref. 23. For analysis of ROHs we filtered the imputed panel described above for genotype posterior values over 0.95 and identified segments of homozygosity-by-descent using IBDseq (v.r1206). These segment lengths were converted to cM using the HapMap Phase II genetic map (https://github.com/johnbowes/CRAFT-GP/tree/master/source_data/genetic_map_HapMapII_GRCh37) and summarized in six bins based on segment length.

### Kinship reconstruction

Initially we tested five different approaches to reconstruct kinship based on ancient DNA data (KING[49], ngsRelate[50,51], READ[52], KIN[53] and the relationship inference algorithm based on pairwise identity-by-state sharing presented in ref. 54). We found that all five approaches produced very similar results, but in some edge cases we observed that ngsRelate outperformed the four other approaches. Accordingly we decided to use ngsRelate for our analysis. We ran ngsRelate on a subset of the imputed panel described above, focusing on ancient Northern Europeans (816 individuals) and on 2,086,279 transversion-only SNPs with minor allele frequency over 0.1% in the panel. Next we classified all pairs of individuals as either zero-, first-, second- or third-degree relatives or as unrelated, based on pairwise relatedness ($R_{AB}$). We used the midpoint between the theoretical estimates of pairwise relatedness to distinguish between each degree of relatedness. For example, for zero-, first- and second-degree relatives the theoretical pairwise relatedness values are 1.00, 0.50 and 0.25, respectively. Accordingly we define first-degree relatives as pairs of individuals with $R_{AB}$ values between 0.75 (midpoint between 1.0 and 0.5) and 0.375 (midpoint between 0.50 and 0.25; Supplementary Table 9). Lastly, to distinguish relationship type for second-degree relatives we used results from the likelihood files produced by KIN. To run KIN we first ran KINgaroo on filtered BAM files, focusing on 2,086,279 transversion-only SNPs with minor allele frequency over 0.1%. KIN was then run using default settings.

For pedigree reconstruction we used a combination of the automated tool PRIMUS (pedigree reconstruction and identification of a maximum unrelated set[55]) and the manual triangulation approach described in ref. 36 (Supplementary Note 2).

## Pathogen screening

We carried out a screening for hits to all known human pathogens on all samples sequenced. The detailed description of the pathogen-screening pipeline can be found elsewhere[56]. In brief, we classified all reads using KrakenUniq[57] (v.0.5.8) on a custom database of microbial reference genomes, with a focus on human pathogens and common environmental microbes. For genera represented by more than 50 $k$-mer in a sample, traditional mapping was carried out. For a given genus/sample combination we used minimap2 (v.2.17-r941)[58] to map all reads classified to that genus to corresponding reference genomes of all species within that genus. Next we calculated standard mapping statistics for each sample/species combination, such as depth of coverage, ancient DNA damage, average nucleotide identity, mean edit distance and mean read length. We also calculated an estimate of coverage evenness (actual breadth of coverage/expected breadth of coverage given the depth of coverage) and ranked each species within a genus based on its number of unique $k$-mers identified by krakenUniq. We define a positive microbe detection as species represented by more than 50 reads, that are ranked as having the highest number of unique $k$-mers within their genus, have an average number of soft clippings under eight, have an average nucleotide identity of over 0.97 and have a coverage evenness score of over 0.8.

## Plague phylogenetic analysis

For all samples in which *Y. pestis* was detected in the pathogen screening, we mapped trimmed reads from all libraries of that sample to the reference plague genome (CO92; GCA_000009065.1) using bowtie2 (v.2.3.2) with the following parameters: -D 20 -R 3 -N 1 -L 20 -i S,1,0.50 --end-to-end --no-unal. Next we merged and sorted BAM files, and marked duplicates using MarkDuplicates from Picard Tools. Lastly, duplicate reads and reads of mapping quality under 30 were removed from the BAM files using Samtools.

The reference plague panel was generated by downloading raw read data from 400 modern plague strains, 30 ancient plague strains and data from the *Y. pseudotuberculosis* reference genome (GCF_000834295.1) at the European Nucleotide Archive. Raw fastq read files were trimmed with AdapterRemoval and mapped to the reference plague genome as described above, followed by filtering and calculation of the average depth of coverage using BEDTools genomecov. We called genotypes for plague samples with an average depth of coverage over ten for modern samples and over four for ancient samples, with the exception of Gok2, which was included despite having lower coverage. Genotype calling was carried out using Genome Analysis Toolkit[59] (v.4.1.9.0). First, sample-wise genomic variant call formats (GVCFs) were generated using HaplotypeCaller with a record at every single site (-ERC BP_RESOLUTION). Next, GVCFs were merged with CombineGVCFs and genotypes were called jointly with GenotypeGVCFs. Using VariantFiltration, genotypes were set to no-call if genotype quality was below 50 or if depth was over 1,000 or under four or ten, for ancient and modern samples, respectively. Furthermore, genotypes were set to no-call if genotype was supported by less than 90% of the reads at a given position. Lastly we filtered variants using VariantFiltration based on variant confidence/quality by depth, Phred-scaled *P* value, root mean squared mapping quality, $z$-score of Alt versus Ref read mapping qualities, $z$-score of Alt versus Ref read position bias, as follows: --filter-expression "QD < 2.0||FS > 60.0||MQ < 40.0 || MQRankSum < −12.5 || ReadPosRankSum < −8.0".

To generate the phylogenetic tree we converted the vcf file to a multifasta using bcftools consensus, replacing missing data with N (flag: -M N) and filtering out regions with high proportions of reads of mapping quality zero (flag: -m [bedfile]; Supplementary Note 2). We used RAxML-NG (v.0.9.0)[60] to generate the phylogenetic tree, applying the GTR + G model and using the *Y. pseudotuberculosis* reference genome (GCF_000834295.1) as an outgroup. Bootstrap support values were calculated with raxml using both the transfer bootstrap expectation and Felsenstein's bootstrap proportions metric. Both sets of bootstrap values produced similar support for the main nodes of the phylogeny. Lastly we plotted the best tree generated by RAxML-NG using the ggtree[61,62] (v.2.0.4) package in R (v.4.2.2).

For lower-coverage plague genomes (0.01–1.0×) for which traditional genotype calling was not feasible, we used EPA-ng (v.0.3.8)[63] to place each sample onto the phylogeny described above. For this approach we used major allele genotypes called by selecting the most common allele passing filters (mapping quality (MQ) over 30, base quality (BQ) over 30) from Samtools mpileup, and converted the vcf to fasta format as described above. Next we subset the fasta to the same SNPs as in the reference panel using BEDTools getfasta. We ran the evolutionary placement algorithm on the resulting fasta file, with default parameters supplying the best tree and the associated model from RAxML-NG as reference. Placements were plotted using the ggtree[61,62] (v.2.0.4) package in R (v.4.2.2).

## Pangenome analysis

We used a pangenome variation graph framework to investigate variation in pangenome content among ancient *Y. pestis* lineages. We selected 81 complete reference assemblies of the *Y. pseudotuberculosis* complex (56 *Y. pestis*, 24 *Y. pseudotuberculosis* and one *Y. similis*) from NCBI (assembly level 'chromosome' or 'complete') and built a pangenome variation graph using *pggb*[64] with the following parameters:
- segment length: 5 kb (-s 5,000)
- pairwise identity: 0.9 (-p 0.9)
- minimum match length: 47 (-k 47)
- target sequence length for POA: 700, 900, 1,110 (-G 700,900,1,100)
- score parameters for POA: asm5 (-P asm5)

We built graphs separately for the chromosome and plasmids and merged them into a single-reference graph using the 'vg combine' function of the vg toolkit[65]. The resulting graph was then indexed for short-read mapping with the Giraffe[66] mapper, with default parameters except $k$-mer length (-k 15) and window length (-w 5) for the minimizer. Shotgun-sequencing reads for each sample were then mapped to this index using 'vg giraffe'. We filtered the resulting graph alignment files (GAM format) for mapping quality of 30 or above (-q 30) and normalized alignment score of 0.9 or above (-r 0.9) and removed reads with tandem repeat motifs at the read ends (-E 3). To obtain coverage statistics we then computed read support for each node in the reference graph using 'vg pack' with the '-d' option. We considered a reference graph node covered in an ancient sample if the proportion of the node length covered by sequencing reads was 0.2 or above. To classify nodes into groups based on their presence or absence among the three *Y. pseudotuberculosis* complex species in the reference graph, we obtained a path coverage haplotype matrix for the reference contigs across all nodes in the graph using 'odgi paths'[67] with the '-H' option. Each node was then labelled according to the combination of *Y. pseudotuberculosis* complex species with coverage observed. For each ancient sample we aggregated coverage across nodes and classes to obtain the total amount of pangenome sequence contained in each class.

To identify the gene content of the *Y. pseudotuberculosis* complex pangenome in ancient *Y. pestis* lineages absent from modern *Y. pestis* strains, we extracted all nodes of the reference graph from classes not containing modern *Y. pestis* (that is, shared by *Y. similis* and *Y. pseudotuberculosis* or observed only within one of the two) in which at least one ancient sample had coverage. The reference graph start/end coordinates for those nodes were then translated into reference path positions using 'odgi position', and nodes were collapsed into regions if their translated position was within 200 base pairs of each other on the same reference contig. From this list of initial candidate regions we selected all of length greater than 500 base pairs, extended them by 10 kb on each side and merged overlapping regions to yield a

final list of genomic regions harbouring putative pangenome content retained in ancient *Y. pestis*. To further investigate coverage across genes in those regions we split the reference graph into region-specific graphs using 'vg chunk' and produced single-reference alignments (BAM format) for each ancient sample and region using 'vg surject'. The resulting BAM files were filtered by removal of duplicates (picard) and retaining only reads with mapping quality over 30, minimum number of aligned bases over 30 and, at most, five soft-clipped bases. Final per-gene coverage statistics were obtained from the filtered BAM files using 'bedtools coverage'[68].

### Radiocarbon dating and modelling
For Frälsegården a total of 91 radiocarbon dates are available, of which 46 are produced within the present project. The new datings targeted individuals in the pedigree, those showing plague and those with Steppe ancestry or HG (hunter-gatherer) admixture. Some samples were also redatings of individuals with anomalous previous dates. Seven new datings were made on samples from the Landbogården passage grave and six from the Nästegården site (Supplementary Table 4).

New datings were performed at the Keck carbon cycle AMS facility, University of California, Irving. The samples were decalcified in 1 N HCl, gelatinized at 60 °C and pH 2.0 and ultrafiltered to select a high molecular weight fraction (over 30 kDa). $\delta^{13}$C and $\delta^{15}$N values were measured to a precision level of below 0.1‰ and below 0.2‰, respectively, on aliquots of ultrafiltered collagen, using a Fisons NA1500NC elemental analyser/Finnigan Delta Plus isotope ratio mass spectrometer. Further details can be found at https://sites.ps.uci.edu/kccams/education/protocols/.

All samples were dated successfully, and showed C:N ratios well within the accepted range for well-preserved collagen, 2.9–3.6 (ref. 69). $\delta^{13}$C and $\delta^{15}$N values indicate a generally terrestrial diet with little or no contribution from marine or freshwater protein, suggesting that reservoir effects are negligible (Supplementary Table 4).

Datings were calibrated in Oxcal 4.4.4 using the Intcal20 calibration curve[70,71]. Because the time period of interest shows a large plateau in the calibration curve, the calibrations result in broad ranges, usually approximately 5250–4850 cal. BP ($2\sigma$). Chronological modelling of individuals within the pedigrees was made, allowing considerably higher precision. The model makes the simple assumption that genetically identified generations can be regarded as sequential phases. Tooth dates were offset by 20 ± 5 years to make them comparable to those from bone, based on formation time. The assumption of a generational interval of 29 ± 19 years, as suggested in ref. 27, was tested but did not improve the result. Four dates were identified as outliers by both low-agreement indices and outlier analysis (HJE012, FRA023, FRA002 and FRA009). These were marked as outliers in the Oxcal codes but not modelled. Detailed results and Oxcal code can be found in Supplementary Note 3 and Supplementary Tables 11–13.

### Strontium isotope analysis
Ten new Sr isotope values were produced within the present project, all on samples from Frälsegården (samples F11500–F11510; Supplementary Table 5). In addition, 24 samples from Frälsegården, Firse sten, Hunnebostrand and Rössberga were previously published[11,72,73]. A further 34 samples were previously analysed and are reported here for the first time (Supplementary Table 5). The measurements are bulk values taken on small pieces of tooth enamel, representing an average over the time of enamel formation. All measurements of $^{87}$Sr:$^{86}$Sr ratios were conducted at the Geochronology and Isotope Geochemistry Laboratory, Department of Geological Sciences, University of North Carolina, Chapel Hill. Samples were dissolved in nitric acid and the strontium fraction purified by ion selective chromatography (Eichrom Sr resin) before analysis by thermal ionization mass spectrometry (TIMS) on a VG Sector 54 mass spectrometer run in dynamic mode. Internal precision in the laboratory is consistently around 0.0007% standard

error (or $1\sigma$ = 0.00006 in the ratio of a particular sample). Long-term, repeated measurements of SRM-987 are around 0.710260—an acceptable difference from the recognized value of 0.710250—and raw sample values from individual runs are standardized to the recognized value of SRM-987. Baseline values from Falbygden and surrounding areas in western Sweden were published in ref. 74.

### Reporting summary
Further information on research design is available in the Nature Portfolio Reporting Summary linked to this article.

## Data availability
Raw fastq files from this study have been deposited in the European Nucleotide Archive under accession number PRJEB76142; see Supplementary Table 15 for an overview of European Nucleotide Archive accession numbers and their corresponding sample and library identifiers. Furthermore, human (hg19) and *Y. pestis* (GCF_000009065) BAM alignment files and genotype data in vcf file format are available from https://doi.org/10.17894/ucph.d2098f40-1263-4e28-afed-a0d50ff25c0c. The map of Scandinavia in Fig. 1 was created with the rworldmap package in R using data from the Natural Earth dataset, and the insert map of Falbygden was created with data from OpenStreetMap. The map of Scandinavia in Fig. 2b was created with the R package 'maps' using data from the Natural Earth dataset.

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

**Acknowledgements** This study was supported by the Lundbeck Foundation (no. R322-2019-2610 to F.V.S.), L. J. Hägglunds Stiftelse (to F.V.S) and Riksbankens Jubileumsfond (no. M 21-0018). Furthermore, the research at the University of Copenhagen was carried out under the Lundbeck Foundation GeoGenetics Centre, which is supported by the the Lundbeck Foundation (nos. R302-2018-1799 2155 and R155-2013-16338), the Novo Nordisk Foundation (no. NNF18SA0035006), the Wellcome Trust (no. UNS69906), the Carlsberg Foundation (no. CF18-0024), the Danish National Research Foundation (no. 44113220) and the University of Copenhagen (KU2016 programme). The research at Uppsala University was supported by The Swedish Research Council (nos. 2013-1905 to M.J., A.G. and J. Storå, and 2017-02503 to H. Malmström), Riksbankens Jubileumsfond (no. M13-0904:1 to M.J., A.G. and J. Storå) and the Knut and Alice Wallenberg foundation (Atlas of Ancient Human Genomes in Sweden project to M.J., A.G. and J. Storå). Computations and data handling were enabled by resources provided by the Swedish National Infrastructure for Computing at the Uppsala Multidisciplinary Center for Advanced Computational Science, partially funded by the Swedish Research Council through grant agreement no. 2018-05973. Sequencing was performed by The National Genomics Infrastructure Uppsala and at GeoGenetics Sequencing Core, Copenhagen. Investigations at Frälsegården were funded by Länsstyrelsen Västra Götaland and Riksbankens Jubileumsfond, grant no. P2006-0011:1-E.

**Author contributions** F.V.S., K.-G.S and M.S. led the study. A.G., E.W., M.J., H. Malmström and M.S. supervised research. K.-G.S., M.B., A.F., M.E.A., T. Ahlström and J. Storå collected samples. F.V.S., E.M.S., J. Staring, M.F., T.P., H. McColl, C.G., T.R.B., L.G. and B.V.-R. conducted laboratory work. F.V.S., K.-G.S., J.K. and M.S. analysed data and produced the figures. K.-G.S., A.F., T. Axelsson, T. Ahlström, J. Storå, K.K. and H. Malmström provided interpretation of the archaeological context. K.-G.S., T.D.P. and T. Ahlström collected and curated bioarchaeological data. T.D.P. analysed strontium samples. A.K.N.I. provided interpretation of pathogen host interactions. F.V.S., K.-G.S. and M.S. wrote the manuscript with input from all co-authors.

**Competing interests** The authors declare no competing interests.

**Additional information**
**Correspondence and requests for materials** should be addressed to Frederik Valeur Seersholm or Martin Sikora.

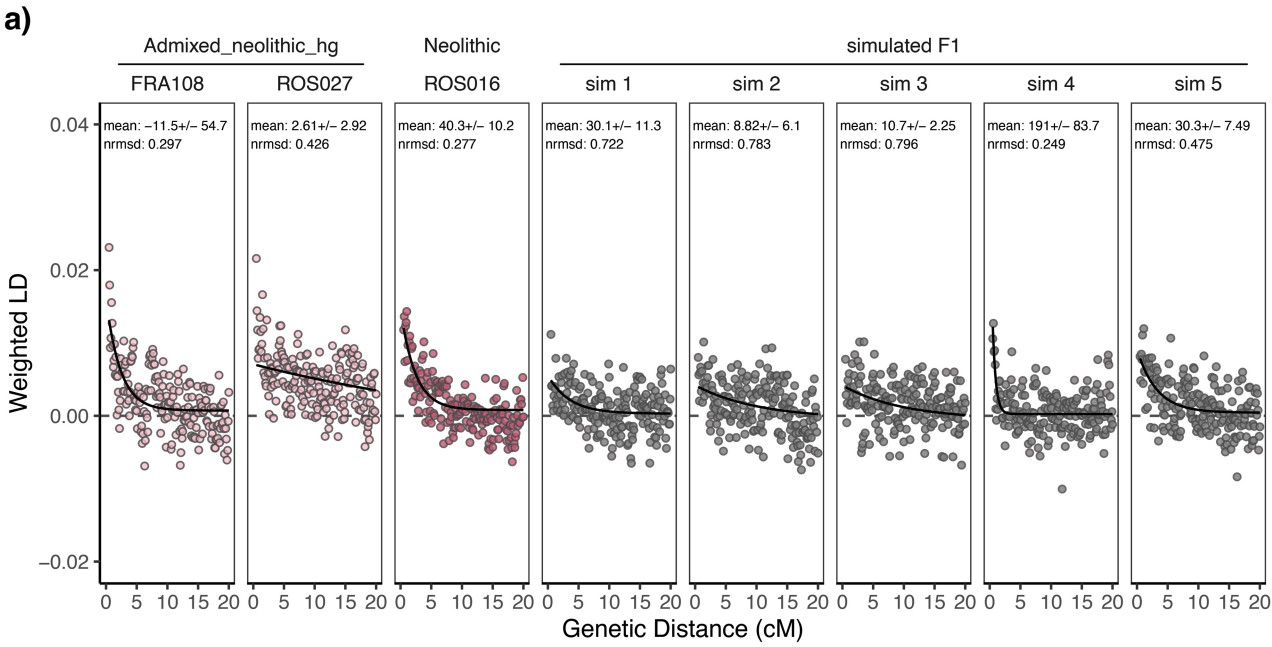

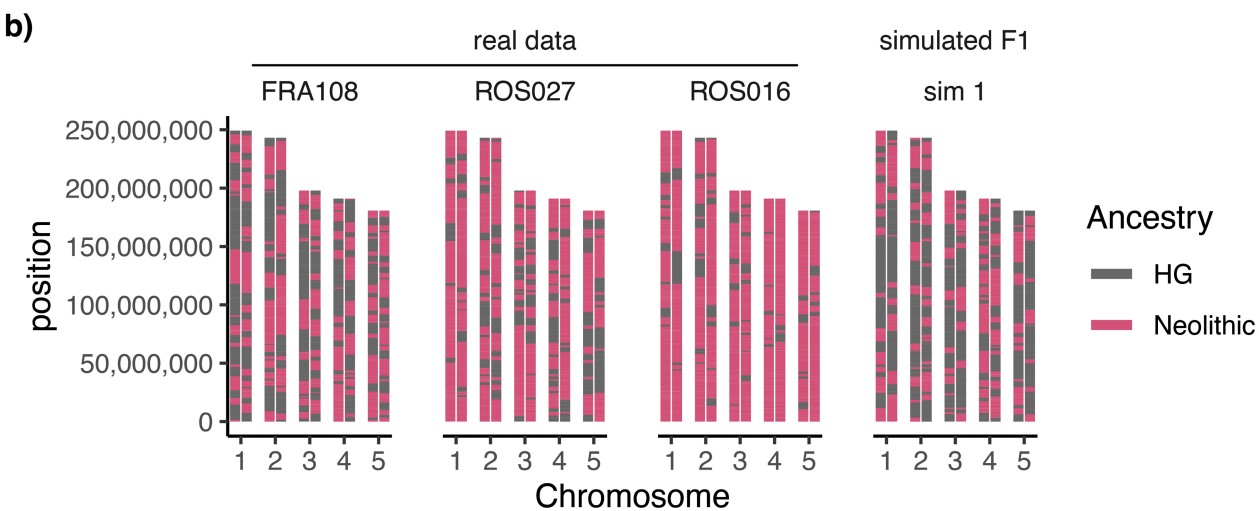

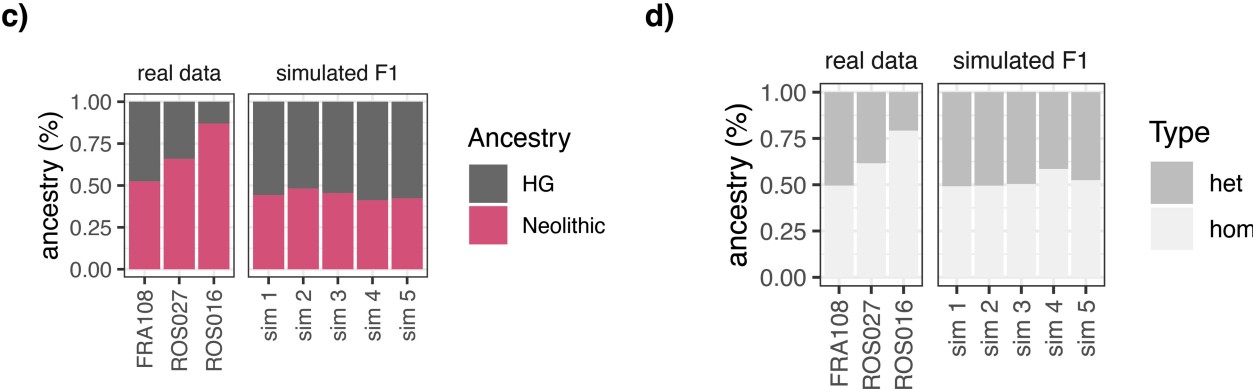

**Extended Data Fig. 1 | An investigation of the two admixed individuals.**
**a)** Estimates of admixture timing in admixed and simulated F1-individuals. Admixture time estimates were calculated with DATES using Swedish Neolithic farmer related individuals and PWC individuals as source groups (see methods). One un-admixed individual (ROS016) was included as a control. **b)** Local ancestry inference of admixed individuals across chromosomes one to five. RFmix estimates of hunter-gatherer (PWC) and Neolithic Swedish ancestries for admixed individuals (FRA108 and ROS027), a Neolithic individual with no

evidence of recent admixture (ROS016) and simulated F1 individuals with equal parts HG and Neolithic DNA. To simplify the plot, only chromosomes one to five and one simulated individual is shown. **c)** Total fraction of each ancestry type across the genome of each individual. **d)** Total fraction of genotype class across the genome of each individual, where 'het' indicates regions where one allele is of HG ancestry while the other is of Neolithic ancestry, while 'hom' indicates that both alleles are of the same ancestry.

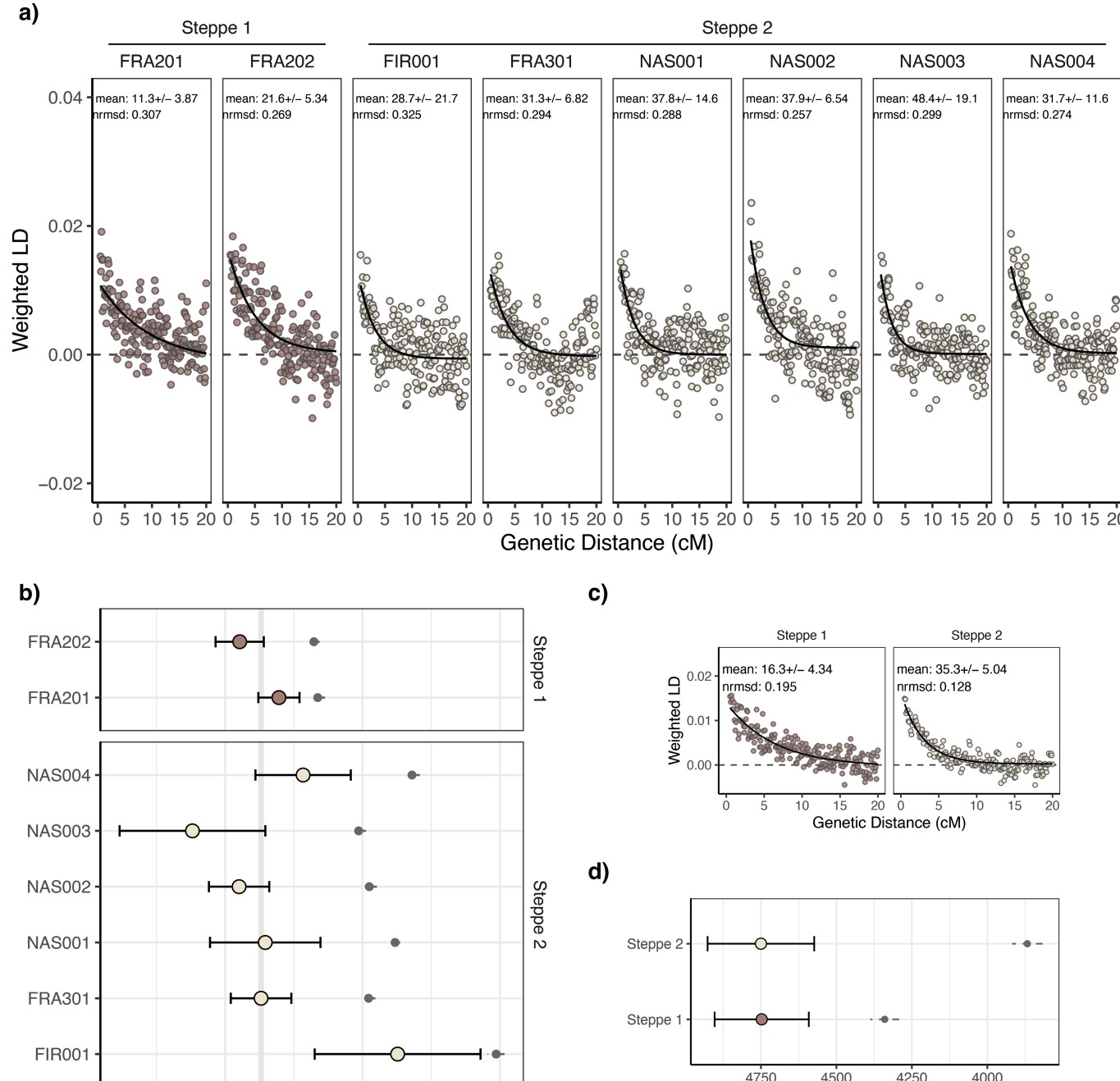

**Extended Data Fig. 2 | Estimates of admixture timing in Steppe related Individuals.** Admixture time estimates were calculated with DATES using Swedish Neolithic farmer related individuals and Yamnaya individuals as source groups (see methods). **a)** Sample-wise weighted linkage disequilibrium measures against genetic distance in centimorgan. In the top right corner of each plot, mean relative admixture time and standard error is shown in generations ago. **b)** Estimates of absolute admixture time using a generation time of 25 years and the calibrated ages of each sample (Supplementary Table 4; shown in grey). Highlighted in grey is the time period between 4,718 and 4,758 cal. BP where confidence intervals of all samples except FIR001 overlap. **c)** Weighted linkage disequilibrium measures against genetic distance estimated for the two Steppe related groups. **d)** Estimates of absolute admixture time for the two Steppe related groups using the average age of samples in each group (shown in grey) as the group age.

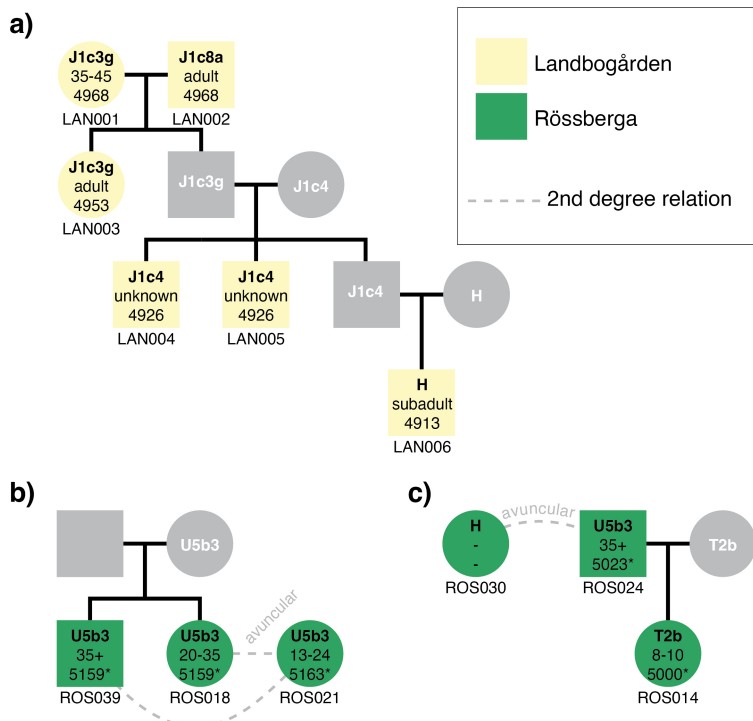

**Extended Data Fig. 3 | Pedigrees from Landbogården and Rössberga.**
**a)** Pedigree 2 from Landbogården. **b)** Pedigree 3 from Rössberga. **c)** Pedigree 4 from Rössberga. Squares and circles represent males and females, respectively, and information on mitochondrial haplogroup, osteological age estimate, and radiocarbon date for each individual is indicated inside each shape. Yellow and green colours indicate the sites Landbogården and Rössberga, respectively, while grey colour represents unsampled individuals. Solid black lines indicate first degree relations. Dashed grey lines signify unknown 2<sup>nd</sup> degree relationships. *For individuals from Rössberga radiocarbon dates could not be modelled, accordingly dates for these individuals are just reported as the median of the calibrated date.

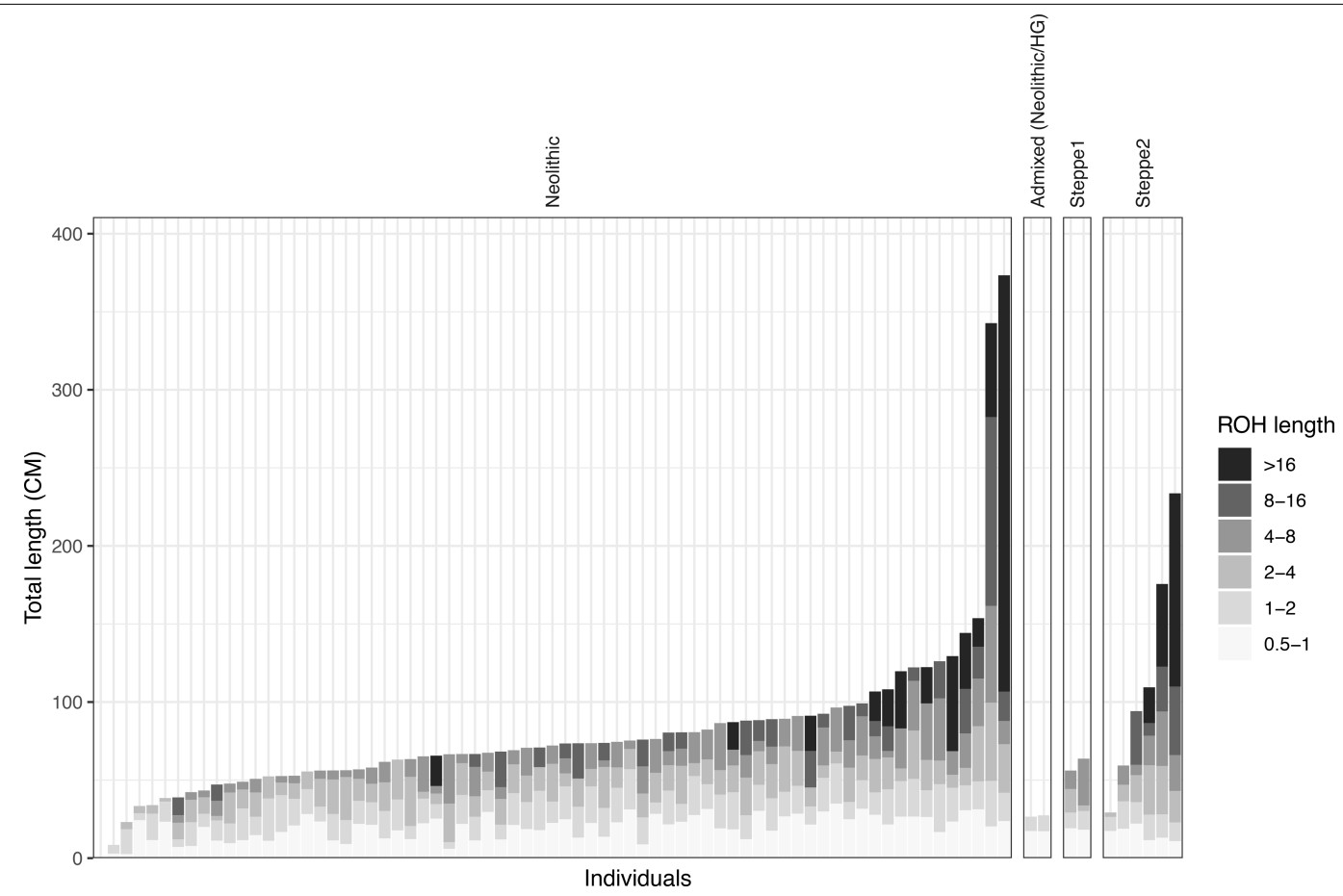

**Extended Data Fig. 4 | Runs of Homozygosity.** Total length of long Runs of Homozygosity (>0.5 cM) for each individual coloured by ROH size. Plot is split into four panels representing each ancestry group (see Fig. 2).

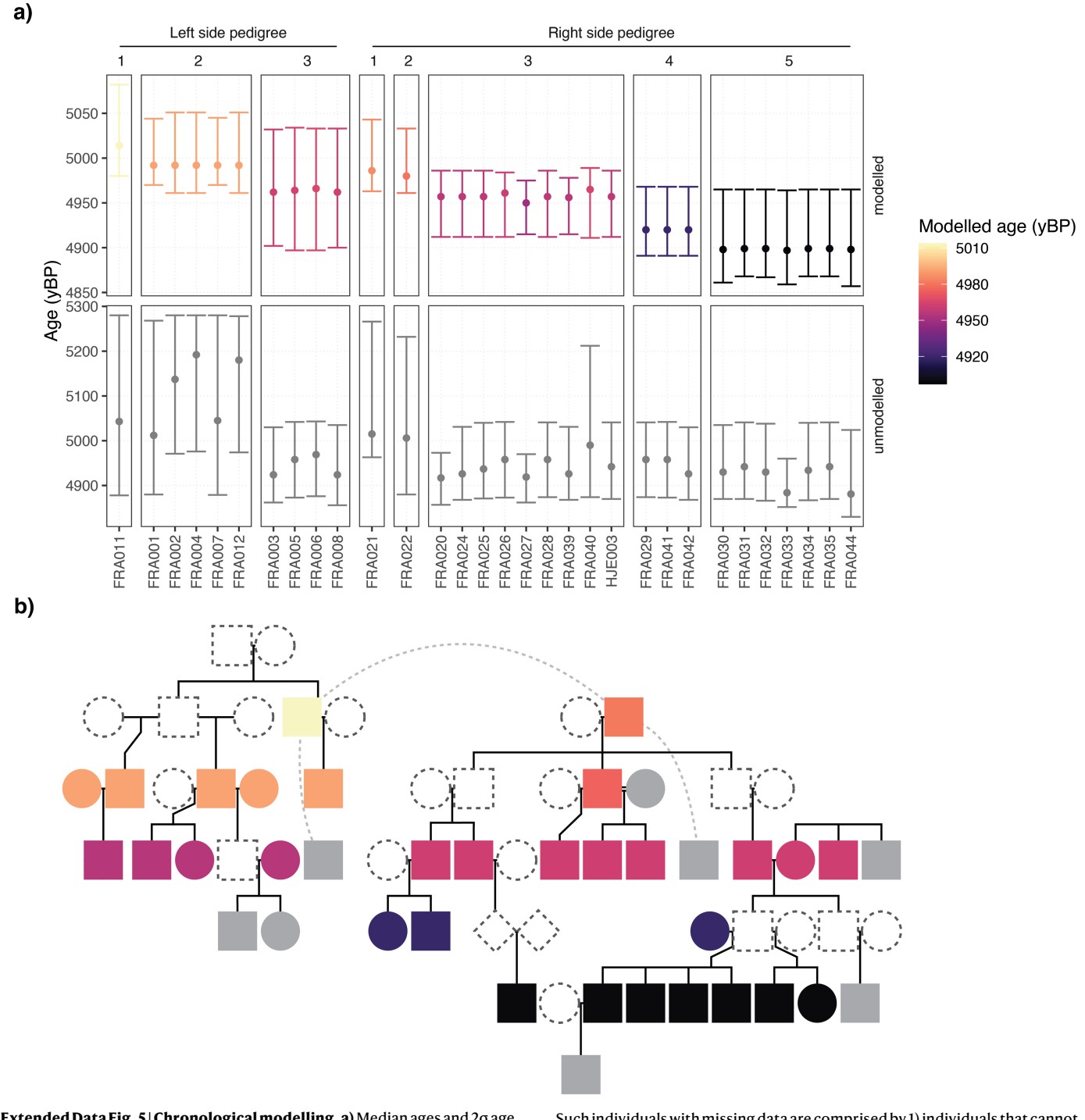

**Extended Data Fig. 5 | Chronological modelling. a)** Median ages and 2σ age ranges for modelled and unmodelled dates. Plot is stratified by pedigree branch (left side and right side, respectively) and generation in pedigree. **b)** Pedigree 1 from Frälsegården coloured by modelled age, following the color scheme of a. Grey color indicates individuals were no modelled age is available.

Such individuals with missing data are comprised by 1) individuals that cannot be placed in the pedigree (stippled lines), 2) individuals with dates classified as outliers (Supplementary Table 4), or 3) individuals that are linked to the pedigree through unknown second degree relationships.

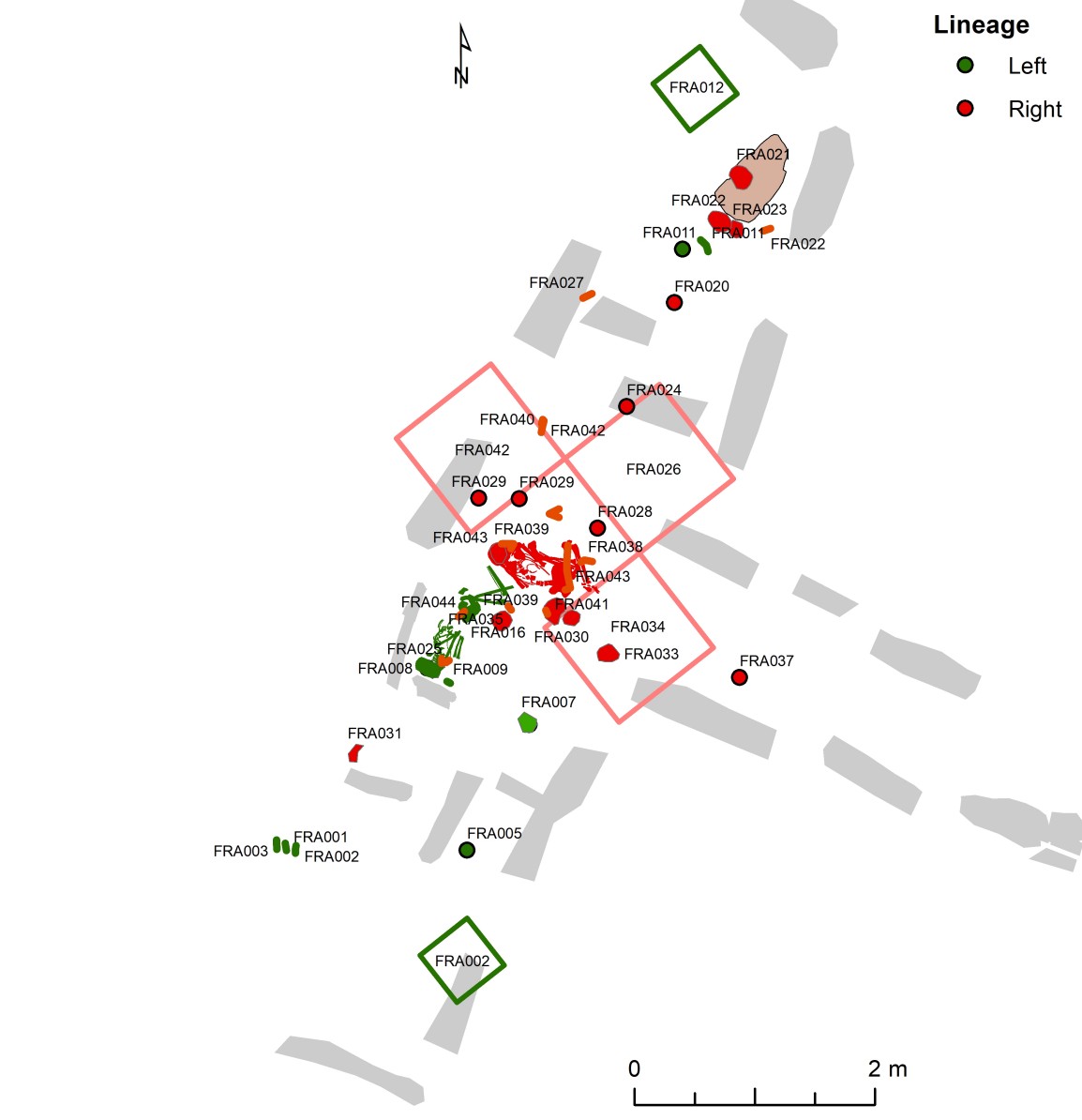

**Extended Data Fig. 6 | Burial locations within the Frälsegården passage grave coloured by lineage.** Green colour represents the left side pedigree from Fig. 3, while red colour represents the right side pedigree. For samples without exact coordinates, the quadrant from which the sample was excavated is plotted instead.

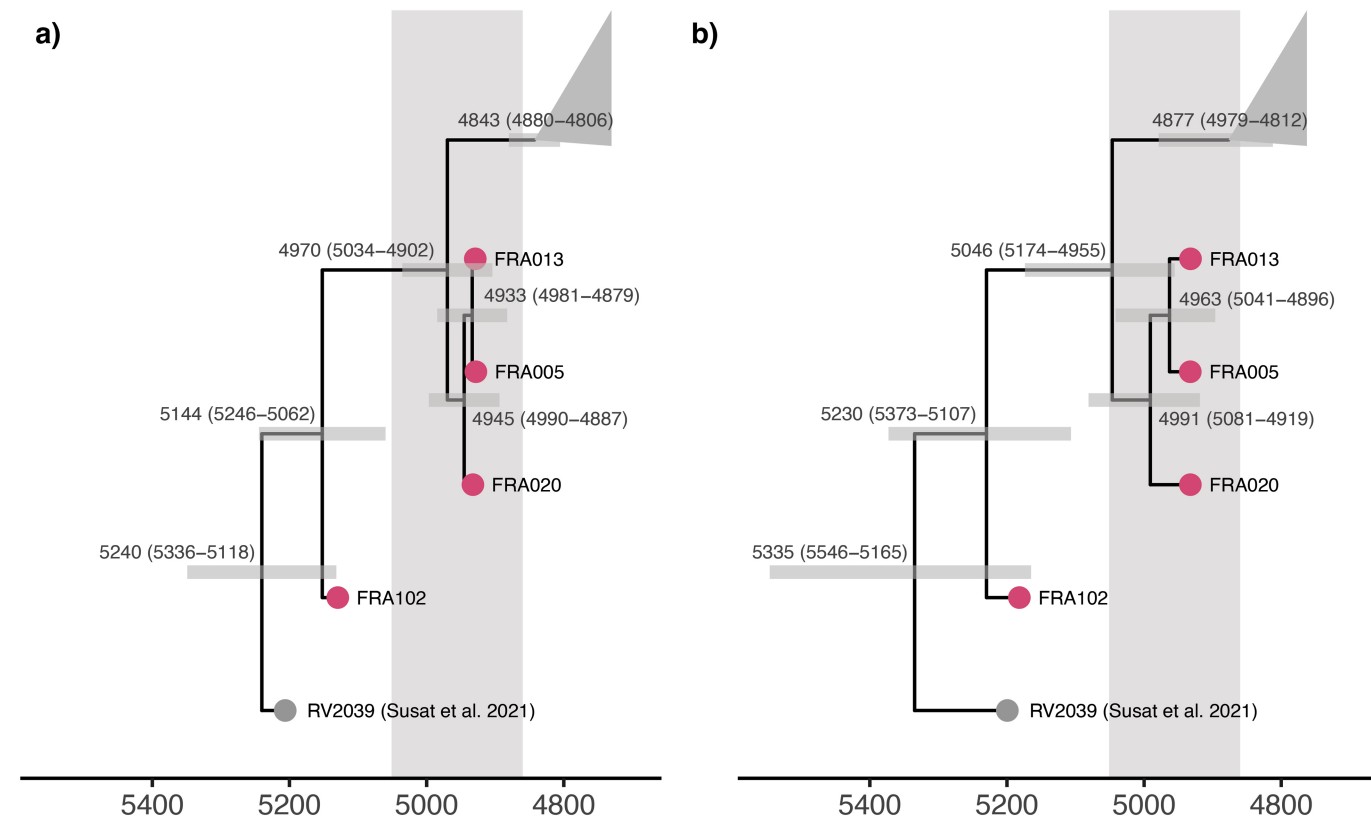

**Extended Data Fig. 7 | Molecular dating of ancient plague strains.** Dating analysis were carried out on preLNBA strains (grey and pink color) and LNBA-strains (collapsed in grey triangle). The modelled chronological span of the pedigree from Frälsegården is highlighted in grey. **a)** BEAST analysis using the Coalescent Bayesian Skyline demographic model and assuming an Optimised Relaxed Clock and the GTR substitution model with four gamma categories and empirical frequencies. **b)** BactDating analysis using the 'relaxedgamma' model.

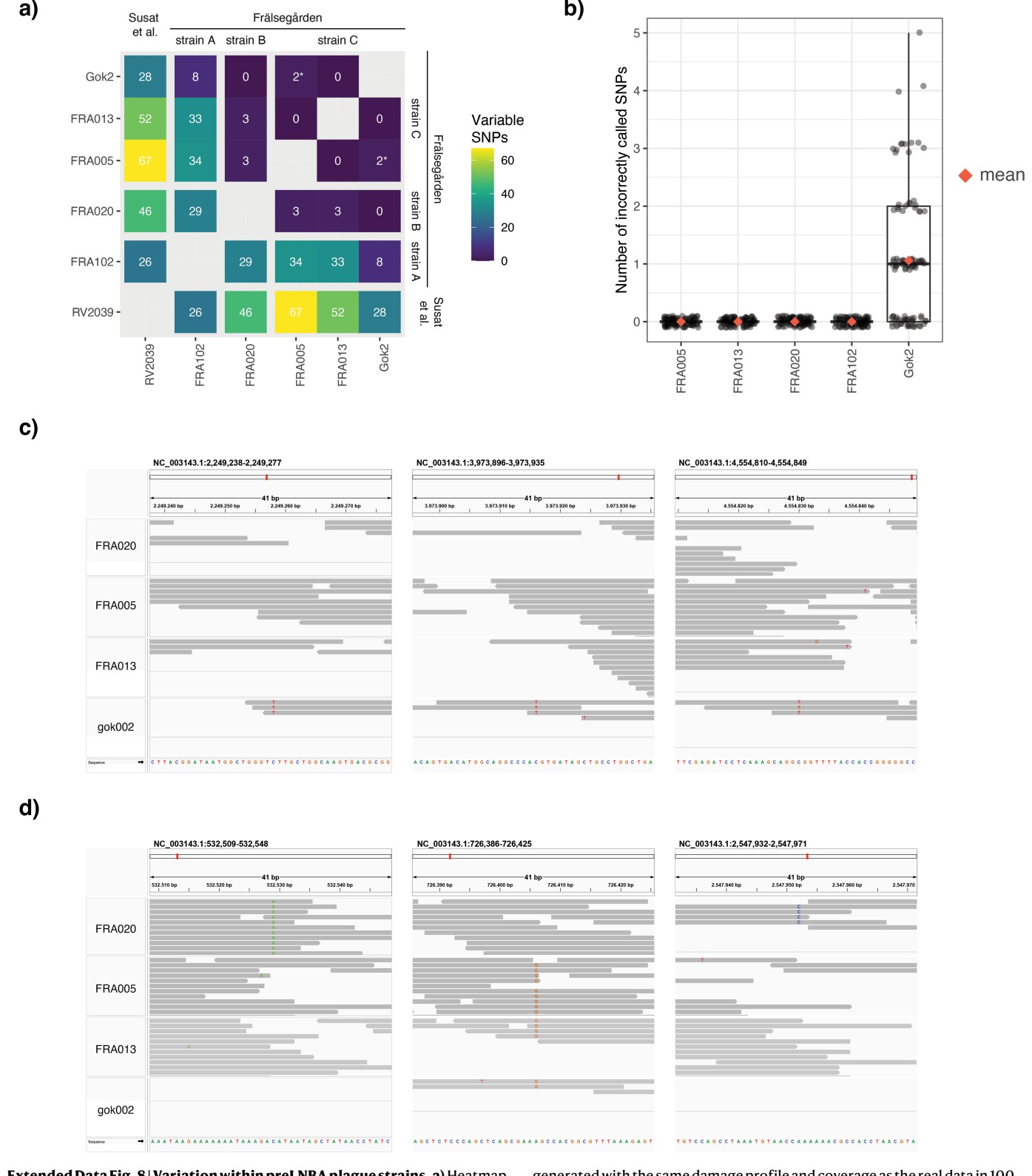

**Extended Data Fig. 8 | Variation within preLNBA plague strains. a)** Heatmap of pairwise SNP diversity of preLNBA plague strains. Plague genomes identified in this study (from Frälsegården) are divided into strains A, B, and C (see Fig. 4). *Unreliable SNPs (discussed in Supplementary Note 4). **b)** Number of incorrectly called SNPs from simulated data. For each sample, simulated plague data was generated with the same damage profile and coverage as the real data in 100 replicates. After variant calling the total number of incorrectly called SNPs were counted for each replicate. **c)** SNPs unique to gok002. **d)** SNPs unique to FRA020.

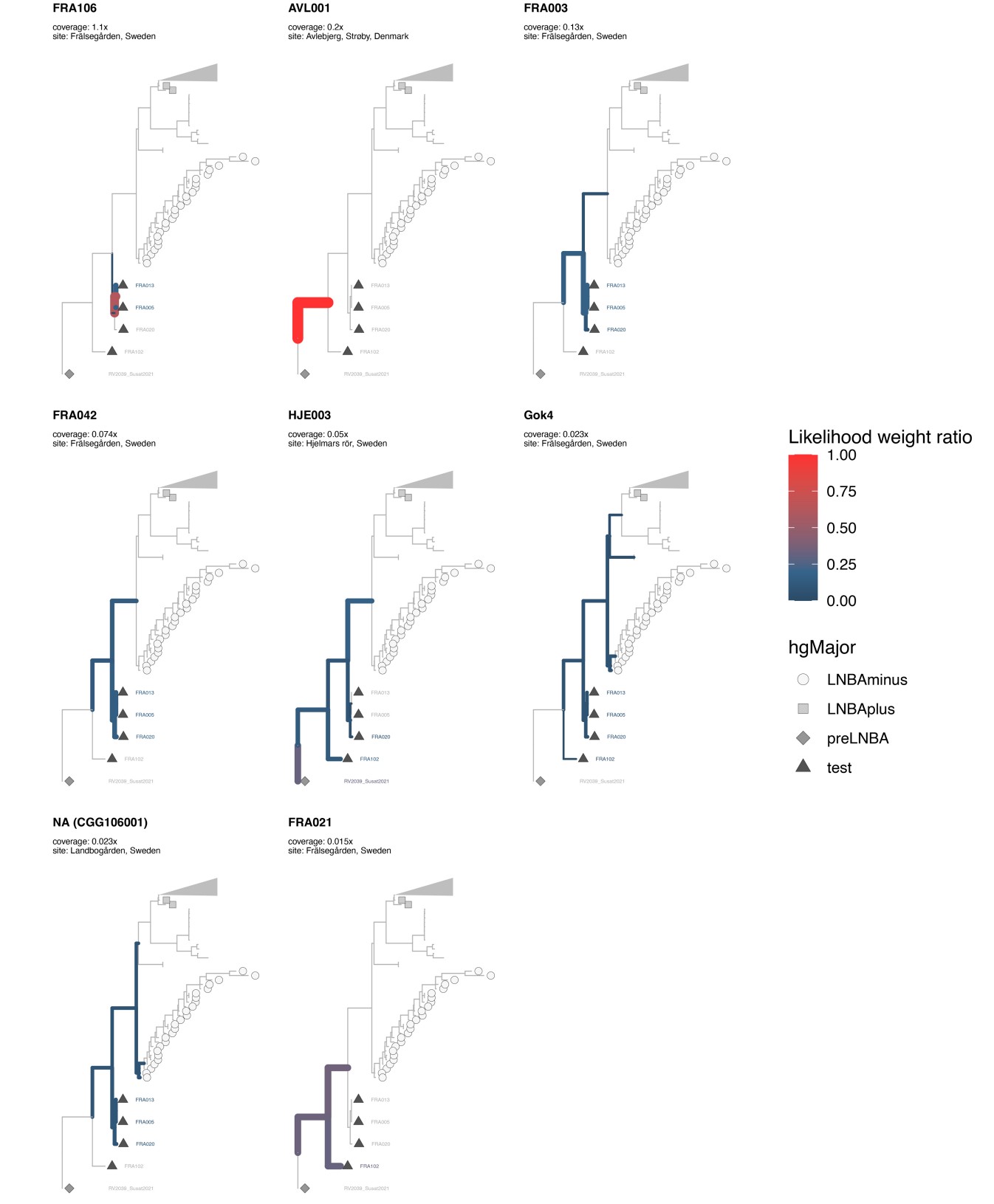

**Extended Data Fig. 9 | Phylogenetic placements of lower-coverage partial genomes (0.01-1x).** Branch colouring and stroke indicates likelihood of placement on a given branch.

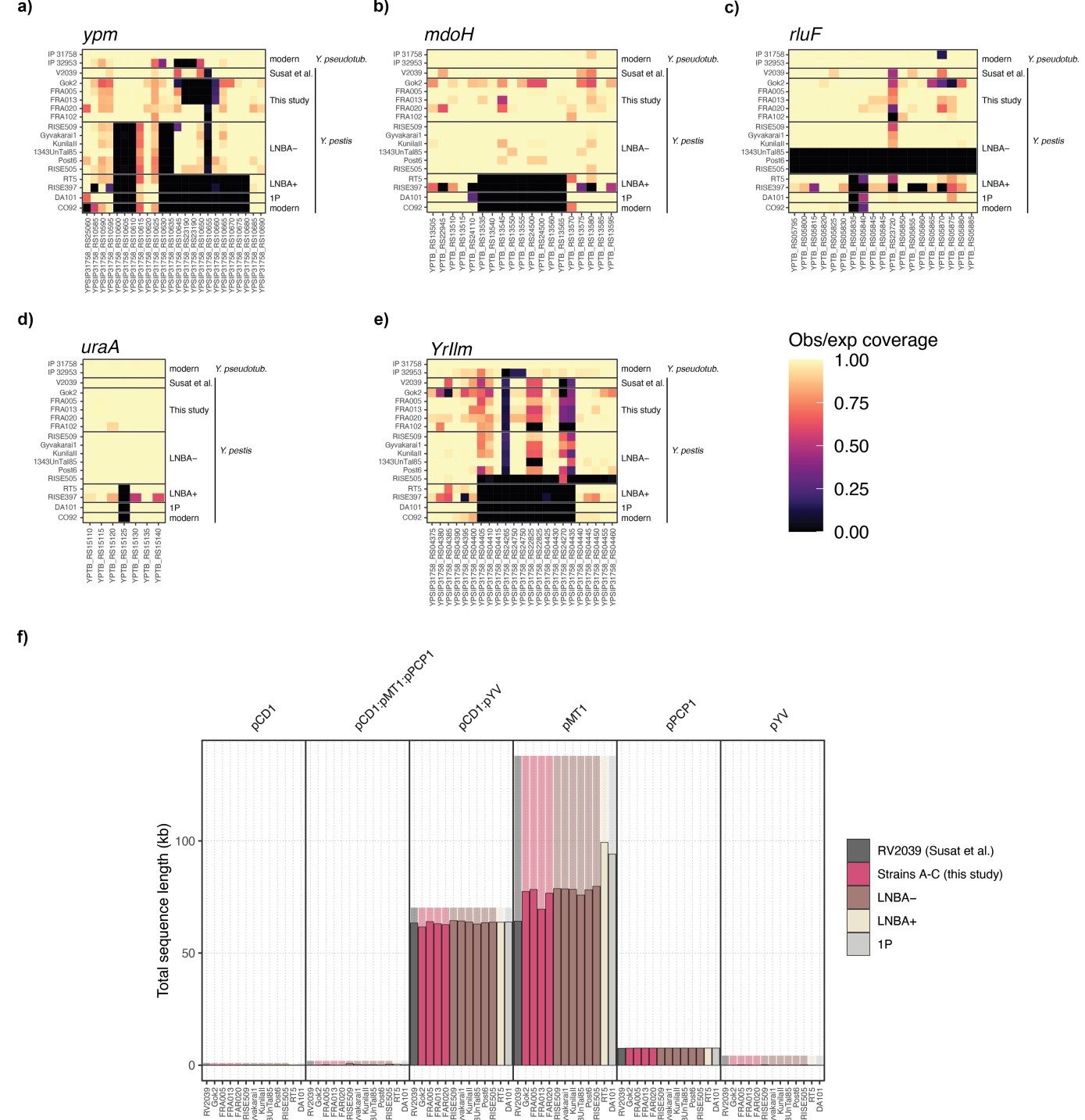

**Extended Data Fig. 10 | Pangenomic analysis of ancient plague genomes. a-e)** Per-gene coverage statistics from *Y. pseudotuberculosis* complex pangenome variation graphs. Heatmaps showing presence/absence of genes in five genomic regions where ancient *Y. pestis* lineages harbour *Y. pseudotuberculosis* complex pangenome gene content absent in modern *Y. pestis*. Fill colour indicates the ratio of observed over expected breadth of coverage given the depth of coverage across the genome for each gene and ancient sample. **f)** Variations in plasmid gene content among select *Y. pestis* strains. Barplot showing total

length of reference graph nodes covered in a given sample. The plot is stratified into categories of reference plasmid presence/absence at a given node, and only nodes covered by any of the three *Y. pestis* plasmids are shown. I.e. 'plasmid pCD1:plasmid pYV' refers to gene content present in both pCD1 (from *Y. pestis*) and pYV (from *P. pseudotuberculosis*). Shaded background indicates the total amount of sequence within each class and coloured bars represent the total length of nodes within that category present in a particular sample. Groups of ancient strains are indicated by bar colour (1 P: first pandemic).

# Reporting Summary

## Statistics

For all statistical analyses, confirm that the following items are present in the figure legend, table legend, main text, or Methods section.

| n/a | Confirmed | |
|---|---|---|
| ☐ | ☒ | The exact sample size ($n$) for each experimental group/condition, given as a discrete number and unit of measurement |
| ☐ | ☒ | A statement on whether measurements were taken from distinct samples or whether the same sample was measured repeatedly |
| ☒ | ☐ | The statistical test(s) used AND whether they are one- or two-sided<br>*Only common tests should be described solely by name; describe more complex techniques in the Methods section.* |
| ☒ | ☐ | A description of all covariates tested |
| ☒ | ☐ | A description of any assumptions or corrections, such as tests of normality and adjustment for multiple comparisons |
| ☐ | ☒ | A full description of the statistical parameters including central tendency (e.g. means) or other basic estimates (e.g. regression coefficient) AND variation (e.g. standard deviation) or associated estimates of uncertainty (e.g. confidence intervals) |
| ☒ | ☐ | For null hypothesis testing, the test statistic (e.g. $F$, $t$, $r$) with confidence intervals, effect sizes, degrees of freedom and $P$ value noted<br>*Give P values as exact values whenever suitable.* |
| ☒ | ☐ | For Bayesian analysis, information on the choice of priors and Markov chain Monte Carlo settings |
| ☒ | ☐ | For hierarchical and complex designs, identification of the appropriate level for tests and full reporting of outcomes |
| ☒ | ☐ | Estimates of effect sizes (e.g. Cohen's $d$, Pearson's $r$), indicating how they were calculated |

*Our web collection on statistics for biologists contains articles on many of the points above.*

## Software and code

Policy information about availability of computer code

| Data collection | Illumina NovaSeq system |
|---|---|
| Data analysis | bcftools (1.16)<br>bedtools (v2.31.0)<br>convertf (version 5000)<br>epa-ng (v0.3.8)<br>gappa (v0.8.0)<br>GATK (4.3.0.0)<br>gcta64 (v1.94.1)<br>GLIMPSE (v1.1.1)<br>guppy (v1.1.alpha19-0-g807f6f3)<br>krakenuniq (0.5.7)<br>mapDamage (2.2.0-86-g81d0aca)<br>metadmg ()<br>minimap2 (2.24-r1122)<br>mosdepth (0.3.3)<br>openjdk (1.8.0_92)<br>pathPhynder (1.a)<br>phynder<br>picard MarkDuplicates (2.27.4)<br>plink (v1.90b6.21) |

```
primus (v1.9.0)
raxml-ng (1.2.0)
samtools (1.17)
seqtk (1.3-r106)
smartpca (eigensoft v. 8.0.0)
yhaplo (1.1.2)
KIN (0.1.0)
KINgaroo (0.1.0)

R packages:
viridis_0.6.2
viridisLite_0.4.1
treeio_1.22.0
tidyr_1.3.0
stringr_1.5.0
scales_1.2.1
 readr_2.1.4
RColorBrewer_1.1-3
purrr_1.0.1
kinship2_1.9.6
quadprog_1.5-8
Matrix_1.5-4
igraph_1.4.2
googlesheets4_1.1.0
ggtree_3.6.2
ggh4x_0.2.4
ggplot2_3.4.2
forcats_1.0.0
dplyr_1.1.1
doParallel_1.0.17
iterators_1.0.14
foreach_1.5.2
```

For manuscripts utilizing custom algorithms or software that are central to the research but not yet described in published literature, software must be made available to editors and reviewers. We strongly encourage code deposition in a community repository (e.g. GitHub). See the Nature Portfolio guidelines for submitting code & software for further information.

## Data

Policy information about availability of data

All manuscripts must include a data availability statement. This statement should provide the following information, where applicable:
- Accession codes, unique identifiers, or web links for publicly available datasets
- A description of any restrictions on data availability
- For clinical datasets or third party data, please ensure that the statement adheres to our policy

Fastq files will be made available on the European Nucleotide Archive upon publication of the manuscript.

## Research involving human participants, their data, or biological material

Policy information about studies with human participants or human data. See also policy information about sex, gender (identity/presentation), and sexual orientation and race, ethnicity and racism.

| | |
|---|---|
| Reporting on sex and gender | Not appllicable |
| Reporting on race, ethnicity, or other socially relevant groupings | Not appllicable |
| Population characteristics | Not appllicable |
| Recruitment | Not appllicable |
| Ethics oversight | Not appllicable |

Note that full information on the approval of the study protocol must also be provided in the manuscript.

# Field-specific reporting

Please select the one below that is the best fit for your research. If you are not sure, read the appropriate sections before making your selection.

☒ Life sciences    ☐ Behavioural & social sciences    ☐ Ecological, evolutionary & environmental sciences

# Life sciences study design

All studies must disclose on these points even when the disclosure is negative.

| | |
|---|---|
| Sample size | No tests were carried out to predetermine sample size. Sample size was determined by the availability of archaeological material, and on the DNA preservation in these samples. |
| Data exclusions | 7 DNA libraries were excluded from this study based on high contamination estimates as determined by ContamMix (v1.0.10; please see Supplementary Table 3 for details). Furthermore, another 3 libraries were flagged as suspected sample swaps and excluded because they differed fundamentally with other libraries from the same sample. Lastly all samples with a final depth of coverage under 0.01X were excluded from downstream analyses. |
| Replication | Out of the 109 genetic individuals analysed in this study, 95 individuals are represented by more than one sequencing library. Having multiple sequencing libraries for each sample serves to validate sequencing results and to pinpoint potential sample swaps. Apart from this type of replication, replication of experimental findings is generally not applicable for this kind of ancient DNA study because of the unique nature of ancient human remains. |
| Randomization | Not appllicable |
| Blinding | Not appllicable |

# Reporting for specific materials, systems and methods

We require information from authors about some types of materials, experimental systems and methods used in many studies. Here, indicate whether each material, system or method listed is relevant to your study. If you are not sure if a list item applies to your research, read the appropriate section before selecting a response.

## Materials & experimental systems

| n/a | Involved in the study |
|---|---|
| ☒ | Antibodies |
| ☒ | Eukaryotic cell lines |
| ☐ | ☒ Palaeontology and archaeology |
| ☒ | Animals and other organisms |
| ☒ | Clinical data |
| ☒ | Dual use research of concern |
| ☒ | Plants |

## Methods

| n/a | Involved in the study |
|---|---|
| ☒ | ChIP-seq |
| ☒ | Flow cytometry |
| ☒ | MRI-based neuroimaging |

## Palaeontology and Archaeology

| | |
|---|---|
| Specimen provenance | Hunnebostrand (SHM 7532:107a+b) was sampled in 2006 at Statens Historiska Museum (SHM) in Stockholm by verbal agreement with Leena Drenzel, SHM. Frälsegården (VGM 1M16-107047) is stored in-house at the Department of Historical Studies, Gothenburg university. Sampling for DNA and other analyses is ongoing by verbal agreement with Maria Vretemark, Västergötlands museum, Skara. Avleberg (NM A 37692-701) is stored at the National Museum, Copenhagen. It was sampled and published by a previous project (Allentoft et al 2022, 2023). The samples from Rössberga were analysed as part of the Atlas project in 2013 under permit number 33-696-2013, issued by SHM (Statens Historiska Museum). The tooth from Firse sten was sampled in 2019-10-30 as part of the project Megalitgravar på Falbygden with permission from Västergötlands museum. |
| Specimen deposition | Leftover DNA digests, extract and sequencing libraries are stored at the DNA laboratory facilities at Globe Institute, Copenhagen. Upon completion of this project, leftover bone material will be returned to respective museum or university collections from which they were sampled (see above). |
| Dating methods | Datings were performed at the Keck carbon cycle AMS facility, University of California, Irving. The samples were decalcified in 1N HCl, gelatinized at 60°C and pH 2, and ultrafiltered to select a high molecular wt fraction (>30kDa). δ13C and δ15N values were measured to a precision of <0.1‰ and <0.2‰, respectively, on aliquots of ultrafiltered collagen, using a Fisons NA1500NC elemental analyzer/ Finnigan Delta Plus isotope ratio mass spectrometer. Datings were calibrated in Oxal 4.4.4 using the Intcal20 calibration curve. |

☒ Tick this box to confirm that the raw and calibrated dates are available in the paper or in Supplementary Information.

| | |
|---|---|
| Ethics oversight | No ethical approval was required for this study. |

Note that full information on the approval of the study protocol must also be provided in the manuscript.

