## [Peer Review file · Nature]

Manuscript Title: Repeated Plague Infections Across Six Generations of Neolithic Farmers

Reviewer Comments & Author Rebuttals

Reviewer Reports on the Initial Version:

Referees' comments:

Referee #1 (Remarks to the Author):

A. Summary of the key results: Seerholm et al. obtain ancient DNA data from 109 Scandinavian Neolithic individuals preserved in 8 megalithic graves and one stone cist (out of a total of 174 samples tested). Kinship analysis reveals a patrilineal society, with women joining the community of their partner, and tolerance for close-kin unions (third degree). A total of 17% of the individuals analyzed turn out to be positive for plague, suggesting plague epidemics as a key driver of the so-called 'Neolithic decline' which dramatically impacted the European demography some 5,300-4,900 years ago. Implementing the first variant graph-based pan-genome reconstruction of ancient plague strains, the authors identify the presence of several candidate mutations potentially enhancing strain virulence and disease outcome. They propose that plague epidemics were a key driver of the collapse of Neolithic society in Scandinavia.

B. Originality and significance: The study is original as it investigates *Yersinia pestis* as a key driver of human population collapse by the end of fourth millennium BCE in Scandinavia. It follows up on a hypothesis previously proposed by Rascovan et al. *Cell* (2019), which was built on a single plague occurrence. The substantial proportion of individuals positive for plague, the spread from Hunnebostrand (Sweden) to Avlebjerg (South Denmark) (ie over 600 kilometers) the infection of a five-generations pedigree, and the identification of 3 genetically-distinct strains, indicate that the disease was clearly part of the reality of the populations living at the time, and not an isolated case. It contradicts previous interpretations, by Susat et al. *Cell Rep* (2021), of the disease as a result of sporadic zoonotic cases.

C. Data & methodology: The integration of human and pathogen DNA together with isotopes is excellent, and only rarely used in similar studies. The analyses based on variant-graph pan-genome reconstruction are particularly novel in this context, and significantly advances the state-of-the-art. It provides a new, powerful approach, and will represent the new standard for future studies.

D. Appropriate use of statistics and treatment of uncertainties: The methodology implemented is sound and robust. I only have a few, minor suggestions (see below), where I have found possible improvements.

E. Conclusions: The conclusions are generally valid and robust. They provide solid arguments for solving an important debate about the driver of a major population collapse in Neolithic Scandinavia. The study, however, leaves me with a bit of a paradox as the authors suggest plague as a key factor for the Neolithic decline. However, the fact that the community is maintained through 5 generations

is not suggestive of outbreaks that could entirely sweep out full communities. I thus wonder how much virulent could the strains detected be, if it infected so many people but left enough surviving ones to give birth to a new generation over and over again. This paradox must be addressed.

F. Suggested improvements: While most individuals show genetic profiles typical of European Neolithic populations, a total of 10 individuals associated with younger radiocarbon dates showed steppe-related genetic ancestry. The authors should date the time of the steppe-related admixture, eg using DATES. Similarly, the time of admixture for the two individuals showing substantial (up to 50%) hunter-gatherer ancestry should be dated as it has important implications for the timing of survival of hunter-gatherer populations in the region, and more broadly in Europe (see Seguin-Orlando et al. *Curr Biol* (2021) and Rivolat et al. *Sci Adv* (2020) for a discussion). Finding a 50-50% ancestry mixture in one individual does not imply that the person was a first generation hybrid between two populations. Node support for the phylogenetic tree shown in Supp Fig 9 should be provided. The authors seem to have tested whether the strains showed different copy numbers of plasmids (relative to the circular chromosome), as this may drive important differences in virulence (Supp Fig 13). This analysis is, however, left uncommented in the main text (and apparently un-referenced). It also remains unclear how such patterns could be driven by the number of probes represented in the capture kit utilised.

G. References: The study provides ample credit for previous work and analyses.

H. Clarity and context: The manuscript is well written, although generally un-necessarily wordy. For example, the abstract and the introduction include 1200+ words, which represents about half of the recommended size for such an article, and should be shortened. Statements such as 'we are in a much better position to...' should be avoided. While interesting for archaeologists, the long development done in the fourth paragraph of the introduction would remain far-too detailed for most readers. I would suggest to sharpen the introduction by focusing on the plague debate, leaving out marital rules. Those will be uncovered as the human DNA will be analysed anyway. The Results section should be shortened as well, especially the part describing the genealogy presented (and alternatives), which is un-necessarily lengthy. At the end of the day, the findings can be summarized in only a few words: genetic relatedness and isotope profiling support individuals belonging to a five-generation genealogy, organized around patrilineal marital rules and woman exogamy, and somehow permissive to inbreeding. The chronological span/modeling of the five-generations, which is presented over 3 paragraphs, could also be summarized in one sentence, and transferred to the Supp Material. The same holds to the section entitled 'Kinship reflects burial locations in the tomb', which is over-lengthy and not novel following the work done at Hazelton and Gurgy. The statement that 'siblings and couples tend to be buried close to each other' requires, however, formal statistical testing. I would also not qualify 'high-coverage genomes' genomes covered at least 1X. What are the lethality rates of the Far East Scarlet-like fever will be unclear to most readers, and should be clarified.

Referee #2 (Remarks to the Author):

As requested, I will focus on the radiocarbon dating and modelling only. I re-ran the OxCal models

and got comparable results - the added genetic constraints regarding kinship considerably reduce the chronological uncertainties of the individual unmodelled dates. I suggest the authors also present the OxCal output as additional Supplementary Figures - this to convey the remaining decadal-scale dating uncertainty.

In radiocarbon age-modelling it has become common to report 2sd ranges instead of 1sd ranges, as the latter are seen as not covering many possible uncertainties related to calibration and multi-modal peaks.

Given the highly coastal locations of the sites, it is surprising perhaps that no influence of any marine diet was found. Wrongly assuming no marine diet component could offset resulting modelled ages by centuries. The $\delta^{13}\text{C}$ values especially (Supp Table 4) clearly indicate a mostly terrestrial, non-marine diet - please highlight this (you do mention freshwater fish). Perhaps add a reference to the lack of marine diet. Was FTIR performed on the new Keck samples?

The radiocarbon age numbers in many of the Figures (Fig. 3, Supp.Figs 3-5) are reduced to a single number (the median, thus ignoring most of the considerable uncertainties) and printed in a tiny, largely unreadable font. Could the modelled cal BP ages in Fig. 3 instead be indicated by icon colour, e.g. graduating from red to blue between say 5150 and 4740 cal BP?

What are the pink polygons in Fig. 2a? I think they have to do with the calibrated radiocarbon dates, but it is unclear to me how. Could these be replaced by summed probabilities, or even better, OxCal-derived KDE distributions?

References to OxCal (e.g., Bronk Ramsey 2009) and IntCal20 (Reimer et al. 2020) will have to be added:

Bronk Ramsey, C. (2009). Dealing with outliers and offsets in radiocarbon dating. *Radiocarbon*, 51(3), 1023–1045.

Reimer, P., Austin, W., Bard, E., Bayliss, A., Blackwell, P., Bronk Ramsey, C., Butzin, M., Cheng, H., Edwards, R., Friedrich, M., Grootes, P., Guilderson, T., Hajdas, I., Heaton, T., Hogg, A., Hughen, K., Kromer, B., Manning, S., Muscheler, R., Palmer, J., Pearson, C., van der Plicht, J., Reimer, R., Richards, D., Scott, E., Southon, J., Turney, C., Wacker, L., Adolphi, F., Büntgen, U., Capano, M., Fahrni, S., Fogtmann-Schulz, A., Friedrich, R., Köhler, P., Kudsk, S., Miyake, F., Olsen, J., Reinig, F., Sakamoto, M., Sookdeo, A., & Talamo, S. (2020). The IntCal20 Northern Hemisphere radiocarbon age calibration curve (0–55 cal kBP). *Radiocarbon*, 62.

Referee #3 (Remarks to the Author):

This is an extremely interesting paper that synthesises a large quantity of (mainly genomic) data on Neolithic Scandinavians and their pathogens. The abstract, introduction and conclusions clearly outline the key points, which should be of interest to a wide range of readers. My expertise is mainly

in isotopes rather than DNA, so my comments focus on the strontium isotope component of the manuscript.

Of the 71 $^{87}\text{Sr}/^{86}\text{Sr}$ values in this paper, only 10 were measured for this study. 24 have been published previously. The largest number (37) are reported as "Sjögren pers comm", which means we know nothing about where or how they were measured. This is not acceptable. These data points need to be properly reported as new analyses for this paper.

Of these 47 new analyses, 3 were of bone and 44 of teeth (each a single tooth). Bone is well known to take up strontium from the burial environment. What steps were taken to investigate whether this was a problem for these three bone samples (all reported as "Sjögren pers comm")? How confident can we be that the values reported here are good indicators of the values in life?

For the teeth, what tissue was sampled - tooth enamel? How were the samples taken? e.g. over the height of the crown, thus averaging the period of crown formation, or some other way?

The statement "Sr isotopes in HUN002 are consistent with childhood in Falbygden, while HUN001 could originate from several places in Scandinavia outside Denmark" is somewhat misleading. Values for both individuals are within the Falbygden local range as defined by Blank et al 2021. Since $^{87}\text{Sr}/^{86}\text{Sr}$ values are not unique to particular localities, both individuals could also have originated from many places in Scandinavia. This sentence should be re-written.

On p. 18 of the combined document, under the heading "Strontium isotope analysis", the authors state that the "recognised value" for the SRM 987 is 0.710250. They standardise their measured $^{87}\text{Sr}/^{86}\text{Sr}$ to this value. As far as I am aware, the certified $^{87}\text{Sr}/^{86}\text{Sr}$ value for SRM 987 is 0.71034 ± 0.00026 (95% confidence interval).

Where are the supplementary references?

Referee #4 (Remarks to the Author):

The manuscript is very well written and the results are very well presented. Particularly exciting is the relationship between the groups studied, which reflects a certain correlation between the individual burial groups in Sweden.

The results on the early plague cases are very exciting and the findings are plausible. However, the interpretation of the results should be put into perspective. The graves do not show an increased mortality, so that it cannot be assumed that there was an excess mortality as a result of a pneumogenic outbreak. The high infestation rather speaks for a chronic disease.

I would therefore suggest softening the authors' interpretation. An epidemic outbreak and also single zoonoses have been disproven, so it seems to be a middle way.

Referee #5 (Remarks to the Author):

A. Summary of the key results

In this study, with title "Repeated Outbreaks of Plague Across Five Generations of Neolithic Farmers" Seersholm and colleagues present aDNA screening of 174 samples and subsequent human genomic analysis of 109 individuals from Neolithic farmer communities in nine Scandinavian archaeological

sites. The authors additionally present a metagenomic analysis of all individuals, identifying 18 *Y. pestis* infections (some of which are tentative) as well as the potential detection of additional pathogens such as *Borrelia recurrentis* and *Yersinia enterocolitica* in the studied datasets. The vast majority of presented data derive from a single site, Fräsegården, in Sweden, from where a pedigree of 35 individuals was reconstructed alongside four *Y. pestis* genomes with genomic coverage >1X. Human genomic and strontium isotopic results are used to infer a patrilineal social organisation among the studied communities. Finally, the authors make a major claim i.e., that the pathogen results shown here present sufficient evidence for the role of *Y. pestis* in the collapse of Neolithic populations of Scandinavia.

B. Originality and significance: if not novel, please include reference

- The vast majority of results presented in this study derive from the archaeological site of Fräsegården, in western Sweden, a site that has previously been subject to ancient DNA analysis in two publications, which show consistent results with those presented here. Specifically, previously studies have characterised both human aDNA and *Y. pestis* infections from Fräsegården (see Skoglund et al., *Science*, 2014 and Rascovan et al., *Cell*, 2019) and led to the reconstruction of a *Y. pestis* genome (Gökhem 2) identical to genomes characterised here. The study by Seersholm and colleagues is based on more individuals and, hence, contributes additional genomic data from the site.

- Moreover, the Neolithic kinship structures presented here are consistent, albeit with some expected distinctions, with previously published Neolithic communities from Britain (Fowler, Olalde et al., *Nature*, 2021) and France (Rivollat et al., *Nature*, 2022).

C. Data & methodology: validity of approach, quality of data, quality of presentation

-The study follows standard methods in the aDNA field and uses data generation and computational methodologies, both for human biological kinship reconstructions and for the phylogenetic placement of bacterial strains, that are on par with practices in the field.

- The paper is generally well written, although in parts overstatements are made. I find these to be unnecessary, for example:

L365 "...one swift plague outbreak that eradicated the entire Neolithic community in Falbygden."

L447 "For the first time, we report numerous plague cases from a single community...". This statement is inaccurate. Previous studies have also reported multiple *Y. pestis* cases from single prehistoric communities.

L448 "could have been following the pneumonic transmission route..." While this may have been the case, there is currently no evidence towards its support, therefore it remains speculative.

D. Appropriate use of statistics and treatment of uncertainties

- The term “prevalence” is used frequently within the manuscript. Prevalence is an epidemiological parameter, which is non-trivial to estimate from incomplete and biased archaeological datasets, such as the one presented here. I strongly suggest the use a different term to characterise *Y. pestis* identification rates.

- Datasets of 0.01X coverage, where <1% of the *Y. pestis* genome is expected to be covered (Table S6) should not be called “genomes”, these are at best genome-wide data.

- Some of the genomes that seem identical based on the *Y. pestis* phylogeny (Figure 4) are consistent with having at least some SNP differences according to Supplementary Figure 10 (e.g., FRA005 and Gok2). How are these differences interpreted? Given the slow mutation rate of *Y. pestis*, these might reflect different infection timings. If true, the notion of contemporaneous infections in FRA005, FRA013 and Gok2 might need to be reconsidered.

E. Conclusions: robustness, validity, reliability

- While the *Y. pestis* genomes presented here add new data to our understanding of prehistoric plague, I find their causal attribution to the Neolithic decline to be unsubstantiated. It is important to note that such early forms of plague have not only been identified during the period of the Neolithic decline, but were instead also present throughout the Bronze Age and all the way to the Iron Age (reflects the temporal interval of LNBA- lineages). Based on published data, infections with *Y. pestis* during prehistory seem to have been frequent across Eurasia for almost 3,000 years. Furthermore, these variants of *Y. pestis* were non-flea-adapted, meaning that their functional differences in terms of disease presentation, virulence and transmissibility compared to modern-day (and historical) plague are unknown and, so far, entirely hypothetical. Finally, societal collapse is an extremely complex phenomenon and a very challenging one to characterise in prehistoric contexts, therefore, its attribution to a single infectious disease seems highly simplistic and unrealistic.

-The authors mention that their estimated frequency of 17% is an underestimation of the true *Y. pestis* frequency in the population. Such argument is questionable given that the presented estimates include some very low-abundance identifications that may not be reliable.

- The authors seem to be clearly favouring a one-sided interpretation of their results, when several alternative possibilities for their findings can be considered. I strongly suggest the presentation of more balanced/nuanced statements in several part of the paper, for example:

L363 “Given the high prevalence of plague positive individuals in this study, and the relatively narrow time frame in which plague was detected, we hypothesised that our data represented one swift plague outbreak that eradicated the entire Neolithic community in Falbygden.” Such claims appear exaggerated on the basis of four reconstructed *Y. pestis* genomes. Moreover, I disagree that a ~150-year interval is narrow and could reflect a “swift” plague outbreak.

L380 “This idea is supported by the differential plague prevalence in the different sub-branches of the pedigree: In the last two generations of the left subfamily plague is detected in 5/6 individuals (83%).” The authors should be aware that individuals from multiple generations within a pedigree

can overlap in their lifetime, therefore, the *Y. pestis* frequency calculated for generation 5 does not reflect a prevalence estimate for a population, and is overall misleading.

L391 “Hence, it is likely that this female travelled to the Falbygden area during her lifetime, bringing the plague with her.” This statement makes the multi-fold assumption that (1) an individual was able to travel from Denmark to Sweden while being infected with a self-limiting and likely lethal pathogen, (2) that the source of the pathogen strain is consistent with that of the individual’s genomic and strontium background, and (3) that the region where the individual died could have not locally housed the retrieved pathogen diversity. Given that such assumptions are not supported by any evidence, alternative scenarios should have equal weight in the authors interpretation.

L411 “Partitioning the variant graph nodes into groups based on their presence pattern in the three species, we found that early divergent ancient plague lineages (preLNBA-, Frälsegården, and LNBA-) harbour up to 50 kb gene content present in some or all *Y. pseudotuberculosis*/*Y. similis* strains that is absent in later strains (LNBA+, 1P, modern; Figure 4b).” These results may also support the idea of Neolithic strains being an intermediate between *Y. pestis* and other, less virulent, *Yersinia pseudotuberculosis* complex diversity, and therefore may alternatively suggest an attenuated pathogenicity in contrast to what the authors claim. Why is this possibility not considered?

- Related to the point above, there is considerable variation in the detection of the *ypM* superantigen among LNBA- genomes, even for genomes that are seemingly phylogenetically identical (e.g., Gok2, FRA005 and FRA013). Yet the presence of this gene is used as evidence of highly virulent infections in Neolithic individuals. The authors should clarify this interpretation as Figure 4 does not show clear presence of the gene in all strains.

F. Suggested improvements: experiments, data for possible revision

- In Table S10, the headers are not self-explanatory and so the content cannot be evaluated in detail. Nevertheless, the table is meant to present an overview of all pathogen candidates that have passed authenticity filters based on the authors screening approach. In that regard, do the authors consider all those identifications to be authentic (ancient) and, if so, why have they specifically chosen to mention only the concurrent infections with *B. recurrentis* and *Y. enterocolitica* in the main text?

- Related to the point above, libraries with as little as ~70 detected reads are considered confidently positive for *Y. pestis*. As far as I can understand from Table S6 several of these were not subjected to whole-genome capture. Given the high false-positive rates known for ancient pathogen DNA in metagenomic datasets, how can these identifications be secure?

- Further related to Table S6, the number of identified *Y. pestis* reads are not fully informative if the total sequencing effort is not stated. This is especially true for low-abundance identifications which are of lower certainty if sequencing efforts are very deep.

- Previous studies have presented vastly different results for the divergence between *Y. pestis* and *Y. pseudotuberculosis* (see Rasmussen et al., Cell 2015, Rascovan et al., Cell, 2019 and Susat et al., Cell Reports, 2021). Given the new genomes presented here, a molecular dating analysis including all

new and previously published Neolithic datasets may provide new insights to this discussion.

G. References: appropriate credit to previous work?

- The authors should clearly state within the main text that the previously published Gok2 *Y. pestis* genome and the FRA genomes presented here are from the same archaeological site and temporal horizon.

H. Clarity and context: lucidity of abstract/summary, appropriateness of abstract, introduction and conclusions

- The abstract is presented clearly with the exception of the concluding sentence. "Taken together, our findings provide the first direct reconstruction of plague transmission within a large patrilineal kinship group and suggest that widespread ancient plague epidemics in Neolithic Scandinavian populations played a pivotal role in their collapse." I find this statement to be misleading as the manuscript does not formally infer pathogen transmission dynamics in this community, neither does it present evidence for population collapse.

- Finally, I find that the title overstates the results and conclusions of the study, given that the four presented *Y. pestis* genomes are insufficient evidence of "repeated outbreaks". First, the accompanying data on strains A and B do not conclusively show that they represent different time points and may well reflect diversity present contemporaneously within Frälsegården. Importantly, the individual associated with strain B has not been radiocarbon dated. Second, there is no evidence of strain C spreading beyond the single outlier individual from which it was retrieved and, therefore, is insufficient to indicate an outbreak. In addition, its date seems questionable given its basal positioning in the phylogeny (and short branch length), which comes in contrast with the individual's retrieved younger radiocarbon date.

Referee #6 (Remarks to the Author):

A. Summary of the key results/validity

This is an important paper which sells itself (in its title) primarily on the identification of plague across five generations during the Neolithic. This is certainly a very interesting and significant discovery, with ramifications for our understanding of possible Late Neolithic population decline. This finding will be of interest to historians of disease (and others working in the biological sciences more widely) as well as archaeologists. The paper is also important with regard to its discussion of other elements of Neolithic society in Sweden, including evidence for patrilinearity/patrilocality and kinship/spatial relations within the tomb – these are important issues that will be of significant interest to Neolithic scholars in particular.

The manuscript does not have any flaws, to my knowledge, that should prohibit its publication. In fact, I found the paper to be well written, measured in its tone, and sensible in its argument and conclusions. It pushes the boundaries of interpretation, but (except at the very end – see below) does not push them too far, or unnecessarily hard.

B. Originality and significance

The paper is original in terms of its identification of plague in relatively high numbers of individuals, across an extended period of time (five generations). It identifies different strains of plague and unpicks their chronological and ancestral relationship well. The paper is certainly of interest to archaeologists (within and also beyond those working on the Neolithic of Europe) and will, I believe, be of interest to historians of disease, etc. as well. In addition to these findings, the evidence laid out for patrilinearity/patrilocality is also very original and interesting for prehistorians.

C. Data & methodology: validity of approach, quality of data, quality of presentation

I am not an expert in aDNA and therefore will not attempt to comment on that element of the paper's methodology. The archaeology is dealt with well and the decisions made are clear. The quality of data and presentation are good.

D. Appropriate use of statistics and treatment of uncertainties

This is not my area of expertise (especially in relation to aDNA) and so I do not feel qualified to comment. The assumptions with regard to the radiocarbon dating seem sensible to me.

E. Conclusions: robustness, validity, reliability

The broad conclusions drawn from the study are, as mentioned above, rigorous, measured and appropriate. The few issues I have relate to clarity of expression and thought process. As they are all fairly minor, I have outlined them in the 'suggested improvements' section below.

F. Suggested improvements: experiments, data for possible revision

Line 28. "detected in at least 17% of the population" – please be clearer that this is 17% of the sampled population, which itself is a probably skewed sub-sample (i.e. the people who were actually buried in tombs which is likely to be a substantial minority) of the total population.

Line 234-236. The argument would benefit from some clarification here. The second interpretation ("all their offspring were daughters, who were married out and buried in other tombs") is to my mind much more probable than the first ("but did not produce offspring before they passed away"). If the authors want to keep this line in, they should be clearer that the first is actually "but did not produce offspring *who were buried within the tomb and have therefore been sampled* before they passed away". This is an important distinction.

Line 324. "Surprisingly" – why is this surprising? Remove or clarify.

Line 364. "we hypothesised that our data represented one swift plague outbreak that eradicated the entire Neolithic community in Falbygden". This seems an odd hypothesis given that, as explained in Lines 373-374, it occurred over 5 generations and probably over more than a century according to the C14 dates. If this was just a playful experiment, explain that. If not, explain why this was assumed given the time distance.

Line 399. "the high number of infected individuals analysed here suggests a relatively rapid spread of

the disease within the population". Again, clarify, given the 5 generations/120 year estimate.

Lines 429-430. Can these "lineages" be tied into the proposed Strains A/B/C mentioned previously? If they are the same, reference the latter here as well.

Line 457. "We show that the social organisation in Middle Neolithic Sweden was patrilineal...". As touched upon above, it might be worth clarifying that this is actually "We show that the social organisation in Middle Neolithic Sweden *as represented by the population sampled within these tombs* was patrilineal...".

Line 474. I do not think "appears to have been married off to..." is an appropriate term to use here as it implies a lack of agency on the part of the female (which is not certain) and a knowledge of "marriage" in the Neolithic (that we do not have). Could the wording simply be "appears to have moved to..."?

Lines 478-482. Within what is a really good paper, which generally does not overstate its findings, I did not feel that this ending fitted well, as it makes several (presumably intended to be headline grabbing) inaccurate claims. I suggest simply deleting this paragraph or re-wording it.

- "The social structure strongly favoured males" – we do not know this, we only know/surmise that the society was patrilocal/patrilineal, which is not the same thing.

- "females moved away" – this is not true, we have direct evidence for one female moving away.

- "was infecting large parts" – this again is not well worded, "was infecting a significant proportion" might be more accurate.

- "it [plague] may have contributed to undermining the social viability of society, leading to the collapse of this form of Neolithic society". I do not understand what is meant here – surely plague killed people which could have led to population "collapse"? If this is not what is meant, please clarify exactly how plague relates to society in more than a biological sense.

G. References: appropriate credit to previous work?

Yes.

H. Clarity and context: lucidity of abstract/summary, appropriateness of abstract, introduction and conclusions

Good.

Duncan Garrow, University of Reading, UK

Author Rebuttals to Initial Comments:

General remarks to reviewers

We would like to thank the reviewers for their time - their comments and suggestions have been a big help. We present a revised manuscript that we believe is a significant improvement over the last iteration in almost every aspect. We have made major revisions in three main areas: 1) Conclusions related to the impact on plague have been softened and we have rewritten the entire discussion on plague to better present alternative interpretations of our data. 2) We have significantly shortened the paper, by moving the section on chronological modelling to the Supplementary Information and by shortening the Introduction and the section on burial practices. 3) We have carried out numerous new analyses to support the main conclusions, including a molecular dating analysis of ancient plague strains, simulations of the effects of damage on SNP calling, a DATES analysis, a BLAST analysis of putative plague reads, RFMix chromosome painting and simulations of read data from F1 individuals. Furthermore, we have obtained a new date for the individual with strain C that clarifies the ancestral position of this strain in the phylogeny. Lastly, we have adopted the reviewer's suggestions throughout the manuscript.

We have addressed all of the comments below and hope that the revised version of the manuscript is suitable for publication in Nature.

For easier reading, we have copied all reviewer comments into this letter (blue text color). The response to each comment is in black text color.

Lastly, please note that we managed to add three more individuals to pedigree 1. We have updated relevant figures and the text accordingly.

REVIEWER COMMENTS:

Referee #1 (archaeogenetics):

A. Summary of the key results: Seerholm et al. obtain ancient DNA data from 109 Scandinavian Neolithic individuals preserved in 8 megalithic graves and one stone cist (out of a total of 174 samples tested). Kinship analysis reveals a patrilineal society, with women joining the community of their partner, and tolerance for close-kin unions (third degree). A total of 17% of the individuals analyzed turn out to be positive for plague, suggesting plague epidemics as a key driver of the so-called 'Neolithic decline' which dramatically impacted the European demography some 5,300-4,900 years ago. Implementing the first variant graph-based pan-genome reconstruction of ancient plague strains, the authors identify the presence of several candidate mutations potentially enhancing strain virulence and disease outcome. They propose that plague epidemics were a key driver of the collapse of Neolithic society in Scandinavia.

B. Originality and significance: The study is original as it investigates *Yersinia pestis* as a key driver of human population collapse by the end of fourth millennium BCE in Scandinavia. It follows up on a hypothesis previously proposed by Rascovan et al. *Cell* (2019), which was built on a single plague occurrence. The substantial proportion of individuals positive for plague, the spread from Hunnebostrand (Sweden) to Avlebjerg (South Denmark) (ie over 600 kilometers) the infection of a five-generations pedigree, and the identification of 3 genetically-distinct strains, indicate that the disease was clearly part of the reality of the populations living at the time, and not an isolated case. It contradicts previous interpretations, by Susat et al. *Cell Rep* (2021), of the disease as a result of sporadic zoonotic cases.

C. Data & methodology: The integration of human and pathogen DNA together with isotopes is excellent, and only rarely used in similar studies. The analyses based on variant-graph pan-genome reconstruction are particularly novel in this context, and significantly advances the state-of-the-art. It provides a new, powerful approach, and will represent the new standard for future studies.

D. Appropriate use of statistics and treatment of uncertainties: The methodology implemented is sound and robust. I only have a few, minor suggestions (see below), where I have found possible improvements.

E. Conclusions: The conclusions are generally valid and robust. They provide solid arguments for solving an important debate about the driver of a major population collapse in Neolithic Scandinavia. The study, however, leaves me with a bit of a paradox as the authors suggest plague as a key factor for the Neolithic decline. However, the fact that the community is maintained through 5 generations is not suggestive of outbreaks that could entirely sweep out full communities. I thus wonder how much virulent could the strains detected be, if it infected so many people but left enough surviving ones to give birth to a new generation over and over again. This paradox must be addressed.

We agree. We have addressed this issue by rewriting the entire discussion on plague. Most importantly we have clarified our interpretation of the YPM data, which could suggest that a

recombination event rendered strain A more virulent than strains C and B. Furthermore, we have clarified what our main findings are, how we interpret these findings, and outlined other possible scenarios that could explain our data. We interpret our data as evidence of an early outbreak of strain C which the family line survived, followed by later outbreaks of strains A and B. In particular, the outbreak of strain A in the left-side subfamily seems to have been prevalent and it follows many of the characteristics that we would expect in a deadly epidemic. However, any assumptions on excess mortality from these plague cases remain entirely hypothetical. We have outlined this in the rewritten discussion on plague:

The distribution of plague positive individuals in the pedigree presented in Figure 3, does not readily support a swift and deadly plague epidemic, because plague is detected in all generations except generation two. However, when information on the different plague strains is taken into account, it becomes apparent that plague infections are stratified both chronologically and phylogenetically into two separate clusters. The most ancestral form of the plague (strain C) is also detected in the oldest individual from this study (FRA102) who was buried in the northern part of the chamber with individuals from generations one and two despite being unrelated to everyone in the pedigree. Based on placements of lower-coverage genomes, we also identify a plague form similar to strain C in FRA021, the progenitor of the right-side subfamily. This finding suggests a spread of an early form of the disease in the first generation of the pedigree. The mortality rate of this form of the plague is unknown, but the pedigree clearly illustrates that the family as a whole and the line of FRA021 survived the disease.

The second cluster of plague infections occur in generations 3-5 and is caused by strains A and B. Strain A is found in generation four of the left-side family, where all individuals appear to have been infected by the plague. Strain B on the other hand, is found in a single individual from the right-side subfamily, most likely from generation three. Given the genetic distance between strains A and B and their different distribution across the pedigree, it is possible that the two strains represent separate outbreaks of the disease: one in the left-side subfamily (strain A), and another in the right-side subfamily (strain B). Even though both strains appear to be contemporaneous in Figure 3, they could be temporally distinct as the error margins of the chronological modelling allows for considerable variation in the modelled dates (Supplementary Figure 24). Unfortunately, it is not possible to further assess this hypothesis as we are unable to distinguish strains A and B in the lower-coverage genomes (Supplementary Figure 8).

We can only speculate on the impact that the disease had on the local population, but we note that all of the hallmarks expected in a swift and deadly epidemic is present for the spread of strain A in the left-hand subfamily: 1) The frequency of plague positive individuals is exceptionally high 2) The disease is restricted to the last two generations, and 3) all higher-coverage genomes from this subfamily are identical. Taken together, these observations could suggest that an outbreak of strain A led to the demise of the left-side subfamily, perhaps driven by increased pathogenicity due to recombination around the ypm locus in this strain (discussed below).

Furthermore, we have updated our discussion on the YPM findings:

In order to distinguish the three known ypm loci (ypmA, ypmB and ypmC) we characterised gene presence/absence in the genetic environment surrounding this unstable locus (figure 4). We identify a pattern of genetic organisation resembling the ancestral locus, ypmB, in RV2039, and Falbygden strains C and B, while we identify two hitherto unknown combinations of genes in the ypm locus for strain A and later LNBA- strains. Yersinia pseudotuberculosis strains carrying the ypmB allele have been associated with low pathogenicity in humans²⁹, however the virulence of the unknown ypm allele identified in strain A cannot be determined. It is, however, well established that some ypm alleles have epidemic potential. [...]

F. Suggested improvements: While most individuals show genetic profiles typical of European Neolithic populations, a total of 10 individuals associated with younger radiocarbon dates showed steppe-related genetic ancestry. The authors should date the time of the steppe-related admixture, eg using DATES. Similarly, the time of admixture for the two individuals showing substantial (up to 50%) hunter-gatherer ancestry should be dated as it has important implications for the timing of survival of hunter-gatherer populations in the region, and more broadly in Europe (see Seguin-Orlando et al. Curr Biol (2021) and Rivolat et al. Sci Adv (2020) for a discussion). Finding a 50-50% ancestry mixture in one individual does not imply that the person was a first generation hybrid between two populations.

As suggested, we carried out a DATES analysis to estimate the timing of admixture in the steppe related groups and in the admixed individuals with Neolithic and hunter-gatherer ancestry. For the steppe related groups, we estimate that admixture occurred 16.3 ± 4.34 and 35.3 ± 5 generations ago for the Steppe 1 and Steppe 2 groups, respectively. Taking the radiocarbon dating results from each group into account, and assuming a generation time of 25 years, this result dates the admixture of steppe and Neolithic ancestry to 4,748 BP (4,904-4,592) and 4,750 BP (4,927-4,573). These estimates suggest that admixture happened in a single pulse, probably prior to the arrival of steppe DNA in Sweden. We present these results in Supplementary Figure 20 and address it in the paragraph on IBD results from the steppe-related groups in the main text:

Using DATES, we were able to date the admixture of 'Steppe' and 'Farmer' DNA in these two groups²⁴. For both groups we found that admixture most likely happened around 4,750 cal. BP (Supplementary Figure 20). In agreement with recent results showing that Steppe related groups first appeared in Eastern Europe around 4,800 years ago²³, this finding suggests that admixture occurred in a single pulse prior to the arrival of CWC groups in Sweden.

For reference the supplementary figure is appended below:

Supplementary Figure 20. Estimates of admixture timing in Steppe related Individuals. Admixture time estimates were calculated with DATES using Swedish Neolithic farmer related individuals and Yamnaya individuals as source groups (see methods). **a)** Sample-wise weighted linkage disequilibrium measures against genetic distance in centimorgan. In the top right corner of each plot, mean relative admixture time and standard error is shown in generations ago. **b)** Estimates of absolute admixture time using a generation time of 25 years and the calibrated ages of each sample (Supplementary Table 4; shown in grey). Highlighted in grey is the time period between 4,718 BP and 4,758 BP where confidence intervals of all samples except FIR001 overlap. **c)** Weighted linkage disequilibrium measures against genetic distance estimated for the two Steppe related groups. **d)** Estimates of absolute admixture time for the two Steppe related groups using the average age of samples in each group (shown in grey) as the group age.

For the two admixed individuals with Neolithic and hunter-gatherer ancestry, we also carried out a DATES analysis. The results from this test were not as clear as for the Steppe related groups described above and we decided to carry out further analyses using RFMix and simulated F1-individuals to better address the origin of hunter-gatherer ancestry in the two admixed individuals. Based on these tests we tentatively conclude that FRA108 is the first-generation offspring of parents of the Pitted Ware Culture and of the Funnel Beaker culture, whereas ROS027 might have been an F2- or F3 individual. We discuss this in detail in Supplementary Note 5:

Supplementary Note 5 - An investigation of the two admixed individuals

We identify significant proportions of hunter-gatherer DNA in the two individuals FRA108 and ROS027, both of which appear to be shifted towards individuals of the Pitted Ware Culture on our PCA plot (Figure 2). As the Pitted Ware Culture coexisted with Neolithic Farmers in Sweden for at least 600 years¹⁰¹, and given the high proportions of hunter-gatherer DNA in these individuals, we hypothesised that a relatively recent admixture event must have occurred. In order to investigate this theory, we ran DATES²⁴ on the two admixed individuals and one Neolithic individual with no evidence of recent admixture (ROS016) as a control. As source populations we used individuals of the Pitted Ware culture and Neolithic individuals from this study (see Supplementary Note 2 - Supplementary Methods). Using this approach we found evidence for very recent admixture in ROS027 (2.6 ± 2.9 generations ago; Supplementary Figure 21), while for FRA108, we did not get meaningful results (-11.5 ± 55 generations ago; Supplementary Figure 21). In the case of an F1-individual, DATES would fail since there is no crossover between the two ancestries in a first generation individual. Hence, the negative result from DATES for FRA108 may suggest that this individual is in fact a first-generation Neolithic/Hunter-gatherer offspring. Similarly, with the results from DATES we can rule out that ROS027 is a first generation offspring, and given the low estimate of time of admixture (1.4 ± 2.2 generations ago), it is most likely that ROS027 represents an F2- or F3 individual.

In order to further investigate how DATES behaves in the case of an F1-individual, we simulated DNA reads of five F1-individuals with the same coverage as FRA108 (see Supplementary Methods). As depicted in Supplementary Figure 21, DATES produced highly different results for each of the five simulated F1-individuals, and thus this test did not bring us closer to characterise the ancestry of FRA108.

Next, we decided to test RFMix⁹⁸ to paint local ancestry across the genomes of the two admixed individuals using Pitted-Ware hunter-gatherers and Neolithic Farmers as sources (Supplementary Figure 22). We found that FRA108 had 47.5% hunter-gatherer DNA and 52.5% Neolithic DNA, while ROS027 had 34.1% hunter-gatherer DNA and 65.9% Neolithic DNA. In a perfectly phased F1-individual, each position across the genome should have one allele each of the two ancestries. This is not what we observe for the suspected F1-individual FRA108,

instead we only found this pattern in 50.4% of the genome (Supplementary Figure 22c). However, the genome of FRA108 is only at 1.3X coverage, and is not perfectly phased. Hence, to test the effects of the imputation in combination with the relatively low coverage for these types of analyses, we ran RFMix on the five simulated F1-individuals described above. These five simulated genomes behaved similarly as FRA108, both in terms of ancestry proportions and the proportion of ‘ancestry-wise heterozygous’ positions. Hence, based on these observations, we tentatively conclude that FRA108 is the first-generation offspring of parents of the Pitted Ware Culture and of the Funnel Beaker culture.

For reference the two Supplementary Figures 21 and 22 are included below:

Supplementary Figure 21. Estimates of admixture timing in admixed and simulated F1-individuals. Admixture time estimates were calculated with DATES using Swedish Neolithic farmer related individuals and PWC individuals as source groups (see methods). One unadmixed individual (ROS016) was included as a control.

Supplementary Figure 22. Local ancestry inference of admixed individuals. RFmix estimates of hunter-gatherer (PWC) and Neolithic Swedish ancestries for admixed individuals (FRA108 and ROS027), a Neolithic individual with no evidence of recent admixture (ROS016) and five simulated F1 individuals with equal parts HG and Neolithic DNA. **a)** Ancestry painting across each chromosome. To simplify the plot, only chromosomes one to five and one simulated individual is shown. **b)** Total fraction of each ancestry type across the genome of each individual. **c)** Total fraction of genotype class across the genome of each individual, where ‘het’ indicates regions where one allele is of HG ancestry while the other is of Neolithic ancestry, while ‘hom’ indicates that both alleles are of the same ancestry.

Node support for the phylogenetic tree shown in Supp Fig 9 should be provided.

Corrected. We tested both of the bootstrap metrics implemented in raxml: the traditional Felsenstein's bootstrap proportions (FBP) metric and the Transfer Bootstrap Expectation (TBE) metric suited for large datasets. Both metrics produced similar results, and we have plotted the TBE bootstrap values onto the tree in Supplementary Figure 9:

Supplementary Figure 9. Full phylogenetic tree. Phylogenetic relationship between all previously published plague strains and the data produced for this study. Each circle represents one plague genome. Phylogenetic clades relevant for this study have been highlighted in colour and only ancient samples were labelled. Transfer Bootstrap Expectation (TBE) values are shown for relevant nodes.

The authors seem to have tested whether the strains showed different copy numbers of plasmids (relative to the circular chromosome), as this may drive important differences in virulence (Supp Fig 13). This analysis is, however, left uncommented in the main text (and apparently un-referenced). It also remains unclear how such patterns could be driven by the number of probes represented in the capture kit utilised.

Thank you for your comment, the description of Supplementary Figure 13 was not clear. The figure illustrates variations in plasmid gene content among different samples, and characterises the reference source(s) of that gene content. While some nodes are unique only to a single plasmid, others are shared among multiple plasmids. The *Y. pestis* plasmid pCD1 and the *Y. pseudotuberculosis* plasmid pYV, for example, share large homologous regions totalling around 70kb. The shaded background in the plot represents the total length of that category, whereas the colored bars represent the total length of nodes within that category present in a particular sample. The difference in gene content for pMT1 between the LNBA- and preLNBA strains on one hand and the LNBA+ and 1p strains on the other, is explained by the absence of the *ymt* gene in the oldest plague strains. We have clarified this in the figure legends of Supplementary Figure 13 (and Figure 4b), which now reads:

Supplementary Figure 13. Variations in plasmid gene content among select *Y. pestis* strains. Barplot showing total length of reference graph nodes covered in a given sample. The plot is stratified into categories of reference plasmid presence/absence at a given node, and only nodes covered by any of the three *Y. pestis* plasmids are shown. I.e. 'plasmid pCD1:plasmid pYV' refers to gene content present in both pCD1 (from *Y. pestis*) and pYV (from *P. pseudotuberculosis*). Shaded background indicates the total amount of sequence within each class and coloured bars represent the total length of nodes within that category present in a particular sample. Groups of ancient strains are indicated by bar colour (1P: first pandemic).

As suggested, we also investigated the effect of capture enrichment on copy-number estimates in our data. As depicted below (Review Figure 1), we found a linear trend between estimates from shotgun and capture libraries, with a slight increase in the coverage of pPCP1 for the captured libraries. Based on this plot, we conclude that the effect of our capture approach on copy number estimates should be relatively small.

Review Figure 1. Comparison of plasmid coverage between shotgun and captured libraries. Coverage was normalised to the coverage of the chromosome for each library.

Next, we compared copy number estimates for all ancient and modern plague strains used in our main phylogeny. As depicted below (Review figure 2), we found elevated copy numbers for pPCP1 in our samples compared to most other ancient groups. An increase in pPCP1 copy numbers has before been associated with increased virulence because of the virulence gene *pla* encoded on the pPCP1 plasmid. As such, this result could point towards a higher virulence for our test samples compared to other ancient samples, such as the LNBA- strains. We are, however, hesitant to draw this conclusion and to present these results in the paper, as multiple confounding factors could affect these estimates (e.g. sample material, extraction protocol and library build protocol). Furthermore, while we have demonstrated that capture enrichment does not have a large effect on copy number estimates in our data, this assumption does not necessarily hold true for other capture strategies used to generate publicly available data. We have thus not included these figures in the paper, but we are willing to include them if the reviewer finds it necessary.

Review figure 2. Comparison of plasmid coverage between ancient and modern samples. Coverage was normalised to the coverage of the chromosome for each library. Test: this study.

G. References: The study provides ample credit for previous work and analyses.

H. Clarity and context: The manuscript is well written, although generally un-necessarily wordy. For example, the abstract and the introduction include 1200+ words, which represents about half of the recommended size for such an article, and should be shortened. Statements such as 'we are in a much better position to...' should be avoided. While interesting for archaeologists, the long development done in the fourth paragraph of the introduction would remain far-too detailed for most readers. I would suggest to sharpen the introduction by focusing on the plague debate, leaving out marital rules. Those will be uncovered as the human DNA will be analysed anyway. The Results section should be shortened as well, especially the part describing the genealogy presented (and alternatives), which is un-necessarily lengthy. At the end of the day, the findings can be summarized in only a few words: genetic relatedness and isotope profiling support individuals belonging to a five-generation genealogy, organized around patrilineal marital rules and woman exogamy, and somehow permissive to inbreeding. The chronological span/modeling of the five-generations, which is presented over 3 paragraphs, could also be summarized in one sentence, and transferred to the Supp Material. The same holds to the section entitled 'Kinship reflects burial locations in the tomb', which is over-lengthy and not novel following the work done at Hazelton and Gurgy.

We agree. As suggested, we transferred the chronological modelling to the supplement and shortened the introduction. Regarding the burial locations, we have done some shortening and rephrasing but

we still feel this is an important result, and we have kept this section in for now. Yet, if the reviewer finds it necessary we will also move this section to the Supplementary Information.

The statement that 'siblings and couples tend to be buried close to each other' requires, however, formal statistical testing.

Thank you for noticing this. We have investigated this further, and while some couples and siblings are indeed buried together, we do not find evidence of these being buried near each other systematically. Hence, we have removed this statement.

I would also not qualify 'high-coverage genomes' genomes covered at least 1X.

Thank you for your comment. To correct this, we decided to rename our three coverage categories as follows: tentative detections (<0.01x), lower-coverage partial genomes (0.01-1x) and higher-coverage partial genomes (>1x).

What are the lethality rates of the Far East Scarlet-like fever will be unclear to most readers, and should be clarified.

Most literature on Far East Scarlet-like fever relates to the 1959 outbreak in Vladivostok and is written in Russian and published in obscure journals. Hence, we have not been able to obtain an estimate of lethality rates. However, we have clarified the epidemic potential of the disease in the main text:

*FESLF has been described as an 'epidemic manifestation' of *Y. pseudotuberculosis* infection in humans, and in 1959 an outbreak in Vladivostok, U.S.S.R, caused the hospitalisation of 200 out of 300 patients^{30,31}.*

Referee #2 (radiocarbon dating):

As requested, I will focus on the radiocarbon dating and modelling only. I re-ran the OxCal models and got comparable results - the added genetic constraints regarding kinship considerably reduce the chronological uncertainties of the individual unmodelled dates. I suggest the authors also present the OxCal output as additional Supplementary Figures - this to convey the remaining decadal-scale dating uncertainty.

Corrected. Oxcal plots have been added as Supplementary Figures 14-16, and the models have been re-run with a couple of small modifications.

In radiocarbon age-modelling it has become common to report 2sd ranges instead of 1sd ranges, as the latter are seen as not covering many possible uncertainties related to calibration and multi-modal peaks.

Corrected. All calibrated dates are reported as 2sd ranges in Table S4 and Tables S11-13. 2sd ranges have also been added to the text in Supplementary Note 3.

Given the highly coastal locations of the sites, it is surprising perhaps that no influence of any marine diet was found. Wrongly assuming no marine diet component could offset resulting modelled ages by centuries. The $\delta^{13}\text{C}$ values especially (Supp Table 4) clearly indicate a mostly terrestrial, non-marine diet - please highlight this (you do mention freshwater fish). Perhaps add a reference to the lack of marine diet.

In fact, only one site is coastal, the Hunnebostrand site in Bohuslän. All sites in Falbygden are located some 150 km from the coast. The lack of marine $\delta^{13}\text{C}$ signal is therefore not so surprising in most cases. Freshwater input is also marginal according to the isotope results. As regards Hunnebostrand, some marine input could be expected but no marine isotope signal is at hand. This is in line with previous results suggesting a selective emphasis on domestic resources in the middle Neolithic TRB. The paragraph in question has been modified:

All samples were dated successfully, and showed C/N ratios well inside the accepted range for well preserved collagen, 2.9-3.6⁶⁵. $\delta^{13}\text{C}$ and $\delta^{15}\text{N}$ values indicate a fully terrestrial diet with little or no contribution from marine or freshwater protein, suggesting reservoir effects are negligible (Supplementary Table 4).

Was FTIR performed on the new Keck samples?

No, FTIR was not carried out on any of the samples as none of the bones was cremated.

The radiocarbon age numbers in many of the Figures (Fig. 3, Supp.Figs 3-5) are reduced to a single number (the median, thus ignoring most of the considerable uncertainties) and printed in a tiny, largely unreadable font. Could the modelled cal BP ages in Fig. 3 instead be indicated by icon colour, e.g. graduating from red to blue between say 5150 and 4740 cal BP?

Thank you for your suggestion. We agree that none of our figures properly displayed the uncertainties of the chronological modelling. To fix this, we initially drafted several versions of Figure 3 with varying

amounts of additional information on the modelling. However, all iterations of this figure became too crowded, so we opted for an alternative solution where we compiled all of the relevant information in one Supplementary Figure:

Supplementary Figure 24. Chronological modelling. a) Median ages and 2σ age ranges for modelled and unmodelled dates. Plot is stratified by pedigree branch (left side and right side, respectively) and generation in pedigree. **b)** Pedigree 1 from Frälsegården coloured by modelled age, following the color scheme of a. Grey color indicates individuals where no modelled age is available. Such individuals with missing data are comprised by 1) individuals that cannot be placed in the pedigree (stippled lines) or 2) individuals with dates classified as outliers (Supplementary Table 4).

What are the pink polygons in Fig. 2a? I think they have to do with the calibrated radiocarbon dates, but it is unclear to me how. Could these be replaced by summed probabilities, or even better, OxCal-derived KDE distributions?

Corrected. KDE distributions were used for Figure 2a.

References to OxCal (e.g., Bronk Ramsey 2009) and IntCal20 (Reimer et al. 2020) will have to be added:

Bronk Ramsey, C. (2009). Dealing with outliers and offsets in radiocarbon dating. *Radiocarbon*, 51(3), 1023–1045.

Reimer, P., Austin, W., Bard, E., Bayliss, A., Blackwell, P., Bronk Ramsey, C., Butzin, M., Cheng, H., Edwards, R., Friedrich, M., Grootes, P., Guilderson, T., Hajdas, I., Heaton, T., Hogg, A., Hughen, K., Kromer, B., Manning, S., Muscheler, R., Palmer, J., Pearson, C., van der Plicht, J., Reimer, R., Richards, D., Scott, E., Southon, J., Turney, C., Wacker, L., Adolphi, F., Büntgen, U., Capano, M., Fahrni, S., Fogtmann-Schulz, A., Friedrich, R., Köhler, P., Kudsk, S., Miyake, F., Olsen, J., Reinig, F., Sakamoto, M., Sookdeo, A., & Talamo, S. (2020). The IntCal20 Northern Hemisphere radiocarbon age calibration curve (0–55 cal kBP). *Radiocarbon*, 62.

Corrected. References have been added.

Referee #3 (archaeology/isotope analysis):

This is an extremely interesting paper that synthesises a large quantity of (mainly genomic) data on Neolithic Scandinavians and their pathogens. The abstract, introduction and conclusions clearly outline the key points, which should be of interest to a wide range of readers. My expertise is mainly in isotopes rather than DNA, so my comments focus on the strontium isotope component of the manuscript.

Of the 71 $^{87}\text{Sr}/^{86}\text{Sr}$ values in this paper, only 10 were measured for this study. 24 have been published previously. The largest number (37) are reported as “Sjögren pers comm”, which means we know nothing about where or how they were measured. This is not acceptable. These data points need to be properly reported as new analyses for this paper.

Corrected. The methods section has been reformulated and the relevant data are reported as “this study” in table S5. All Sr determinations were made at the same lab.

Of these 47 new analyses, 3 were of bone and 44 of teeth (each a single tooth). Bone is well known to take up strontium from the burial environment. What steps were taken to investigate whether this was a problem for these three bone samples (all reported as “Sjögren pers comm”)? How confident can we be that the values reported here are good indicators of the values in life?

We are aware of this problem, and the bone samples were not used for mobility estimates but rather as indicators of the local baseline. They turned out to be similar to values in teeth from small animals, strengthening this assumption. Thus, we have decided to remove the bone samples as they are not used in any argument, and values from teeth are available for these individuals.

For the teeth, what tissue was sampled - tooth enamel? How were the samples taken? e.g. over the height of the crown, thus averaging the period of crown formation, or some other way?

Tooth enamel was sampled in all cases, in the form of bulk samples averaging the time of crown formation. These details were added to the methods section:

The measurements are bulk values taken on small pieces of tooth enamel, representing an average over the time of enamel formation.

The statement “Sr isotopes in HUN002 are consistent with childhood in Falbygden, while HUN001 could originate from several places in Scandinavia outside Denmark” is somewhat misleading. Values for both individuals are within the Falbygden local range as defined by Blank et al 2021. Since $^{87}\text{Sr}/^{86}\text{Sr}$ values are not unique to particular localities, both individuals could also have originated from many places in Scandinavia. This sentence should be re-written.

Thanks for spotting this. The passage has been rephrased:

Sr isotopes in the two individuals confirm different childhood residence (Supplementary Table S5). Both are consistent with childhood in Falbygden, but they could also originate from other places in Scandinavia outside Denmark.

On p. 18 of the combined document, under the heading “Strontium isotope analysis”, the authors state that the “recognised value” for the SRM 987 is 0.710250. They standardise their measured $^{87}\text{Sr}/^{86}\text{Sr}$ to this value. As far as I am aware, the certified $^{87}\text{Sr}/^{86}\text{Sr}$ value for SRM 987 is 0.71034 ± 0.00026 (95% confidence interval).

Thanks for pointing this out. In the literature, a “certified value” of 0.71034 is reported, as mentioned by the reviewer. An “accepted value” of 0.71025 is however also reported and has been used here. We believe the difference does not substantially change the conclusions, and is within the 95% confidence interval.

Where are the supplementary references?

Corrected. In the initial submission references from the main text and SI were merged into one, for the revision the two documents have been split.

Referee #4 (archaeogenetics, infectious disease):

The manuscript is very well written and the results are very well presented. Particularly exciting is the relationship between the groups studied, which reflects a certain correlation between the individual burial groups in Sweden.

The results on the early plague cases are very exciting and the findings are plausible. However, the interpretation of the results should be put into perspective. The graves do not show an increased mortality, so that it cannot be assumed that there was an excess mortality as a result of a pneumogenic outbreak. The high infestation rather speaks for a chronic disease. I would therefore suggest softening the authors' interpretation. An epidemic outbreak and also single zoonoses have been disproven, so it seems to be a middle way.

We agree. Following this comment and similar comments from reviewers one and five, we have decided to rewrite the entire section. In this paragraph we have outlined our interpretation of the data, along with alternative interpretations. We have also clarified that assumptions of excess mortality are speculations. We believe that this rewritten paragraph addresses this discussion in a more nuanced way, with a better representation of alternative interpretations of the data at hand. The rewritten paragraph now reads:

The distribution of plague positive individuals in the pedigree presented in Figure 3, does not readily support a swift and deadly plague epidemic, because plague is detected in all generations except generation two. However, when information on the different plague strains is taken into account, it becomes apparent that plague infections are stratified both chronologically and phylogenetically into two separate clusters. The most ancestral form of the plague (strain C) is also detected in the oldest individual from this study (FRA102) who was buried in the northern part of the chamber with individuals from generations one and two despite being unrelated to everyone in the pedigree. Based on placements of lower-coverage genomes, we also identify a plague form similar to strain C in FRA021, the progenitor of the right-side subfamily. This finding suggests a spread of an early form of the disease in the first generation of the pedigree. The mortality rate of this form of the plague is unknown, but the pedigree clearly illustrates that the family as a whole and the line of FRA021 survived the disease.

The second cluster of plague infections occur in generations 3-5 and is caused by strains A and B. Strain A is found in generation four of the left-side family, where all individuals appear to have been infected by the plague. Strain B on the other hand, is found in a single individual from the right-side subfamily, most likely from generation three. Given the genetic distance between strains A and B and their different distribution across the pedigree, it is possible that the two strains represent separate outbreaks of the disease: one in the left-side subfamily (strain A), and another in the right-side subfamily (strain B). Even though both strains appear to be contemporaneous in Figure 3, they could be temporally distinct as the error margins of the chronological modelling allows for considerable variation in the modelled dates (Supplementary Figure 24). Unfortunately, it is not possible to further assess this

hypothesis as we are unable to distinguish strains A and B in the lower-coverage genomes (Supplementary Figure 8).

We can only speculate on the impact that the disease had on the local population, but we note that all of the hallmarks expected in a swift and deadly epidemic is present for the spread of strain A in the left-hand subfamily: 1) The frequency of plague positive individuals is exceptionally high 2) The disease is restricted to the last two generations, and 3) all higher-coverage genomes from this subfamily are identical. Taken together, these observations could suggest that an outbreak of strain A led to the demise of the left-side subfamily, perhaps driven by increased pathogenicity due to recombination around the ypm locus in this strain (discussed below).

And:

*Given the presence of ancestral *Y.pseudotuberculosis* variation in the preLNBA and LNBA-strains, it is possible that these plague forms followed a faecal-oral transmission route and displayed attenuated pathogenicity. Yet, the presence of the ypm gene, and in particular, the finding of a hitherto unknown combination of genes in the ypm locus for strain A could suggest increased virulence. Hence, while the infectivity, morbidity and mortality might have varied among these early plague strains, it is possible that a recombination event in the ypm locus rendered strain A virulent. In combination with other factors, and perhaps other diseases, strain A could thus have played a role in the Neolithic Decline. Yet, all individuals from pedigree 1 were buried in the Fräsegården tomb, and someone must have survived to bury them. Furthermore, the demographic profile of the graves do not indicate catastrophic mortality¹³, which together with the high infection rates could suggest a less severe or chronic disease manifestation. Lastly, it is well established that the later LNBA- clade of plague was prevalent and widespread across Eurasia for more than 2,000 years without affecting the Bronze Age population sizes considerably⁸.*

Referee #5 (archaeogenetics, infectious disease):

A. Summary of the key results

In this study, with title “Repeated Outbreaks of Plague Across Five Generations of Neolithic Farmers” Seersholm and colleagues present aDNA screening of 174 samples and subsequent human genomic analysis of 109 individuals from Neolithic farmer communities in nine Scandinavian archaeological sites. The authors additionally present a metagenomic analysis of all individuals, identifying 18 *Y. pestis* infections (some of which are tentative) as well as the potential detection of additional pathogens such as *Borrelia recurrentis* and *Yersinia enterocolitica* in the studied datasets. The vast majority of presented data derive from a single site, Fräsegården, in Sweden, from where a pedigree of 35 individuals was reconstructed alongside four *Y. pestis* genomes with genomic coverage >1X. Human genomic and strontium isotopic results are used to infer a patrilineal social organisation among the studied communities. Finally, the authors make a major claim i.e., that the pathogen results shown here present sufficient evidence for the role of *Y. pestis* in the collapse of Neolithic populations of Scandinavia.

B. Originality and significance: if not novel, please include reference

- The vast majority of results presented in this study derive from the archaeological site of Fräsegården, in western Sweden, a site that has previously been subject to ancient DNA analysis in two publications, which show consistent results with those presented here. Specifically, previously studies have characterised both human aDNA and *Y. pestis* infections from Fräsegården (see Skoglund et al., *Science*, 2014 and Rascovan et al., *Cell*, 2019) and led to the reconstruction of a *Y. pestis* genome (Gökhem 2) identical to genomes characterised here. The study by Seersholm and colleagues is based on more individuals and, hence, contributes additional genomic data from the site.

- Moreover, the Neolithic kinship structures presented here are consistent, albeit with some expected distinctions, with previously published Neolithic communities from Britain (Fowler, Olalde et al., *Nature*, 2021) and France (Rivollat et al., *Nature*, 2022).

C. Data & methodology: validity of approach, quality of data, quality of presentation

-The study follows standard methods in the aDNA field and uses data generation and computational methodologies, both for human biological kinship reconstructions and for the phylogenetic placement of bacterial strains, that are on par with practices in the field.

- The paper is generally well written, although in parts overstatements are made. I find these to be unnecessary, for example:

L365 “...one swift plague outbreak that eradicated the entire Neolithic community in Falbygden.”

We agree. This section has been rewritten and the paragraph was deleted.

L447 “For the first time, we report numerous plague cases from a single community...”. This statement is inaccurate. Previous studies have also reported multiple *Y. pestis* cases from single prehistoric communities.

Corrected. We have rephrased the sentence to reflect this. The sentence now reads:

For the first time, we track the spread of multiple plague strains across an extended pedigree, suggesting that plague infection was common in the community.

L448 “could have been following the pneumonic transmission route...” While this may have been the case, there is currently no evidence towards its support, therefore it remains speculative.

We agree. As outlined above the sentence was rephrased.

D. Appropriate use of statistics and treatment of uncertainties

- The term “prevalence” is used frequently within the manuscript. Prevalence is an epidemiological parameter, which is non-trivial to estimate from incomplete and biased archaeological datasets, such as the one presented here. I strongly suggest the use a different term to characterise *Y. pestis* identification rates.

Corrected. We have replaced *prevalence* with either *frequency* or *rate* throughout the manuscript.

- Datasets of 0.01X coverage, where <1% of the *Y. pestis* genome is expected to be covered (Table S6) should not be called “genomes”, these are at best genome-wide data.

Corrected. Following reviewer one's comment on the use of ‘high coverage genomes’, we characterised the three coverage categories as ‘tentative detections’, ‘lower-coverage partial genome’ and ‘higher-coverage partial genomes’.

- Some of the genomes that seem identical based on the *Y. pestis* phylogeny (Figure 4) are consistent with having at least some SNP differences according to Supplementary Figure 10 (e.g., FRA005 and Gok2). How are these differences interpreted? Given the slow mutation rate of *Y. pestis*, these might reflect different infection timings. If true, the notion of contemporaneous infections in FRA005, FRA013 and Gok2 might need to be reconsidered.

We agree with the reviewer on this point. The grounds for classifying the Gok2 genome as ‘strain A’ were not clearly stated in the manuscript. Our decision was based on a visual inspection of the three SNPs in question, which suggested that these base calls might be false positives due the high damage in the sample. In order to properly address this issue, we carried out simulations of damaged data, visualised all relevant SNPs distinguishing strains A and B (Supplementary Figures 17 and 18), and included a Supplementary section addressing this:

Supplementary Note 4 - Classification of the Gok2 strain

It is challenging to assign the previously published Gok2 genome⁴ to a specific plague strain from this study (strain A, B, or C), because of its relatively low depth of coverage (1.82X on average) in combination with its high rates of C to T and G to A misincorporations at the 5' and 3' ends, respectively. As indicated in Supplementary Figure 10, the Gok2 genome shares the highest similarity with genomes of strains A and B. But, the number of different SNPs between Gok2, FRA013, FRA005 and FRA020 does not readily allow for a classification of Gok2, and the distances between these genomes are internally inconsistent because of missing data. E.g. based on the relatively limited number of SNPs called for Gok2, this genome is 100% identical to FRA020 (strain B) and FRA013 (strain A), but differs from FRA005 on three positions (see Supplementary Figure 17). As depicted in the figure, these SNPs share multiple similarities that might question their authenticity: 1) All three SNPs are C>T substitutions also caused by ancient DNA damage, 2) All three SNPs are covered only by three reads, which is the minimum threshold required to call a SNP in our pipeline, 3) Of a total of nine reads covering the SNPs in question, 5 reads have the alternate allele within 5bp of the read end. 4) Even though none of these SNPs were called in FRA020 and FRA013 both samples have reads supporting the reference allele. Based on these observations, we hypothesized that these three SNPs were incorrect base calls arising from C to T misincorporations.

*In order to test this hypothesis, we simulated data with similar sequencing profiles as the samples FRA020, FRA013, FRA005 and Gok2. For each sample, we simulated 100 replicates of *Yersinia pestis* fastq files with the same damage patterns, read length distributions and mean depth of coverage as the original sample, using the *Yersinia pestis* reference genome (GCF_000009065.1) as template (see Supplementary Methods - Supplementary Note 2). We then processed these fastq files following our pipeline for mapping and basecalling of plague data. As depicted in Supplementary Figure 19, we found that between zero and five incorrectly called SNPs (mean: 1.06) were called for Gok2, while no incorrect SNPs were called for the three high coverage genomes (FRA005, FRA013, FRA102). For the other low coverage sample FRA102 a single replicate had an incorrectly called SNP. As Gok2 and FRA102 have similar coverages the difference in false positive SNPs between the two samples can be attributed to the difference in damage: Gok2 has high rates of C>T and A>G misincorporations, while FRA102, which was USER-treated, has low damage levels. Based on this analysis, we conclude that the three SNPs unique to Gok2 are most likely incorrectly called variants due to damage.*

Having established that the SNPs unique to Gok2 are less reliable, we next turned to the SNPs unique to FRA020 (strain B) to classify Gok2 as either strain A or strain B. Of the three SNPs distinguishing strains A and B, only one SNP (position 726,406 on the main chromosome, NC_003143.1) is covered by any reads from Gok2. The two reads from Gok2 covering position 726,406 both support the allele from strain A (Supplementary Figure 18). Although this position is covered by only two reads from Gok2 at this position, we note that the SNP is an A->G substitution, and that the variant is located in the middle of both reads, suggesting that this pattern is unlikely to have arisen from ancient DNA damage. Accordingly, we tentatively assign Gok2 to strain A.

E. Conclusions: robustness, validity, reliability

- While the *Y. pestis* genomes presented here add new data to our understanding of prehistoric plague, I find their causal attribution to the Neolithic decline to be unsubstantiated. It is important to note that such early forms of plague have not only been identified during the period of the Neolithic decline, but were instead also present throughout the Bronze Age and all the way to the Iron Age (reflects the temporal interval of LNBA- lineages). Based on published data, infections with *Y. pestis* during prehistory seem to have been frequent across Eurasia for almost 3,000 years. Furthermore, these variants of *Y. pestis* were non-flea-adapted, meaning that their functional differences in terms of disease presentation, virulence and transmissibility compared to modern-day (and historical) plague are unknown and, so far, entirely hypothetical. Finally, societal collapse is an extremely complex phenomenon and a very challenging one to characterise in prehistoric contexts, therefore, its attribution to a single infectious disease seems highly simplistic and unrealistic.

Thank you for your comment. Following similar comments from reviewers one and four, we have rewritten the entire plague discussion to better capture the nuances of this discussion. We have also clarified that a casual relationship between the plague and the Neolithic decline is entirely hypothetical, and removed any assumptions of causality from the abstract. Please see our reply to reviewer #4 for the full rewritten discussion.

-The authors mention that their estimated frequency of 17% is an underestimation of the true *Y. pestis* frequency in the population. Such argument is questionable given that the presented estimates include some very low-abundance identifications that may not be reliable.

We agree that our lowest coverage samples have very low read counts, which could question the authenticity of these hits. We do however feel confident that these low coverage detections are reliable because of the stringent filtering applied in our screening pipeline in combination with some of the observations we made in the first phases of this project where we had very little data. The pathogen pipeline used here allows us to retain only hits to species that are the best hit within their genus, that have an even coverage, and that have high similarity to the reference. Furthermore, in several cases we initially identified a pathogen hit based on very few reads, that was later confirmed by deeper sequencing. For AVL001 for example, the pathogen pipeline initially identified plague based on 34 reads. This observation prompted us to sequence deeper, and we decided to generate two single stranded libraries, which enabled us to increase the number of plague reads to 7,743. When we subsequently applied capture to these two libraries the read count was increased further to 27,309 reads. We have never observed the opposite, e.g. where increased sequencing resulted in the loss of a plague hit.

To further test if our putative plague reads could represent environmental taxa not in our database, we carried out a blast analysis of all plague reads passing filters from samples with less than 0.01X coverage. This analysis consistently found *Yersinia pestis* to be the most common species among the top blast hits, followed closely by *Yersinia pseudotuberculosis*. We have included this analysis as Supplementary Figure 25:

Supplementary Figure 25. Blast results for tentative plague detections (<math><0.01X</math>). Number of reads where a given species was among the best blast hits as defined by the Blast eValue. Shaded bars represent the total number of input reads. Only the seven most common taxa are shown.

Lastly, we have clarified that these low coverage plague hits should be considered tentative detections in the main text by changing the label from 'low coverage detections' to 'tentative plague detections'.

- The authors seem to be clearly favouring a one-sided interpretation of their results, when several alternative possibilities for their findings can be considered. I strongly suggest the presentation of more balanced/nuanced statements in several part of the paper, for example:

L363 "Given the high prevalence of plague positive individuals in this study, and the relatively narrow time frame in which plague was detected, we hypothesised that our data represented one swift plague outbreak that eradicated the entire Neolithic community in Falbygden." Such claims appear exaggerated on the basis of four reconstructed *Y. pestis* genomes. Moreover, I disagree that a ~150-year interval is narrow and could reflect a "swift" plague outbreak.

We agree that this statement was misleading. The paragraph was removed.

L380 "This idea is supported by the differential plague prevalence in the different sub-branches of the pedigree: In the last two generations of the left subfamily plague is detected in 5/6 individuals (83%)." The authors should be aware that individuals from multiple generations within a pedigree can overlap in their lifetime, therefore, the *Y. pestis* frequency calculated for generation 5 does not reflect a prevalence estimate for a population, and is overall misleading.

Thank you for highlighting this. As part of the rewriting of the plague paragraph we removed the statement in questions.

L391 “Hence, it is likely that this female travelled to the Falbygden area during her lifetime, bringing the plague with her.” This statement makes the multi-fold assumption that (1) an individual was able to travel from Denmark to Sweden while being infected with a self-limiting and likely lethal pathogen, (2) that the source of the pathogen strain is consistent with that of the individual’s genomic and strontium background, and (3) that the region where the individual died could have not locally housed the retrieved pathogen diversity. Given that such assumptions are not supported by any evidence, alternative scenarios should have equal weight in the authors interpretation.

We agree that this statement made several unsupported conclusions. We have rewritten the entire discussion on plague, and have removed the statement in question. The new paragraph on FRA102 and strain C now reads:

The most ancestral form of the plague (strain C) is also detected in the oldest individual from this study (FRA102) who was buried with individuals from generations one and two despite being unrelated to everyone in the pedigree (Supplementary Figure 6). Based on placements of lower-coverage genomes, we also identify a plague form similar to strain C in FRA021, the progenitor of the right-side subfamily. This finding suggests a spread of an early form of the disease in the first generation of the pedigree.

L411 “Partitioning the variant graph nodes into groups based on their presence pattern in the three species, we found that early divergent ancient plague lineages (preLNBA-, Frälsegården, and LNBA-) harbour up to 50 kb gene content present in some or all *Y. pseudotuberculosis*/*Y. similis* strains that is absent in later strains (LNBA+, 1P, modern; Figure 4b).” These results may also support the idea of Neolithic strains being an intermediate between *Y. pestis* and other, less virulent, *Yersinia pseudotuberculosis* complex diversity, and therefore may alternatively suggest an attenuated pathogenicity in contrast to what the authors claim. Why is this possibility not considered?

Good point. We have addressed this in the last paragraph of the pangenomic analysis:

*Given the presence of ancestral *Y. pseudotuberculosis* variation in the preLNBA and LNBA- strains, it is possible that these plague forms followed a faecal-oral transmission route and displayed attenuated pathogenicity.*

- Related to the point above, there is considerable variation in the detection of the *ypM* superantigen among LNBA- genomes, even for genomes that are seemingly phylogenetically identical (e.g., Gok2, FRA005 and FRA013). Yet the presence of this gene is used as evidence of highly virulent infections in Neolithic individuals. The authors should clarify this interpretation as Figure 4 does not show clear presence of the gene in all strains.

We thank the reviewer for highlighting this. We agree that there is substantial variation in normalised gene coverage in Figure 4c. This is largely driven by stochastic variation in coverage, which for low coverage samples and short genes results in large variations from the average coverage. Our aim with Figure 4c was to highlight the overall organisation of genes around the *ypm* locus. We realise that the initial version of the figure did not achieve this goal. Hence, to simplify the figure, we converted normalised coverage into discrete presence/absence data, and focused exclusively on the *ypm* gene and the 11 open reading frames associated with *ypm* virulence in Carnoy et al. (2002). We added

presence/absence data around the three known *ypm* alleles, to better illustrate how the ancient strains relate to the modern *Y.pseudotuberculosis* strains. We believe that the updated figure provides a much clearer overview of our findings:

Furthermore, the new version of Figure 4c clearly illustrates that there is a noticeable difference in gene content for strains C and B on one hand, and strain A on the other hand. We addressed this finding in the main text:

In order to distinguish the three known ypm loci (ypmA, ypmB and ypmC) we characterised gene presence/absence in the genetic environment surrounding this unstable locus (figure 4). We identify a pattern of genetic organisation resembling the ancestral locus, ypmB, in RV2039, and Falbygden strains C and B, while we identify two hitherto unknown combinations of genes in the ypm locus for strain A and later LNBA- strains. Yersinia pseudotuberculosis strains carrying the ypmB allele have been associated with low pathogenicity in humans²⁹, however the virulence of the unknown ypm allele identified in strain A cannot be determined. It is, however, well established that some ypm alleles have epidemic potential.

Lastly, we moved the old version of figure 4c to Supplementary Figure 12.

F. Suggested improvements: experiments, data for possible revision

- In Table S10, the headers are not self-explanatory and so the content cannot be evaluated in detail. Nevertheless, the table is meant to present an overview of all pathogen candidates that have passed authenticity filters based on the authors screening approach. In that regard, do the authors consider all those identifications to be authentic (ancient) and, if so, why have they specifically chosen to mention only the concurrent infections with *B. recurrentis* and *Y. enterocolitica* in the main text?

Thank you for highlighting this. We have clarified the contents of this table by adding a line with column descriptions in the table, and by detailing more complex statistics in the table legend. Furthermore, we have highlighted that this table is meant as a 'raw' data table in the legend title:

Supplementary Table 10. Raw data from pathogen screening results. Raw data on pathogen hits that pass all filters, including relevant statistics used to evaluate the credibility of each hit. *coveragePRatio* is a measure of coverage evenness, quantified by normalising the observed breadth of coverage to the expected breadth of coverage given the number of mapped bases.

Lastly, to address the reviewers comment on which identifications we consider authentic, we have added a 'hitClassification' column where we interpret each hit as either environmental, oral microbiome, skin microbiome, or infection. We characterise the following hits as infections: *Yersinia pestis* (n=19), *Borrelia recurrentis* (n=5), *Yersinia enterocolitica* (n=4), *Human betaherpesvirus 6B* (n=1), *Human betaherpesvirus 7* (n=1) and *Rickettsia felis* (n=1). We have clarified this in the main text which now reads:

We identified five other pathogens that we consider authentic (Supplementary Table 10); of these two were identified in more than one individual: Yersinia enterocolitica, the causative agent of yersiniosis (4/109, 4%), and Borrelia recurrentis the cause of louse-borne relapsing fever (LBRF) (5/109, 5%).

- Related to the point above, libraries with as little as ~70 detected reads are considered confidently positive for *Y. pestis*. As far as I can understand from Table S6 several of these were not subjected to whole-genome capture. Given the high false-positive rates known for ancient pathogen DNA in metagenomic datasets, how can these identifications be secure?

Please see our reply above for a detailed answer regarding the authenticity of low coverage hits.

Considering the reviewer's point on capture enrichment for low coverage samples: We have implemented library complexity estimations as part of our plague mapping pipeline using preseq. These estimates provide a theoretical maximum coverage that can be obtained from a given library. We have only carried out capture enrichment on libraries where the estimated maximum coverage is substantial.

In order to test the accuracy of our complexity estimations, we included a couple of low coverage plague samples (FRA021, HUN002 and CGG106001) in our first round of capture. This test confirmed that capture enrichment is not worthwhile in libraries where the maximum estimated coverage is low.

- Further related to Table S6, the number of identified *Y. pestis* reads are not fully informative if the total sequencing effort is not stated. This is especially true for low-abundance identifications which are of lower certainty if sequencing efforts are very deep.

Corrected. We have included the total number of reads sequenced and mapped plague reads per read sequenced (in parts per million) in Supplementary Table 6.

- Previous studies have presented vastly different results for the divergence between *Y. pestis* and *Y. pseudotuberculosis* (see Rasmussen et al., Cell 2015, Rascovan et al., Cell, 2019 and Susat et al., Cell Reports, 2021). Given the new genomes presented here, a molecular dating analysis including all new and previously published Neolithic datasets may provide new insights to this discussion.

We agree that a molecular dating analysis is warranted in this paper. We have decided to focus on dating the most recent common plague ancestor and other early events of plague evolution, as this is where our data set is strongest and where our narrative is focused. In order to properly date the divergence time between *Y.pestis* and *Y.pseudotuberculosis* we would need to include representatives of the entire *Y.pseudotuberculosis* diversity as in Rasmussen et al. 2015, but this is beyond the scope of this paper.

We took two different approaches to date our plague phylogeny: a BEAST based approach following Andrades-Valtueña et al. (2022) and an approach based on BactDating. We found that both approaches dated the most recent common ancestor of *Yersinia pestis* to around 5,250 yBP, and we generally found a large overlap in all divergence dates between the two methods. We included these results as Supplementary Figure 23:

Supplementary Figure 23. Molecular dating of ancient plague strains. Dating analysis were carried out on preLNBA strains (grey and pink color) and LNBA- strains (collapsed in grey triangle). The modelled chronological span of the pedigree from Fräsegården is highlighted in blue. **a)** BEAST analysis using the Coalescent Bayesian Skyline demographic model and assuming an Optimised Relaxed Clock and the GTR substitution model with four gamma categories and empirical frequencies. **b)** BactDating analysis using the 'relaxedgamma' model.

G. References: appropriate credit to previous work?

- The authors should clearly state within the main text that the previously published Gok2 *Y. pestis* genome and the FRA genomes presented here are from the same archaeological site and temporal horizon.

Corrected. We have included this information in the section on plague diversity:

[...] FRA013 and FRA005 were identical to the previously published Gökhem2 genome from the same archaeological site and time period⁴.

H. Clarity and context: lucidity of abstract/summary, appropriateness of abstract, introduction and conclusions

- The abstract is presented clearly with the exception of the concluding sentence. "Taken together, our findings provide the first direct reconstruction of plague transmission within a large patrilineal kinship group and suggest that widespread ancient plague epidemics in Neolithic Scandinavian populations played a pivotal role in their collapse." I find this statement to be misleading as the manuscript does not formally infer pathogen transmission dynamics in this community, neither does it present evidence for population collapse.

Corrected. We have rephrased the sentence replacing the word 'transmission' with 'spread' and avoiding any assumptions on causality related to the Neolithic decline:

Taken together, our findings provide the first direct reconstruction of plague spread within a large patrilineal kinship group and place widespread plague infections at the beginning of the Neolithic Decline.

- Finally, I find that the title overstates the results and conclusions of the study, given that the four presented *Y. pestis* genomes are insufficient evidence of "repeated outbreaks". First, the accompanying data on strains A and B do not conclusively show that they represent different time points and may well reflect diversity present contemporaneously within Frälsegården. Importantly, the individual associated with strain B has not been radiocarbon dated. [...]

Thank you for noticing this. We do have a date for the individual with strain B, but since this individual is not securely placed in the pedigree it was not possible to model the date. We had therefore decided to exclude the date from Figure 3 to avoid confusion. We realise that it is just as confusing to exclude the date as to include it.

To fix this, we have included the date and added an asterisk indicating that the date is not modelled. The date of this individual is 4919 BP (4973-4857), which suggests that this boy is younger than generation one where we initially placed him. Furthermore, given the morphological age of the individual (12-13 years) it is unlikely that he produced offspring, and accordingly he is probably the grandchild or nephew of the progenitor. Given the young date of this individual, we hypothesise that he is a grandchild to the progenitor, and have placed him in generation three.

In the rewritten discussion on plague we have clarified that the spread of strains A and B do not necessarily represent temporally distinct events:

The second cluster of plague infections occur in generations 3-5 and is caused by strains A and B. Strain A is found in generation four of the left-side family, where all individuals appear to have been infected by the plague. Strain B on the other hand, is found in a single individual from the right-side subfamily, most likely from generation three. Given the genetic distance between strains A and B and their different distribution across the pedigree, it is possible that the two strains represent separate outbreaks of the disease: one in the left-side subfamily (strain A), and another in the right-side subfamily (strain B). Even though both strains appear

to be contemporaneous in Figure 3, they could be temporally distinct as the error margins of the chronological modelling allows for considerable variation in the modelled dates (Supplementary Figure 24). Unfortunately, it is not possible to further assess this hypothesis as we are unable to distinguish strains A and B in the lower-coverage genomes (Supplementary Figure 8).

Second, there is no evidence of strain C spreading beyond the single outlier individual from which it was retrieved and, therefore, is insufficient to indicate an outbreak. In addition, its date seems questionable given its basal positioning in the phylogeny (and short branch length), which comes in contrast with the individual's retrieved younger radiocarbon date.

We agree with the reviewer. We were also doubting the date of FRA102 for the same reasons and because this date was carried out on a different bone than the tooth which the DNA analysis was based on. Hence, we decided to obtain a second date using the leftover material from the DNA analysis (the crown). This new date confirmed the reviewers suspicion and placed FRA102 as the oldest individual in the grave with a calibrated age of 5,160 BP (5,317-5,054). While this date is the oldest from this site, it is well within the unmodelled age ranges of other individuals from the first generations (see the new Supplementary Figure 24 on chronological modelling). We thus conclude that her lifetime overlapped with individuals from the first generations of the family, with whom she was also buried.

To address the reviewers second point, that strain C could represent a single outlier, we further investigated the placements of lower-coverage genomes. While it is not possible to distinguish strains A and B based on these placements, we find that two samples have ancestral plague forms similar to strain C: AVL001 and FRA020. AVL001 is an individual from Denmark with a relatively early radiocarbon date 5,160 BP (5,305-5,047), and FRA020 is the progenitor of the right-side subfamily of pedigree 1. This finding supports an early outbreak of strain C in generation one of the family with at least two infected individuals. We have addressed these points in the rewritten discussion on plague:

The distribution of plague positive individuals in the pedigree presented in Figure 3, does not readily support a swift and deadly plague epidemic, because plague is detected in all generations except generation two. However, when information on the different plague strains is taken into account, it becomes apparent that plague infections are stratified both chronologically and phylogenetically into two separate clusters. The most ancestral form of the plague (strain C) is also detected in the oldest individual from this study (FRA102) who was buried in the northern part of the chamber with individuals from generations one and two despite being unrelated to everyone in the pedigree. Based on placements of lower-coverage genomes, we also identify a plague form similar to strain C in FRA021, the progenitor of the right-side subfamily. This finding suggests a spread of an early form of the disease in the first generation of the pedigree. The mortality rate of this form of the plague is unknown, but the pedigree clearly illustrates that the family as a whole and the line of FRA021 survived the disease.

Referee #6 (megalithic archaeology):

A. Summary of the key results/validity

This is an important paper which sells itself (in its title) primarily on the identification of plague across five generations during the Neolithic. This is certainly a very interesting and significant discovery, with ramifications for our understanding of possible Late Neolithic population decline. This finding will be of interest to historians of disease (and others working in the biological sciences more widely) as well as archaeologists. The paper is also important with regard to its discussion of other elements of Neolithic society in Sweden, including evidence for patrilinearity/patrilocality and kinship/spatial relations within the tomb – these are important issues that will be of significant interest to Neolithic scholars in particular.

The manuscript does not have any flaws, to my knowledge, that should prohibit its publication. In fact, I found the paper to be well written, measured in its tone, and sensible in its argument and conclusions. It pushes the boundaries of interpretation, but (except at the very end – see below) does not push them too far, or unnecessarily hard.

B. Originality and significance

The paper is original in terms of its identification of plague in relatively high numbers of individuals, across an extended period of time (five generations). It identifies different strains of plague and unpicks their chronological and ancestral relationship well. The paper is certainly of interest to archaeologists (within and also beyond those working on the Neolithic of Europe) and will, I believe, be of interest to historians of disease, etc. as well. In addition to these findings, the evidence laid out for patrilinearity/patrilocality is also very original and interesting for prehistorians.

C. Data & methodology: validity of approach, quality of data, quality of presentation
I am not an expert in aDNA and therefore will not attempt to comment on that element of the paper's methodology. The archaeology is dealt with well and the decisions made are clear. The quality of data and presentation are good.

D. Appropriate use of statistics and treatment of uncertainties

This is not my area of expertise (especially in relation to aDNA) and so I do not feel qualified to comment. The assumptions with regard to the radiocarbon dating seem sensible to me.

E. Conclusions: robustness, validity, reliability

The broad conclusions drawn from the study are, as mentioned above, rigorous, measured and appropriate. The few issues I have relate to clarity of expression and thought process. As they are all fairly minor, I have outlined them in the 'suggested improvements' section below.

F. Suggested improvements: experiments, data for possible revision

Line 28. “detected in at least 17% of the population” – please be clearer that this is 17% of the sampled population, which itself is a probably skewed sub-sample (i.e. the people who were actually buried in tombs which is likely to be a substantial minority) of the total population.

Corrected. We have addressed this in the *Perspectives and Conclusions* paragraph, which now reads:

Yet, it is worth mentioning that the plague rate of 17% reported here does not necessarily reflect the true prevalence of the disease. For example, the plague detection rate might not be representative of the population as a whole, as it is a measure of disease frequency within the sampled population, which is restricted to well-preserved individuals buried in tombs. Furthermore, only a fraction of plague positive cases is expected to carry detectable levels of DNA from Yersinia pestis. In Schueneman et al.³³, a qPCR screening of known plague victims showed a detection rate of 5.7% in bones and 37% in teeth, suggesting that the true frequency of the Falbygden plague could be significantly higher than 17%.

Line 234-236. The argument would benefit from some clarification here. The second interpretation (“all their offspring were daughters, who were married out and buried in other tombs”) is to my mind much more probable than the first (“but did not produce offspring before they passed away”). If the authors want to keep this line in, they should be clearer that the first is actually “but did not produce offspring *who were buried within the tomb and have therefore been sampled* before they passed away”. This is an important distinction.

Corrected. We have focused on the second interpretation. The text now reads:

This finding confirms the patrilineal social structure at the site and suggests that these 6 women were married into the family but did not produce offspring who were buried within the tomb. While it is possible that these women did not give birth before passing away, it is perhaps more likely that all their offspring were daughters, who moved away and were buried in other tombs.

Line 324. “Surprisingly” – why is this surprising? Remove or clarify.

Corrected, the text now reads:

Plague positive individuals were found not only in the Falbygden area, but as far south as Zealand, Denmark and by the Swedish coast north of Gothenburg.

Line 364. “we hypothesised that our data represented one swift plague outbreak that eradicated the entire Neolithic community in Falbygden”. This seems an odd hypothesis given that, as explained in Lines 373-374, it occurred over 5 generations and probably over more than a century according to the C14 dates. If this was just a playful experiment, explain that. If not, explain why this was assumed given the time distance.

We agree and the paragraph in question has been removed.

Line 399. “the high number of infected individuals analysed here suggests a relatively rapid spread of the disease within the population”. Again, clarify, given the 5 generations/120 year estimate.

We agree. The text now reads:

[...] but the high number of infected individuals analysed here suggests that the disease was able to spread within the population.

Lines 429-430. Can these “lineages” be tied into the proposed Strains A/B/C mentioned previously? If they are the same, reference the latter here as well.

Yes. This statement was inaccurate and the sentence have been rephrased:

By the end of the Neolithic at least three main plague lineages had evolved; the most ancestral RV2039 lineage¹⁰, the Falbygden clade (strains A,B, and C) and the lineage that would eventually evolve into the Bronze Age radiation of plague⁸.

Line 457. “We show that the social organisation in Middle Neolithic Sweden was patrilineal...”. As touched upon above, it might be worth clarifying that this is actually “We show that the social organisation in Middle Neolithic Sweden *as represented by the population sampled within these tombs* was patrilineal...”.

Thank you for the suggestion, it was adopted in the text as follows:

We show that the social organisation in Middle Neolithic Sweden, as represented by the population sampled within these tombs, was patrilineal and patrilocal.

Line 474. I do not think “appears to have been married off to...” is an appropriate term to use here as it implies a lack of agency on the part of the female (which is not certain) and a knowledge of “marriage” in the Neolithic (that we do not have). Could the wording simply be “appears to have moved to...”?

We agree, the sentence has been rephrased:

[...] the female appears to have moved to a neighbouring group where she established her own family.

Lines 478-482. Within what is a really good paper, which generally does not overstate its findings, I did not feel that this ending fitted well, as it makes several (presumably intended to be headline grabbing) inaccurate claims. I suggest simply deleting this paragraph or re-wording it.

Thanks for the comment. We have followed the reviewers suggestions below to improve the paragraph, as we prefer to end the paper with a last unifying statement. As a whole, it now reads:

Taken together, the data presented here provides a highly detailed and intimate snapshot of what life was like in Neolithic Falbygden, Sweden. The social structure was organised along male kinship lines, and females generally came from other kin groups. As plague was infecting a significant proportion of the sampled population, excess mortality associated with the disease could have undermined long term viability of society, leading to the eventual collapse of this form of Neolithic society.

- “The social structure strongly favoured males” – we do not know this, we only know/surmise that the society was patrilocal/patrilineal, which is not the same thing.

Corrected. The text was reworded to:

The social structure was organised along male kinship lines [..]

- “females moved away” – this is not true, we have direct evidence for one female moving away.

Corrected. The sentence now reads:

[...] and females generally came from other kin groups.

- “was infecting large parts” – this again is not well worded, “was infecting a significant proportion” might be more accurate.

Corrected.

- “it [plague] may have contributed to undermining the social viability of society, leading to the collapse of this form of Neolithic society”. I do not understand what is meant here – surely plague killed people which could have led to population “collapse”? If this is not what is meant, please clarify exactly how plague relates to society in more than a biological sense.

We agree. We have clarified that we mean ‘excess mortality’ caused by the disease:

As plague was infecting a significant proportion of the population, excess mortality associated with the disease could have undermined long term viability of society, leading to the eventual collapse of this form of Neolithic society.

G. References: appropriate credit to previous work?

Yes.

H. Clarity and context: lucidity of abstract/summary, appropriateness of abstract, introduction and conclusions

Good.

Duncan Garrow, University of Reading, UK

Reviewer Reports on the First Revision:

Referees' comments:

Referee #1 (Remarks to the Author):

I have carefully read the responses from the coauthors. All my concerns have been satisfactorily addressed. I thus recommend publication at that stage.

Referee #2 (Remarks to the Author):

The authors have dealt well with most of my comments.

For reasons of transparency, I'd prefer for Figures S14, S15 to also plot the outlying dates, and please also keep them within the OxCal code. OxCal has well-established routines to identify outliers, and it downweights outliers during the modelling, so removing them manually from the code/Figures it neither necessary nor recommendable.

I still think that the font of the ages in Figs. 3, S3 and S5 are too small to be readable, and would still recommend e.g. a colour gradient to indicate ages (as in Figure S24), but if the size of this font is acceptable to the Editor then that's fine with me.

Lines 727, 733, please use superscript for 13 and 15.

Referee #3 (Remarks to the Author):

This is a revised version of a manuscript submitted previously. I am satisfied that the authors have addressed all of my comments. I believe that the paper is now acceptable for publication.

Referee #4 (Remarks to the Author):

Thank you for addressing the comments and criticisms to my complete contentment.

All the best

Ben Krause-Kyora

Referee #5 (Remarks to the Author):

In their revision, Seersholm and colleagues have included valuable clarifications and new analyses to their study. I appreciate the new molecular dating analysis, the analysis of strain distinctions and the toning down of some key overstated conclusions in the paper. Still, I find that some points are insufficiently addressed and some discussion statements are confusing. These are:

- I disagree with the statement of repeated outbreaks among the studied Neolithic communities. As the current data suggests infections of more than two individuals associated with strain A, and as multiple individuals from the left pedigree were infected with *Y. pestis* at the time of death, this may -at most- be consistent with a small sustained outbreak associated with strain A. However, the data are insufficient to support repeated outbreaks since strains B and C are more scarcely identified (one genome representative from each strain). Importantly, while some lower-coverage datasets are consistent with the phylogenetic placement of strain C, this does not conclusively support that they were carrying identical strains. Therefore, the title “Repeated outbreaks of plague across five generations of Neolithic farmers” is misleading and should be toned down. I suggest “Repeated *Yersinia pestis* infections across five generations of Neolithic farmers”.
- Regarding the strain assessment of low coverage genomes, how do the authors interpret the contrasting phylogenetic placement of the low coverage dataset Gok4 (FRA004)?
- Regarding the argument “For the first time, we track the spread of multiple plague strains across an extended pedigree”, the word spread should be replaced with presence, as the current study resolution does not allow the monitoring of infection spread within this community and is overstating the fact that three strains (two of which may have been contemporaneous) have been identified.
- Regarding the gene presence/absence analysis, in Figure 4c the authors mention that genes are considered to be present when “gene coverage is higher than 10% the mean depth of coverage for that sample”. Here it is unclear whether the authors focus on depth or breadth of coverage for this analysis. I suggest to use breadth as this will give more information on whether a full or truncated version of a gene is present. For genomes where >90% of the genome covered at 1X, one would expect the full gene to be covered if present in its full-length version.
- Relevant to the above comment, I suggest to include a graphic on ypm variants ypmA, ypmB and ypmC so that they are more clearly discerned between genomes. I find the argument of the unknown combination of genes identified for strain A supposedly indicating a higher virulence to be unconvincing. If the same argument is inverted, then strains B and C likely did not have epidemic potential, therefore further diminishing the “repeated outbreak” conclusions made in this paper. Overall, there is insufficient evidence to make conclusions about the virulence of strains A, B or C based on their gene content.
- Regarding the molecular dating analysis, I was unable find any information on a temporal signal assessment. Was the suitability of the present dataset for molecular dating evaluated?

Referee #6 (Remarks to the Author):

The authors have taken on board all of my comments and responded appropriately to them all.

Author Rebuttals to First Revision:

General remarks to reviewers

We would like to thank the reviewers for their endorsement. We have addressed the remaining comments from reviewers two and five below.

Furthermore, please note that we have reversed the naming of strains A,B and C, so that the most ancestral strain becomes strain A, and the more derived and more prevalent strain is called strain C. Lastly, we have added another round of plague capture data for the samples FRA102 and FRA106, which brings the coverage up to 4.5 and 1.1 for these two samples, respectively. This data further strengthens the phylogenetic placement of the ancestral strain from FRA102, and enables us to tentatively assign the genome from FRA106 to strain C (formerly called strain A), but otherwise this does not change any of our conclusions. We have updated all plague plots and tables to incorporate the new data.

REVIEWER COMMENTS:

Referee #1 (archaeogenetics):

I have carefully read the responses from the coauthors. All my concerns have been satisfactorily addressed. I thus recommend publication at that stage.

Referee #2 (radiocarbon dating):

The authors have dealt well with most of my comments.

For reasons of transparency, I'd prefer for Figures S14, S15 to also plot the outlying dates, and please also keep them within the OxCal code. OxCal has well-established routines to identify outliers, and it downweights outliers during the modelling, so removing them manually from the code/Figures it neither necessary nor recommendable.

We have changed these figures and the Oxcal code as the reviewer suggests.

I still think that the font of the ages in Figs. 3, S3 and S5 are too small to be readable, and would still recommend e.g. a colour gradient to indicate ages (as in Figure S24), but if the size of this font is acceptable to the Editor then that's fine with me.

Thank you for your comment. We have kept the figures unchanged in this version of the manuscript, but we will of course follow any recommendations set by the editor/typesetting team on this matter.

Lines 727, 733, please use superscript for 13 and 15.

Corrected.

Referee #3 (archaeology/isotope analysis):

This is a revised version of a manuscript submitted previously. I am satisfied that the authors have addressed all of my comments. I believe that the paper is now acceptable for publication.

Referee #4 (archaeogenetics, infectious disease):

Thank you for addressing the comments and criticisms to my complete contentment.

All the best

Ben Krause-Kyora

Referee #5 (archaeogenetics, infectious disease):

In their revision, Seersholm and colleagues have included valuable clarifications and new analyses to their study. I appreciate the new molecular dating analysis, the analysis of strain distinctions and the toning down of some key overstated conclusions in the paper. Still, I find that some points are insufficiently addressed and some discussion statements are confusing. These are:

- I disagree with the statement of repeated outbreaks among the studied Neolithic communities. As the current data suggests infections of more than two individuals associated with strain A, and as multiple individuals from the left pedigree were infected with *Y. pestis* at the time of death, this may -at most- be consistent with a small sustained outbreak associated with strain A. However, the data are insufficient to support repeated outbreaks since strains B and C are more scarcely identified (one genome representative from each strain). Importantly, while some lower-coverage datasets are consistent with the phylogenetic placement of strain C, this does not conclusively support that they were carrying identical strains. Therefore, the title “Repeated outbreaks of plague across five generations of Neolithic farmers” is misleading and should be toned down. I suggest “Repeated *Yersinia pestis* infections across five generations of Neolithic farmers”.

Thank you for the comment. We have followed the suggestion and reworded the title to:

Repeated Plague Infections Across Six Generations of Neolithic Farmers

Furthermore, we have replaced 'outbreaks' with 'infection events' throughout the manuscript, except from cases where we are specifically talking about strain C (formerly called strain A).

- Regarding the strain assessment of low coverage genomes, how do the authors interpret the contrasting phylogenetic placement of the low coverage dataset Gok4 (FRA004)?

Thank you for noticing this inconsistency. In order to address this, we investigated which SNPs in the Gok4 genome were responsible for the placement. We found that four SNPs drove this inconsistency. All four SNPs were located in the region between position 3,000 and 4,200 on the PCP1 plasmid, which has previously been shown to be a problematic region because of high similarity to expression vectors (Schuenemann et al. 2011, PNAS). To fix this issue, we added this problematic region to our *Yersinia pestis* mask, and reran the placement. In the resulting placement, the Gok4 genome is placed as expected:

In the section ‘*Yersinia pestis* MQ0 mask’ under Methods, we have outlined the addition of the mask for the PCP1 region:

Lastly, we also filtered out the region from position 3,000 to position 4,200 on the PCP1 plasmid, as this region can be problematic because of high similarity to expression vectors.

- Regarding the argument “For the first time, we track the spread of multiple plague strains across an extended pedigree”, the word spread should be replaced with presence, as the current study resolution does not allow the monitoring of infection spread within this community and is overstating the fact that three strains (two of which may have been contemporaneous) have been identified.

Corrected.

• Regarding the gene presence/absence analysis, in Figure 4c the authors mention that genes are considered to be present when "gene coverage is higher than 10% the mean depth of coverage for that sample". Here it is unclear whether the authors focus on depth or breadth of coverage for this analysis. I suggest to use breadth as this will give more information on whether a full or truncated version of a gene is present. For genomes where >90% of the genome covered at 1X, one would expect the full gene to be covered if present in its full-length version.

We agree that the wording was unclear. The measure we are using to define gene presence/absence is in fact the ratio of observed over expected breadth of coverage. We have clarified this in the figure legend which now reads:

> *A gene is defined as being present if the ratio of observed over expected breadth of coverage (given the depth of coverage across the genome) is over 10% for that gene/sample combination (see Supplementary Figure 12).*

• Relevant to the above comment, I suggest to include a graphic on ypm variants ypmA, ypmB and ypmC so that they are more clearly discerned between genomes. I find the argument of the unknown combination of genes identified for strain A supposedly indicating a higher virulence to be unconvincing. If the same argument is inversed, then strains B and C likely did not have epidemic potential, therefore further diminishing the "repeated outbreak" conclusions made in this paper. Overall, there is insufficient evidence to make conclusions about the virulence of strains A, B or C based on their gene content.

To address this, we have added information on the ypm variants (ypmB-like, unknown variant 1, unknown variant 2, and ypm negative) for all strains analysed in Figure 4c. Furthermore, we have clarified that the depiction of the three known ypm variants are distinct from the plague data by separating them from the rest of the figure with a stippled line:

Moreover, we agree that strains A and B (formerly strains C and B) most likely had a limited epidemic potential. As we have avoided the use of 'outbreak' for strains A and B following the comment above, we now consider our interpretation to be consistent. Lastly, we hope that it is clear that all interpretations of the virulence of strains A, B or C are speculations based on the data at hand. If this is not clear, we are of course happy to revise accordingly.

- Regarding the molecular dating analysis, I was unable find any information on a temporal signal assessment. Was the suitability of the present dataset for molecular dating evaluated?

Yes, it was, but it was not included in the manuscript. To fix this, we have added the root-to-tip analysis as Supplementary Figure 26:

Rate=1.60e-02, MRCA=-5164.77, R2=0.98, p<1.00e-04

Supplementary Figure 26. Temporal signal assessment. Root-to-tip analysis carried out using *BactDating*.

We have referenced this figure in the supplementary section describing the *BactDating* analysis:

Before running the main analysis we checked the temporal signal in the data by running a root to tip analysis, which showed a strong correlation between sample age and distance to the root (R²:0.98, Supplementary Figure 26).

Referee #6 (megalithic archaeology):

The authors have taken on board all of my comments and responded appropriately to them all.

Reviewer Reports on the Second Revision:

Referees' comments:

Referee #5 (Remarks to the Author):

The authors have satisfactorily addressed the technical aspects of my comments. However, while agreeing with my previous comments that attributing the Neolithic Decline to plague is oversimplistic based on this locally focused study, that infection spread could not be directly demonstrated through the present data, and that the functional consequences of virulence profiles are only speculated, several such statements have remained in the abstract of the paper. These statements can mislead, as the presented results do not solve or satisfactorily address the “contentions” stated in the opening sentences of the abstract. The abstract should present a fair overview of the study’s outcome and scale.

Author Rebuttals to Second Revision:

Comments from reviewer #5

The authors have satisfactorily addressed the technical aspects of my comments. However, while agreeing with my previous comments that attributing the Neolithic Decline to plague is oversimplistic based on this locally focused study, that infection spread could not be directly demonstrated through the present data, and that the functional consequences of virulence profiles are only speculated, several such statements have remained in the abstract of the paper. These statements can mislead, as the presented results do not solve or satisfactorily address the “contentions” stated in the opening sentences of the abstract. The abstract should present a fair overview of the study’s outcome and scale.

We are happy to hear that the last technical comments have been addressed satisfactorily. To address the last comment we have rephrased the two sentences in the abstract on the Neolithic Decline. Specifically, we have avoided the use of ‘contentious’ in the second sentence of the abstract, and we have rewritten the last sentence which now reads:

Taken together, our findings provide a detailed reconstruction of plague spread within a large patrilineal kinship group and identify multiple plague infections in a population dated to the beginning of the Neolithic Decline.